

# Mapping and Assessing Variability in the Antarctic Marginal Ice Zone,
# the Pack Ice and Coastal Polynyas
Julienne C. Stroeve[1,2], Stephanie Jenouvrier[3,4] , G. Garrett Campbell[1], Christophe Barbraud[4] and
Karine Delord[4]
[1]National Snow and Ice Data Center, Cooperative Institute for Research in Environmental
Sciences, University of Colorado, Boulder, CO, USA
[2]Center for Polar Observation and Modelling, University College London, London, UK
[3]Woods Hole Oceanographic Institution, Woods Hole, MA, USA
[4]Centre d'Etudes Biologiques de Chizé, UMR 7372 CNRS, 79360 Villiers en Bois, France
# Abstract
Sea ice variability within the marginal ice zone (MIZ) and polynyas plays an important role for
phytoplankton productivity and krill abundance. Therefore mapping their spatial extent, seasonal and
interannual variability is essential for understanding how current and future changes in these biological
active regions may impact the Antarctic marine ecosystem. Knowledge of the distribution of different
ice types to the total Antarctic sea ice cover may also help to shed light on the factors contributing
towards recent expansion of the Antarctic ice cover in some regions and contraction in others. The long-
term passive microwave satellite data record provides the longest and most consistent data record for
assessing different ice types. However, estimates of the amount of MIZ, consolidated pack ice and
polynyas depends strongly on what sea ice algorithm is used. This study uses two popular passive
microwave sea ice algorithms, the NASA Team and Bootstrap to evaluate the distribution and variability
in the MIZ, the consolidated pack ice and coastal polynyas. Results reveal the NASA Team algorithm has
on average twice the MIZ and half the consolidated pack ice area as the Bootstrap algorithm. Polynya
area is also larger in the NASA Team algorithm, and the timing of maximum polynya area may differ by
as much as 5 months between algorithms. These differences lead to different relationships between sea
ice characteristics and biological processes, as illustrated here with the breeding success of an Antarctic
seabird.
# 1. Introduction
Changes in the amount of the ocean surface covered by sea ice play an important role in
the global climate system. For one, sea ice and its snow cover have a high surface reflectivity,
or albedo, reflecting the majority of the sun's energy back to space. This helps to keep the polar
regions cool and moderates global climate. When sea ice melts or retreats, the darker (lower
albedo) ocean is exposed, allowing the ocean to absorb solar energy and warm, which in turn
melts more ice, creating a positive feedback loop. During winter, sea ice helps to insulate the
ocean from the cold atmosphere, influencing the exchange of heat and moisture to the
atmosphere with impacts on cloud cover, pressure distribution and precipitation. These in turn





can lead to large-scale atmospheric changes, affecting global weather patterns [e.g. *Jaiser et*
*al.*, 2012]. Sea ice also has important implications for the entire polar marine ecosystem,
including sea ice algae, phytoplankton, crustaceans, fish, seabirds, and marine mammals, all of
which depend on the seasonal cycle of ice formation in winter and ice melt in summer. For
example, sea ice melt stratifies the water column, producing optimal light conditions for
stimulating bloom conditions that Antarctic sea birds rely upon for their breeding success and
survival [e.g. *Park et al.*, 1999].
In stark contrast to the Arctic, which is undergoing a period of accelerated ice loss in the last
several decades [e.g. *Stroeve et al.*, 2012; *Serreze and Stroeve*, 2015], the Antarctic is
witnessing a modest increase in total sea ice extent [*Parkinson and Cavalieri*, 2012]. Sea ice
around Antarctica reached another record high extent in September 2014, recording a
maximum extent of more than 20 million km$^2$ for the first time since the modern passive
microwave satellite data record began in October 1978. This follows previous record maxima in
2012 and 2013, resulting in an overall increase in Antarctic September sea ice extent of 1.1%
per decade since 1979. While the observed increase is statistically significant, Antarctic's sea
ice extent (SIE) is also highly variable from year to year and region to region [e.g. *Maksym et*
*al.*, 2012; *Parkinson et al.*, 2012; *Stammerjohn et al.*, 2012]. The temporal variability is
underscored by sea ice conditions in 2015 when the winter ice cover returned back to the 1981-
2010 long-term mean. Also, recent sea ice assessments from early satellite images from the
Nimbus program of the late 1960s indicate similarly high but variable SIE as observed over
2012-2014 [*Meier et al.,* 2013; *Gallaher et al.,* 2014]. Mapping of the September 1964 ice edge
indicates that ice extent likely exceeded both the 2012 and 2013 record monthly-average
maximums, at 19.7±0.3 million km$^2$. This was followed in August 1966 by an extent estimated at
15.9±0.3 million km$^2$, considerably smaller than the record low maximum extent of the modern
satellite record (set in 1986). The circumpolar average also hides contrasting regional variability,
with some regions showing either strong positive or negative trends with magnitudes equivalent
to those observed in the Arctic [*Stammerjohn et al.*, 2012]. In short, interannual and regional
variability in Antarctic sea ice is considerable, and while the current positive trend in circumpolar
averaged Antarctic sea ice extent is important, it is not unprecedented compared to
observations from the 1960s and it is not regionally distributed.
Several explanations have been put forward to explain the positive Antarctic sea ice trends.
Studies point to anomalous short-term wind patterns that both grow and spread out the ice,
related to the strength of the Amundsen Sea low pressure [e.g. *Turner et al.,* 2013; *Reid et al.*,
2015]. Other studies suggest melt water from the underside of floating ice surrounding the
continent has risen to the surface and contributed to a slight freshening of the surface ocean
[e.g. *Bintanja et al.*, 2012]. While these studies have helped to better understand how the ice,
ocean and atmosphere interact, 2012 to 2014 showed different regions and seasons
contributing to the net positive sea ice extent, which has made it difficult to establish clear links
and suggests that no one mechanism can explain the overall increase.
While the reasons for the increases in total extent remain poorly understood, it is likely that
these changes are not just impacting total sea ice extent but also the distribution of pack ice, the
marginal ice zone and polynyas. The marginal ice zone (MIZ) is a highly dynamic region of the



ice cover defined by the transition between the open ocean and the consolidated pack ice. In
the Antarctic, wave action penetrates hundreds of kilometers into the ice pack, resulting in small
rounded ice floes from wave-induced fracture [*Kohout et al.,* 2014]. Thus, in contrast to the
Arctic, ocean waves primarily define the dynamic MIZ region, though in the Arctic this may be
changing as the Arctic continues to experience longer and larger ice-free summers with
increased fetch on the later-timed ice edge advance [*Wang et al.,* 2015]. This in turn makes the
MIZ region particularly sensitive to both atmospheric and oceanic forcing, such that during
quiescent conditions, it may consist of a diffuse thin ice cover, with isolated thicker ice floes
distributed over a large (hundreds of kilometers) area. In contrast, during high on-ice wind and
wave events, the MIZ region contracts to a compact ice edge with rafted ice pressed together in
front of the solid ice pack. In general, ocean waves define the dynamic MIZ region, where ice
floes are relatively small due to wave-induced fracture. The smaller the ice floes, the more
mobile they are and large variability in ice conditions can be found in response to changing wind
and ocean conditions. Polynyas on the other hand are open water areas near the continental
margins that often remain open as a result of strong katabatic winds flowing down the Antarctic
plateau. The winds continually push the newly formed sea ice away from the continent, which
influences the outer ice edge as well, thus contributing to the overall increase in total ice extent
in specific regions around the Antarctic continent where katabatic winds are persistent.
Both polynyas and the MIZ are biologically important regions of the sea ice cover that have
important implications for the entire trophic web, from primary productivity [*Yun et al*.,
submitted], to top predator species, such as seabirds. Near the ice edge and in the MIZ, the
stable upper layer of the water column is optimal for phytoplankton production [e.g. *Park et al*.
1999]. This phytoplankton bloom is subsequently exploited by zooplankton, with effects that
cascade up to fish, seabirds and marine mammals. Similarly, within polynyas there is a narrow
opportunity for phytoplankton growth, the timing of which plays an important role in both
biogeochemical cycles [*Smith and Barber*, 2007] and biological production [*Arrigo and van*
*Dijken*, 2003; *Ainley et al*., 2010]. However, while studies have suggested that the timing of sea
ice retreat is synchronized with the timing of the phytoplankton bloom, other factors such as
wind forcing [*Chiswell*, 2011], thermal convection [*Ferrari,* 2014] and iron availability [*Boyd et al*,
2007, and references therein] play important roles as well.
In this study we use the long-term passive microwave sea ice concentration data record to
evaluate variability and trends in the marginal ice zone, the pack ice and polynyas from 1979 to
2014. A complication arises however as to which sea ice algorithm to use. There are at least a
dozen algorithms available, spanning different time-periods, which give sea ice concentrations
that are not necessarily consistent with each other [*see Ivanova et a*l., 2015; 2014 for more
information]. To complicate mattes, different studies have used different sea ice algorithms to
examine sea ice variability and attribution. For example, *Hobbs and Raphael* [2010] used the
Had1SST1 sea ice concentration data set [*Rayner et al.,* 2003], which is based on the NASA
Team algorithm [*Cavalieri et al*., 1999], whereas *Raphael and Hobbs* [2014] relied on the
Bootstrap algorithm [*Comiso and Nishio*, 2008]. To examine the influence in the choice of sea
ice algorithm on the results, we use both the Bootstrap and NASA Team sea ice algorithms.
Results are evaluated hemispheric-wide and also for different regions. We then discuss the
different implications resulting from the two different satellite estimates for biological impact



studies. We focus on the breeding success of snow petrels because seabirds have been
identified as useful indicators of the health and status of marine ecosystems [*Piatt and*
*Sydeman*, 2007].

## 2. Data and Methods

To map different ice types, the long-term passive microwave data record is used, which
spans several satellite missions, including the Scanning Multichannel Microwave Radiometer
(SMMR) on the Nimbus-7 satellite (October 1978 to August 1987), the Special Sensor
Microwave/Imager (SSM/I) sensors -F8 (July 1987 to December 1991), -F11 (December 1991
to September 1995), -F13 (May 1995 to December 2007) and the Special Sensor Microwave
Imager/Sounder (SSMIS) sensor –F17 (January 2007- to present), both on the Defense
Meteorological Satellite Program's (DMSP) satellites. Derived sea ice concentrations (SICs)
from both the Bootstrap (BT) [*Comiso and Nishio*, 2008] and the NASA Team (NT) sea ice
algorithms [*Gloersen et al.,* 1992; *Cavalieri et al.,* 1999] are available from the National Snow
and Ice Data Center (NSIDC) and provide daily fields from October 1978 to present, gridded to
a 25 km polar stereographic grid. While a large variety of sea ice concentration algorithms are
available, the lack of good validation has made it difficult to determine which algorithm provides
the most accurate results during all times of the year and for all regions. Using two algorithms
provides a consistency check on variability and trends.

Using these SIC fields, we define six binary categories of sea ice based on different SIC
thresholds [**Table 1**]. Because the marginal ice zone is highly dynamic in time and space, it is
difficult to precisely define this region of the ice cover. *Wadhams* [1986] defined the MIZ as that
part of the ice cover close enough to the open ocean boundary to be impacted by its presence,
e.g. by waves. Thus the MIZ is typically defined as the part of the sea ice that is close enough to
the open ocean to be heavily influenced by waves, and it extends from the open ocean to the
dense pack ice. In this study, we define the MIZ as extending from the outer sea ice/open ocean
boundary (defined by SIC $\geq$ 0.15 ice fraction) to the boundary of the consolidated pack ice
(defined by SIC = 0.80). This definition was previously used by *Strong and Rigor* [2013] to
assess MIZ changes in the Arctic and matches the upper sea ice concentration limit used by the
National Ice Center in mapping the Arctic MIZ. The consolidated ice pack is then defined as the
area south of the MIZ with ice fractions between 0.80 $\leq$ SIC $\leq$ 1.0. Coastal polynyas are defined
as regions near the coast that have SIC < 0.80.

To automate the detection of different ice types, radial transects from 50 to 90S are
individually selected to construct one-dimensional profiles [**Figure 1**]. The algorithm first steps
from the outer edge until the 0.15 SIC is detected, providing the latitude of the outer MIZ edge.
Next, the algorithm steps from the outer MIZ edge until either the 0.80 SIC is encountered, or
the continent is reached. Data points along the transect between these SIC thresholds are
flagged as the MIZ. In this way, the MIZ includes an outer band of low sea ice concentrations
that surrounds a band of inner consolidated pack ice, *but* sometimes the MIZ also extends all
the way to the Antarctic coastline (as sometimes observed in summer). South of the MIZ, the
consolidated ice pack (0.80 $\leq$ SIC $\leq$ 1.0) is encountered; however, low sea ice concentrations
can appear near the coast inside the pack ice region as well. These are areas of potential





coastal polynyas. While it is difficult to measure the fine scale location of a polynya at 25km
spatial resolution, the lower sea ice concentrations provide an indication of some open water
near the coast, which for sea birds provides a source of open water for foraging. Using our
method of radial transects, the algorithm then steps from the coast northward and flags pixels
with < 0.80 SIC until a 0.80 SIC pixel appears and defines that region as a potential coastal
polynya. Within the consolidated pack ice (and away from the coast), it is also possible to
encounter instances where 0.15 < SIC < 0.80 or SIC < 0.15. These are flagged as open pack
ice and open water areas within the consolidated pack ice, respectively. Finally, an ocean mask
derived from climatology and distributed by NSIDC was applied to remove spurious ice
concentrations at the ice edge as a result of weather effects.
**Figure 2** shows sample images of the classification scheme as applied to the NASA Team
and Bootstrap algorithms on days 70 and 273, respectively, in 2013. During the fall and winter
months when the ice cover is expanding there is a well-established consolidated pack ice
region, surrounded by the outer MIZ. Coastal polynyas are also found surrounding the continent
in both algorithms. As will be discussed in more detail in the results section, the BT algorithm
tends to show a larger consolidated ice pack than NT, particularly during the timing of maximum
extent. During the melt season there is mixing of low and high ice concentrations, leading to
mixtures of different categories, which is still seen to some extent in the March images.
However, during March areas of polynyas (green), open water (pink) and open pack ice
(orange) appear to extend from the coastline in some areas (e.g. southern Weddell and Ross
seas). While any pixel with SIC < 0.8 adjacent to the coastal boundary is flagged as potential
polynya when stepping northwards, if a pixel is already flagged as MIZ or consolidated pack ice
when stepping southwards, it remains flagged as MIZ or pack ice. After that analysis, a check
for pixels with SICs less than 0.8 is done to flag for broken ice or open water. Thus, during these
months (e.g. December to February or March), the physical interpretation of the different ice
classes may be less useful.
Using the binary classification scheme, gridded fields and regional averages are computed.
We show results for the entire Antarctic sea ice cover, as well as for six different regions as
defined previously by *Parkinson and Cavalieri* [2012]. These regions are shown in **Figure 3** for
reference. Climatological mean daily and monthly time-series spanning 1981 to 2010 are
computed for each region and for each ice classification together with the +/- one standard
deviation ($1\sigma$). Monthly trends over the entire time-series are computed by first averaging the
daily fields into monthly values and then using a standard linear least squares, with statistical
significance evaluated at the $90^{th}$, $95^{th}$ and $99^{th}$ percentiles using a student t-test.

## 3. Results

### 3.1 Seasonal Cycle

#### 3.1.1 Circumpolar Extent

We begin with an assessment of the consistency of the outer ice edge between both sea ice
algorithms [**Figure 4**]. As a result of the large emissivity difference between open water and sea
ice, estimates of the outer ice edge location has high consistency between the two algorithms





despite having large differences in sea ice concentration [e.g. *Ivanova et al.*, 2014; 2015]. This
therefore results in similar total sea ice extents between both algorithms during all calendar
months, and similar long-term trends. This is where the similarities end however.
**Figure 5** summarizes the climatological mean seasonal cycle in the extent of the different
ice categories listed in Table 1 for both sea ice algorithms, averaged for the total hemispheric-
wide Antarctic sea ice cover. The one standard deviation is given by the colored shading. The
first notable result is that the BT algorithm has a larger consolidated ice pack than the NT
algorithm, which comes at the expense of a smaller MIZ. Averaged over the entire year, the
NASA Team MIZ area is twice as large as that in the Bootstrap algorithm [see also **Table 2**].
The BT algorithm additionally has a smaller spatial extent of potential coastal polynyas and little
to no broken ice or open water within the consolidated pack ice. Another important result is that
the BT algorithm exhibits less interannual variability in the different ice types, as illustrated by
the smaller standard deviations from the long-term mean (e.g. the shading). Thus, while the
total extents are not dissimilar between the algorithms, how that ice is distributed among the
different ice categories differs quite substantially as well as their year-to-year variability.
The timing of the ice edge advance and retreat are generally similar in both algorithms,
reflecting the fact that both algorithms do well in distinguishing open water from sea ice. In
regards to the consolidated pack ice, it advances in March, with the BT algorithm showing a
distinct peak in September, reaching a maximum extent of 14.89 $10^6$ km$^2$. The NT algorithm
shows a somewhat broader peak, extending from July to October, with the peak extent also
reached in September. In September the NT pack ice extent is a little more than twice the
spatial extent of the MIZ; 11.31 $10^6$ km$^2$ vs. 5.41 $10^6$ km$^2$ [Table 2]. BT on the other hand has a
much smaller fraction (41% less) of ice classified as MIZ (3.19 $10^6$ km$^2$). In both algorithms the
MIZ also begins to expand in March, and continues to expand until November or December,
after which it rapidly declines. However, in the NT algorithm, an initial peak in MIZ coverage is
also reached around September, coinciding with the peak in the consolidated pack ice extent
and stays nearly constant until the end of November. The further increase in the MIZ coverage
after the consolidated ice pack begins to retreat implies that as the pack ice begins to retreat, it
does so in part by first converting to MIZ over a wider area. This is consistent with the idea that
in spring, the pack ice on average undergoes divergence first (in relation to the circumpolar
trough being poleward and south of the ice edge, as reflected by the Semi-Annual Oscillation,
SAO, of the trough). This in turn facilitates increased solar heating of open water areas, which in
turn facilitates increased melt back, thus creating, eventually, a more rapid ice edge retreat (in
Nov-Dec) as compared to the slow ice edge advance in autumn [see *Watkins and Simmonds*,
1999].
Open pack ice is negligible in the Bootstrap algorithm except for a slight peak in
November/December. With the NASA Team algorithm however there is a clear increase in open
pack ice during the ice expansion phase, which continues to increase further as the pack ice
begins to retreat, also peaking in November. Open pack ice in September contributes another
1.28 $10^6$ km$^2$ to the total Antarctic sea ice extent in the NT algorithm, compared to only 0.36 $10^6$
km$^2$ in the BT algorithm. As with the open pack ice, the fraction of potential coastal polynyas
also increases during the ice expansion phase, and then continues to increase as the sea ice





retreats, peaking around November in the NT algorithm, with a total area of $1.02 \ 10^6$ km$^2$, and in
December in BT ($0.81 \ 10^6$ km$^2$). Inner open water within the pack is generally only found
between November and March in both algorithms as the total ice cover retreats and reaches its
seasonal minimum.
**3.2.2 Regional Analysis**
Analysis of the Antarctic-wide sea ice cover however is of limited value given that the sea
ice variability and trends are spatially heterogeneous [*Makysm et al.*, 2012]. For example, while
the ice cover is increasing in the Ross Sea, it has at the same time decreased in the
Bellingshausen/ Amundsen Sea region. Thus, we may anticipate significant regional variability
in the amount, seasonal cycle and trends of the different ice classes (trends discussed in
section 3.3). The Ross Sea for example [**Figure 6, top**] consists of a large fraction of
consolidated ice throughout most of the year (April through November) in both algorithms, with
considerably less MIZ. In the Bellingshausen/Amundsen Sea on the other hand [**Figure 6, 2$^{nd}$**
**row**], the NT algorithm has a MIZ extent that exceeds that of the consolidated pack ice until
May, after which the spread (+/- 1$\sigma$) in MIZ and consolidated pack ice overlaps. The reverse is
true in the BT algorithm, which consistently indicates a more consolidated ice pack, with only
$0.51 \ 10^6$ km$^2$ flagged as MIZ during the maximum extent in September, compared to $0.84 \ 10^6$
km$^2$ in the NT algorithm. On an annual basis, the NT algorithm shows about equal proportion of
MIZ and consolidated pack ice in the Bellingshausen/Amundsen Sea whereas, the BT algorithm
indicates a little more than a third of the total ice cover is MIZ. In the Ross Sea there is also a
very broad peak in the maximum extent of the consolidated pack ice, stretching between July
and October in the NT algorithm, and a peak in MIZ extent in late August/early September with
a secondary peak in December as the pack ice continues to retreat. The BT algorithm shows a
similar broad peak in the pack ice extent, but with less interannual variability, and a nearly
constant fraction of MIZ throughout the advance and retreat of the pack ice. Annually the NT
algorithm shows about 56% more MIZ in the Ross Sea than the BT algorithm. Note that in both
algorithms, the pack ice retreats rapidly after the maximum extent is reached.

In the Weddell Sea, the pack ice extent advances in March in both algorithms and peaks in
August in the NT algorithm, September in BT. The MIZ also begins its expansion in March and
continues to increase until September in NT, and then again until December (both algorithms)
as the pack ice quickly retreats [**Figure 6 (middle)**]. In this region, the sea ice expands
northwards until it reaches a region with strong winds and currents. The open pack ice north of
the pack ice continues to expand either by further freezing or breaking of the pack ice by the
winds and currents. Overall, the Weddell Sea has the largest spatial extent in the MIZ in both
algorithms, as well as the largest distribution of pack ice. In the NT algorithm however, the MIZ
extent within the Weddell Sea is again considerably larger than in the BT algorithm. For
example, in September the NASA Team algorithm gives a climatological mean MIZ extent of
$1.61 \ 10^6$ km$^2$, twice as large as that in the Bootstrap algorithm ($0.83 \ 10^6$ km$^2$).

Finally, in the Indian and Pacific Ocean sectors [**Figure 6, 4$^{th}$ row**] the MIZ extent increases
from March until November in both algorithms, retreating about a month after the peak extent in
the pack ice is reached. However, in the Pacific Ocean sector [**Figure 6, bottom**], the MIZ
comprises a larger percentage of the overall ice cover, being nearly equal in spatial extent in the





NASA Team algorithm, and even exceeding that of the pack ice in September (0.93 (MIZ) vs.
0.76 $10^6$ km$^2$ (pack ice)). This results in an annual mean extent of MIZ that exceeds that of the
consolidated pack ice. This is the only region of Antarctica where this occurs. In the BT
algorithm, the reverse is true, with again a larger annual extent of pack ice than MIZ.
While the above discussion focused on regional differences in the MIZ and the consolidated
pack ice, the spatial extent and timing of coastal polynyas also varies between the algorithms.
For example, in the Bellingshausen/Amundsen sea region, the maximum polynya area occurs in
July in NT (0.17 $10^6$ km$^2$) and in December in the BT algorithm (0.11 $10^6$ km$^2$). Thus, while the
overall maximum spatial extent in polynya area is not all that different in the two algorithms, the
timing of when the maximum is reached differs by 5 months. This is also the case in the Pacific
Ocean where the NASA Team algorithm reaches its largest spatial extent in polynya area in
August (0.14 $10^6$ km$^2$) whereas the Bootstrap shows the maximum polynya area occurring in
November (0.11 $10^6$ km$^2$). In other regions, such as the Indian Ocean, the Ross Sea and the
Weddell Sea, the timing of the maximum polynya area occurs similarly in both algorithms,
during November for the Indian Ocean and December in the Ross and Weddell Seas. The Ross
and Weddell seas have the largest climatological polynya areas, 0.32 (NT)/0.26 (BT) $10^6$ km$^2$
and 0.33 (NT)/0.30 (BT) $10^6$ km$^2$, respectively.

## 3.2 Trends

### 3.2.1 Spatial Expansion/Contraction during September

As mentioned earlier, estimates of the outer ice edge location are similar between both
algorithms. This is also true in terms of the locations where the outer edge is expanding or
contracting. A way to illustrate this is shown in **Figure 7 (top)**, which shows a spatial map of the
trend in the outer edge of the entire ice pack (defined as the 15% SIC contour, equivalent to the
total sea ice extent) for both algorithms during the month of September, the month at which the
ice pack generally reaches its maximum extent. Locations of northward expansion (red areas)
and contraction (blue areas) are remarkably consistent between algorithms as well as the
spatial extent of the expansion and contraction. In both algorithms the ice edge shows trends
towards expansion within the Ross Sea, the Amundsen Sea and the Pacific and Indian Ocean
sectors, except for the Davis Sea, where there is a trend towards contraction of the outer ice
edge. The Bellingshausen and Weddell seas also show trends towards contraction of the outer
ice edge.
While there is general consistency between the algorithms in both the location and changes
of the outer ice edge over time, there are differences as to how the MIZ and pack ice widths are
changing [**Figure 7, middle and bottom**]. In the BT algorithm, the MIZ width is a relatively
constant ring around the edge of the consolidated pack ice, with little change over time. Thus, in
the BT algorithm, the spatial pattern of expansion/contraction of the total ice cover in September
is largely a result of the changes happening in the pack ice [Figure 7, bottom]. The NT algorithm
on the other hand shows more pronounced changes in the MIZ, such that both the MIZ and the
pack ice contribute to the observed spatial patterns and changes in the total ice cover. However,
expansion/contraction of the MIZ and pack ice in the NT algorithm sometimes counter act each
other. For example the contraction of the total ice edge the Bellingshausen Sea is a result of
contraction of the consolidated ice pack while the MIZ width is generally increasing as a result of



the MIZ moving further towards the continent. This is also true in the Weddell Sea and the
Indian Ocean.
Somewhat surprisingly, the spatial pattern of expansion/contraction of the MIZ is broadly
similar between both algorithms, despite overall smaller changes in the BT algorithm. This
highlights the fact that the spatial trends in SIC are similar to the spatial trends in SIE as well as
to the timing of advance/retreat/duration, so that the spatial trends in the MIZ and pack ice will
show the same overall pattern because they rely on SIC. This also highlights the fact that the
spatial pattern persists throughout the regional ice covered area, i.e. from the edge to the
coastal area, which may imply that climate-related regional wind-driven changes at the ice edge
are felt all the way to the coast. Alternatively it may imply that the ocean is also responding to
the same climate-related wind changes, thus communicating the change all the way to the
coast.

### 3.2.2 Circumpolar and Regional Daily Trends

**Figure 8** summarizes daily circumpolar Antarctic trends in the pack ice, MIZ and polynyas
for both algorithms, with monthly mean trends listed in **Table 3**. Both algorithms are broadly
similar during the ice expansion phase, indicating positive trends in the consolidated ice pack
and mostly negative trends in the MIZ until the pack ice reaches its peak extent. Thus, during
these months, the positive trends in total SIE are a result of expansion of the consolidated pack
ice. However, during retreat of the pack ice, trends in the MIZ switch to positive in the NASA
Team algorithm while remaining mostly negative in the Bootstrap algorithm. At the same time,
daily trends in the pack ice become noisy in the NT algorithm, alternating between positive and
negative trends while trends remain positive in the BT algorithm. Table 3 indicates that the
positive trends in the consolidated pack during the ice expansion/retreat phase (March through
November) are statistically significant ($p<0.01$) for the BT algorithm, and from March to July in
the NT algorithm ($p<0.05$). Trends in the MIZ are not statistically significant, except during
September and October at the 90% confidence level in the NT algorithm. Trends in the pack ice
are larger in the BT algorithm, particularly in August through November, in part reflecting a
shrinking MIZ whereas the NT algorithm shows positive trends in the MIZ during those months.
Trends in possible polynyas near the continent are negative throughout most of the year in both
algorithms, except for December and January. However, none of the polynya trends are
statistically significant.
Regionally, there are larger differences between the two algorithms, in particular with
regards to the MIZ as already alluded to in Figure 7. To highlight the regional differences Figure
9 shows daily trends as a function longitude (x-axis) and month (y-axis) for the pack ice (top),
the MIZ (middle) and coastal polynyas (bottom). Monthly trends averaged for each of the 5
sectors are also listed in Table 3. Focusing first on the pack ice trends, we find the spatial
patterns of positive and negative trends are generally consistent between both algorithms,
though the magnitudes of the trends tend to be larger in the Bootstrap algorithm, which in turn
impacts the statistical significance of the trends (see also Table 3). For example, in the Ross
Sea, the largest regional positive trends in total SIE are found at a rate of 119,000 km$^2$ per
decade [e.g. *Turner et al.*, 2015], accounting for about 60% of the circumpolar ice extent
increase. In the BT algorithm this is entirely a result of large positive trends in the pack ice from




March to November (p<0.01). While the Ross Sea sector trends from the NT algorithm are
spatially consistent with the pack ice trends shown in the BT algorithm, trends are only
statistically from April to June (p<0.05). Instead, statistically significant positive trends in the MIZ
dominate August to October in the NT algorithm, which is also the season with the largest
overall trends in the SIE in this region (e.g. Spring). This would suggest perhaps different
interpretation of processes impacting the overall ice expansion in the Ross Sea depending on
which algorithm is used. Several studies have suggested a link between sea ice anomalies in
the Ross Sea and the wind-field associated with the Amundsen Sea Low (ASL) [e.g. *Fogt et al*.,
2012; *Hosking et al*., 2013; *Turner et al*., 2012]. The strengthened southerly winds over the
Ross Sea cause a more compacted and growing consolidated ice cover in the BT algorithm at
the expense of a shrinking MIZ, whereas in the NT algorithm the area of the MIZ is increasing
more than the pack ice during autumn, which may additionally suggest an oceanic influence.
While this is true as averaged over the entire Ross Sea sector, Figure 9 highlights that the area-
averaged trends hide spatial variability, with positive trends in the MIZ in the eastern part of the
Ross Sea and negative trends in the western part.
While the magnitude of pack ice trends are generally larger in the Bootstrap algorithm, there
are some exceptions. For example, in the Weddell Sea, the NT algorithm exhibits larger
negative trends in the pack ice between June and November whereas the BT algorithm shows
mixed positive and negative trends of smaller magnitude. This is also true with regards to MIZ
trends during these months. However, none of the trends are statistically significant. In the
Weddell Sea, expansion of the overall ice cover is only statistically significant during the autumn
months (MAM) [e.g. *Turner et al*., 2015]. During this time-period, both algorithms agree on
statistically significant positive trends in the pack ice area, that extend through May for the NT
algorithm (p<0.05) and through June for the BT algorithm (p<0.05). Statistically significant
trends are also seen during March in the MIZ and polynya area (p<0.05), with larger trends in
the NT algorithm (p<0.01). Thus, overall expansion of sea ice in the Weddell during autumn is in
part driven by expansion of the MIZ early in the season, after which it is controlled by further
expansion of the consolidated pack.
In contrast, the Bellingshausen/Amundsen Sea is a region undergoing declines in the overall
ice cover [e.g. *Parkinson and Cavalieri*, 2012; *Stammerjohn et al*., 2012]. Separating out trends
for both the pack ice and the MIZ reveals negative trends in the consolidated pack ice during the
start of ice expansion in March and April and also during initial retreat (September and October)
in both algorithms, though none of the trends are statistically significant [Table 3]. This is the
only region where the BT algorithm does not show statistically significant trends in the pack ice.
Negative trends are also found in the MIZ during the initial ice advance phase in both algorithms
though again none of them are statistically significant. Interestingly, during June and July, the
NT algorithm shows large positive trends in the pack ice (p<0.01) at the expense of negative
trends in the MIZ, though the MIZ trends are not statistically significant and are smaller than the
positive trends in the pack ice. While the MIZ trends are not statistically significant, these results
are consistent with the observation that the SIE trends in the Bellingshausen/Amundsen Sea
are largely wind-driven, so it would be expected that the wind-driven compaction would lead to
decreased MIZ and increased pack ice. Finally, both algorithms indicate statistically significant





positive trends in coastal polynyas during November for this region (with larger trends in the NT
algorithm, +1,000 km$^2$a$^{-1}$ (p<0.05) and +600 km$^2$a$^{-1}$ (p<0.10), respectively).
Finally, in the Pacific and Indian Oceans we again see spatial consistency in pack ice and
MIZ trends for both algorithms, with generally larger (smaller) pack ice (MIZ) trends for the BT
algorithm, though trends are closer in magnitude in the Pacific sector from March to July. The
BT algorithm indicates statistically significant trends in the pack ice from March to November in
both sectors (p<0.05), while trends in overall SIE are only statistically significant in the Indian
Ocean during MAM and JJA. The inconsistency in statistical significance between total SIE and
pack ice trends is likely a result of corresponding negative trends in the MIZ, particularly in the
Pacific sector, though the negative BT MIZ trends are not statistically significant. The NT
algorithm mostly has statistically significant trends in the pack ice during the initial expansion
phase only (p<0.05). In the Indian Ocean, there are also significant positive trends in MIZ during
March (p<0.05) and April (p<0.10) and also June and July (p<0.10) that would contribute
towards overall positive SIE trends. Both algorithms suggest an increase in polynya area from
March to May (p<0.05) in the Pacific sector, and the NT for the Indian sector in March (p<0.05).
In summary, while the magnitude of trends differs between both algorithms, there is general
spatial consistency in the patterns of positive and negative trends in the consolidated pack ice
and the MIZ. Results suggest that positive trends in total SIE are generally a result of
statistically significant positive trends in the consolidated pack ice in the BT algorithm in all
sectors of the Antarctic, except for the Bellingshausen/Amundsen Sea sector and the Weddell
Sea during ice retreat. The NT algorithm on the other hand suggests more instances of
statistically significant positive trends in the MIZ, though this is highly regionally dependent.
Finally, the largest expansion of polynya area is found in the Bellingshausen/Amundsen Sea
during November, whereas small increases in polynya area are found in both the Indian and
Pacific sector during the ice expansion phase. Outside of these regions/months, no significant
changes in coastal polynya area are observed.

### 3.2.3 Seasonal Trends in MIZ and Pack Ice Width

Finally we compute the overall width of the MIZ and pack ice following *Strong and Rigor*
[2013] and produce seasonal means. Time-series of seasonal means of the circumpolar MIZ
width and pack ice width are shown in **Figure 10** for all seasons except summer when the
results are noisy. As we may expect following the previous results, the NASA Team algorithm
consistently shows greater MIZ width and smaller pack ice width than the Bootstrap algorithm.
During autumn (MAM) however, the differences between the algorithms are reduced, both for
the MIZ and pack ice widths. In addition, during this season, trends in the MIZ and pack ice are
largely consistent, with no trend in the MIZ and increases in the pack ice on the order of 21.2 km
dec$^{-1}$ and 20.0 km dec$^{-1}$ (p<0.01) for the BT and NT algorithms, respectively.
During winter (JJA) and spring (SON) however, the NT and BT algorithms exhibit opposing
trends in the MIZ with the NT algorithm indicating an increase and the BT a decrease. The
largest positive trend in the MIZ width occurs during spring at a rate of +10.3 km dec$^{-1}$ (p<0.01)
in the NT algorithm, indicating a 6% widening over the satellite record. This widening is a result
of the MIZ moving slightly equatorward rather than expanding southwards. However, despite a





statistically significant trend, there remains substantial interannual variability in the SON MIZ
width, with the maximum width recorded in 2003 (310 km) and the minimum in 1985 (217 km),
with a mean SON MIZ width of 248 km. The trend during winter is considerably smaller at +2.7
km dec$^{-1}$, as a result of expansion equatorward and southwards, yet it is not statistically
significant.
For the pack ice, both algorithms show statistically significant positive trends towards
increased width of the pack ice, which are also nearly identical during winter at +18.7 and +18.1
km dec$^{-1}$ (p<0.01) for the BT and NT algorithms, respectively. This represents a widening of the
pack ice of approximately 11% from 1979 to 2014 during winter. As one may expect, differences
in the pack ice width between the algorithms are largely found in spring as a result of the MIZ
expanding in the NT algorithm. During SON the trends in the width of the pack ice are slightly
smaller than during winter, with trends of +16.7 (BT, p<0.01) and +10.0 (NT, p<0.05) km dec$^{-1}$.
Interestingly, the interannual variability in the pack ice is similar between both data sets,
showing correlations between the two algorithms of 0.92 (JJA), 0.77 (SON) and 0.96 (MAM).
For the MIZ, interannual variability is generally about twice as large in the NASA Team
algorithm and the two data sets are not highly correlated except for autumn, with correlations of
0.39 (JJA), 0.43 (SON), and 0.67 (MAM).

## 4. Implications for a Seabird

Here we use data on the MIZ and the consolidated ice pack from both algorithms to
understand the role of sea ice habitat on breeding success of a seabird, the snow petrel
*Pagodroma nivea*. As mentioned in the introduction, the MIZ is a biologically important region
because it is an area of high productivity and provides access to food resources needed by
seabirds [*Ainley et al*., 1992]. During winter, productivity is reduced at the surface in open water,
while it is concentrated within the ice habitat, especially within the ice floes [*Ainley et al.,* 1986].
This patchy distribution of food availability within the MIZ and pack ice provides feeding
opportunities for seabirds such as the snow petrel. Observations suggest that the snow petrel
forages more successfully in areas close to the ice edge and within the MIZ than in consolidated
ice conditions [*Ainley et al.,* 1984, 1992].
Breeding success of snow petrels depends on sufficient body condition of the females,
which in part reflects favorable environmental and foraging conditions prior to the breeding
season. Indeed, female snow petrels in poor early body condition are not able to build up the
necessary body reserves for successful breeding [*Barbraud and Chastel*, 1999]. Breeding
success was found to be higher during years with extensive sea ice cover during the preceding
winter [*Barbraud and Weimerskirch*, 2001]. This is in part because winters with extensive sea
ice are associated with higher krill abundance the following summer [*Flores et al*., 2012; *Loeb et*
*al.,* 1993; *Atkinson et al.,* 2004], thereby increasing the resource availability during the breeding
season. However, extensive winter sea ice may protect the under ice community from predation
and thus reduce food availability, in turn affecting breeding success [*Olivier et al.,* 2005]. By
distinguishing between the areas of MIZ and pack ice, we can expect a better understanding of
the role of sea ice on food availability and hence breeding success of snow petrels.





In the following, we expect that an extensive pack ice may reduce breeding success by
protecting the under ice community from predation, while an extensive MIZ may increase
breeding success by providing easier access to foraging. With the classifications as defined by
both algorithms we calculated the MIZ and pack ice area in a wide rectangular sector defined by
the migration route of the snow petrel [*Delord et al.,* 2013] from April to September [see **Table 4**
for latitude and longitude limits]. We then averaged the MIZ and pack ice extents over the entire
winter from April to September. We next employed a logistic regression approach to study the
effects of MIZ and pack ice area within this sector and evaluate the impacts on breeding
success the following summer. The response variable was the number of chicks $C_t$ in a breeding
season $t$, from 1979 to 2014 collected at Terre Adélie, Dumont D'Urville [*Barbraud and*
*Weimerskirch*, 2001, *Jenouvrier et al*., 2005].
Effects of MIZ and pack ice area were analyzed using Generalized Linear Models (GLM)
with logit-link functions and binomial errors fitted in R using the package glm. We selected the
best model according to the information criteria AIC, the chosen model being the one that
minimizes the AIC, and the ability of two models to describe the data was assumed to be "not
different" if the difference in their AIC was < 2 [*Burnham and Anderson*, 2002]. While non-linear
models may be more appropriate as ecological system relationships are likely more complex
than linear relationships, without *a priori* knowledge of the mechanisms that could lead to such
non-linear relationships, it is extremely difficult to interpret the results.
**Table 5** summarizes model selection. The model with the lowest AIC suggests an effect of
the consolidated pack ice area on breeding success as derived from the Bootstrap algorithm.
The MIZ and pack ice areas calculated from the NT algorithm are not supported (AIC
difference>2).  As expected we found that the effect of consolidated pack ice on breeding
success was negative [**Figure 11**]. In other words, more extensive consolidated pack ice during
winter tends to reduce breeding success the following summer by limiting foraging opportunities.
The effect of the MIZ however was uncertain, contrary to what one may expect given the
increased opportunities for foraging within the MIZ. However, if we had only used ice
classifications based on the NASA Team algorithm, the model with the lowest AIC would have
suggested an importance of the MIZ. We would have then concluded a negative effect of the
MIZ on the breeding success of snow petrels, contrary to what one may expect given that the
MIZ is the main feeding habitat of the species. By using both algorithms, we instead conclude
that the breeding success of snow petrels is negatively affected by the pack ice area as
calculated with the Bootstrap algorithm.

# 5. Discussion

The positive trends in Antarctic sea ice extent are currently poorly understood and are at
odds with climate model forecasts that suggest the sea ice should be declining in response to
increasing greenhouse gases and stratospheric ozone depletion [e.g. *Turner et al*., 2013; *Bitz*
*and Polvani*, 2012; *Sigmond and Fyfe*, 2010]. However, several modeling studies, such as those
used in the phase 5 Coupled Model Intercomparison Project (CMIP5), have suggested that the
sea ice increase over the last 36 years remains within the range of intrinsic of internal variability
[e.g. *Bitz and Polvani*, 2012; *Turner et al*., 2013; *Mahlstein et al*., 2013; *Polvani and Smith,*





2013; *Swart and Fyfe*, 2013]. Earlier satellite from the 1960s and 1970s and from ship
observations suggest periods of high and low sea ice extent, and thus high natural variability
[*Meier et al.*, 2013; *Gallaher et al*., 2014]. Further evidence comes from ice core climate records,
which suggest that the climate variability observed in the Antarctic during the last 50 years
remains within the range of natural variability seen over the last several hundred to thousands of
years [*Thomas et al*., 2013; *Steig et al*., 2013]. Thus, we may require much longer records to
properly assess Antarctic sea ice trends in contrast to the Arctic, where negative trends are
outside the range of natural variability and are consistent with those simulated from climate
models.
While many assessments of how Antarctic sea ice trends and variability compare with
climate models have focused on the net circumpolar sea ice extent, it is the regional variability
that becomes more important. For example, *Hobbs et al*. [2015] argue that when viewing trends
on a regional basis, the observed summer and autumn trends fall outside of the range of natural
variability as simulated by present-day climate models, with the signal dominated by opposing
trends in the Ross Sea and the Bellingshausen/Amundsen seas. These results have questioned
the ability of climate models to correctly simulate processes at the regional level and within the
southern ocean-atmosphere-sea ice coupled system.
The net take-away point from these studies is that the net circumpolar changes in sea ice
extent do not enhance our understanding of how the Antarctic sea ice is changing. Instead our
focus should be on what drives regional and seasonal sea ice changes, including feedbacks
and competing mechanisms. This study aims to better understand regional and total changes in
Antarctic sea ice by focusing not only on the total ice area, but also on how the consolidated
pack ice, the marginal ice zone and coastal polynyas are changing. Differences in climatologies
and trends of the different ice classes may suggest different processes are likely contributing to
their seasonal and interannual variability. In addition, the different contributions of ice types
towards the overall expansion of the Antarctic sea ice cover between algorithms may in turn
influence attribution of the observed increase in SIE. For example, within the highly dynamic
MIZ region, intense atmosphere-ice-ocean interactions take place [e.g. *Lubin and Massom*,
2006] and thus an expanding or shrinking MIZ may help to shed light on the relative importance
of atmospheric or oceanic processes impacting the observed trends in total SIE. Another issue
is whether or not new ice is forming along the outer edge of the pack ice or if it is all being
dynamically transported from the interior.
However, a complication exists, what sea ice algorithm should be used for such
assessments? In this study we focused on using passive microwave satellite data for defining
the different ice types as it is the longest time-series available and is not limited by polar
darkness or clouds. However, results may be highly dependent on which sea ice algorithm is
used to look at the variability in these ice classes, which will also be important in assessing
processes contributing to these changes as well as implications of these changes to the polar
marine ecosystem. In this study, the positive trends in circumpolar sea ice extent over the
satellite data record are primarily driven by statistically significant trends (p<0.05) in expansion
of the consolidated pack ice in both sea ice algorithms. However, an exception occurs in the
NASA Team sea ice algorithm after the ice pack reaches its seasonal maximum extent when





the positive trends in the pack ice are no longer as large, nor statistically significant. Instead,
positive trends in the MIZ dominate during September and October (p<0.10). This is in stark
contrast to the Bootstrap algorithm, which shows a declining MIZ area from March through
November.
The algorithms also give different proportions of how much the total ice cover consists of
consolidated ice, MIZ or polynya area. In some regions, such as the Pacific Ocean sector, the
NT algorithm suggests the MIZ is the dominant ice type whereas in the BT algorithm, the pack
ice is dominant, which is true for all sectors analyzed in the Bootstrap algorithm. Considering the
circumpolar ice cover, the MIZ in the NASA Team algorithm is on average twice as large as in
the Bootstrap algorithm. In the Arctic, *Strong and Rigor* [2013] found the NASA Team algorithm
gave about three times wider MIZ than the Bootstrap algorithm. In this case, the Bootstrap
results agreed more with MIZ widths obtained from the National Ice Center (NIC).
Differences between the algorithms are not entirely surprising as the two algorithms use
different channel combinations with different sensitivities to changes in physical temperature. In
addition, the NT uses previously defined tie points for passive microwave radiances over known
ice-free ocean, and ice types, defined as type A and B in the Antarctic, as the radiometric
signature between first-year and multiyear ice in the Antarctic is lost. The ice is assumed to be
snow-covered when selecting the tie points, which can result in an underestimation of sea ice
concentration if the ice is not snow covered. In addition, seasonal variations in sea ice emissivity
can be very large, leading to seasonal biases in either algorithm. The advantage of the
Bootstrap algorithm is that the ice concentration can be derived without an *a priori* assumption
about ice type, though consolidated ice data points are sometimes difficult to distinguish from
mixtures of ice and open ocean due to the presence of snow cover, flooding or roughness
effects.
While one may expect the Bootstrap algorithm to provide more accurate results than the
NASA Team algorithm, near the coast the BT algorithm has been shown to have difficulties
when temperatures are very cold. Because the NT algorithm uses brightness temperature ratios
it is largely temperature independent. However, during summer or for warmer temperatures, the
NT algorithm may indeed be biased towards lower sea ice concentrations whereas the BT
algorithm may be biased towards higher ice concentrations [e.g. *Comiso et al.,* 1997]. In the
Arctic, the MIZ is not only driven by wave mechanics and flow breaking (dynamic origin), but
also by melt pond processes in summer (thermodynamic origin) [*Arnsten et al.*, 2015]. Thus,
larger sensitivity of the NT algorithm to melt processes may be one reason for the large
discrepancy observed in the Arctic. Interestingly, the BT algorithm shows less interannual
variability in the ice types compared to NT (as shown by the smaller standard deviations). This
would in turn influence assessments of atmospheric or oceanic conditions driving observed
changes in the ice cover. What is clear is that more validation is needed to assess the accuracy
of these data products, especially for discriminating the consolidated pack ice from the MIZ.
Errors likely are larger in the MIZ because of the coarse spatial resolution of the satellite
sensors. Another concern is that mapping of the consolidated ice pack does not always mean a
compact ice cover. The algorithms may indicate 100% sea ice concentration (e.g. a





consolidated pack ice), when in reality the ice consists of mostly brash ice and small ice floes
more representative of the MIZ. Future work will focus on validation with visible imagery.

## Conclusions

Total Antarctic sea ice cover is expanding in response to atmospheric and oceanic variability
that remains to be fully understood. One may expect that these increases would also be
manifested in either equatorward progression of the MIZ or the consolidated pack ice or both. In
this study we identified several different ice categories using two different sets of passive
microwave sea ice concentration data sets. The algorithms are in agreement as to the location
of the northern edge of the total sea ice cover, but differ in regards to how much of the ice cover
consists of the marginal ice zone, the consolidated ice pack, the size of potential polynyas as
well as the amount of broken ice and open water within the consolidated ice pack. Here we use
sea ice concentration thresholds of $0.15 \leq SIC < 0.80$ to define the width of the MIZ and $0.80 \leq$
$SIC \leq 1.0$ to define the consolidated pack ice. Yet applying the same thresholds for both sea ice
algorithms results in a MIZ from the NASA Team algorithm that is on average twice as large as
in the Bootstrap algorithm and considerably more broken ice within the consolidated pack ice.
Total potential coastal polynya areas ($SIC \leq 0.80$) also differ between the algorithms, though
differences are generally smaller than for the other ice types analyzed.
While the spatial extents of the different ice classes may differ, the seasonal cycle is
generally consistent between both algorithms. Climatologically, the advance of the consolidated
ice pack happens over a much longer period (~7-8 months) than the retreat (~4-5 months),
while the MIZ exhibits a longer advance period (~8-10 months). This seasonal cycle in
expansion/contraction of the ice cover is in general agreement with results by *Stammerjohn et*
*al*. [2008] who showed sea ice retreat generally begins in September at the outer most edge of
the sea ice and continues poleward over the next several months. However, what these results
show is that while the pack ice starts to retreat around September, this in turn results in a further
expansion of the MIZ, the amount of which is highly dependent on which algorithm is used. The
timing of when the maximum polynya extent is reached however can differ by several months
between the algorithms in regions such as the Bellingshausen/Amundsen Sea and the Pacific
Ocean.
Since the MIZ is an important region for phytoplankton biomass and productivity [e.g. *Park*
*et al.,* 1999], mapping seasonal and interannual changes in the MIZ is important for
understanding changes in top predator populations and distributions. However, as mentioned
above, results are highly dependent on which sea ice algorithm is used for delineating the MIZ.
Furthermore, accurately mapping the extent of the MIZ from coarse resolution satellite data
such as that from passive microwave sensors remains problematic. The MIZ is very dynamic in
space and time, making it challenging to provide precise delimitations using sea ice
concentrations that are in turn sensitive to melt processes and surface conditions. Nevertheless
we examined the impact the winter MIZ and consolidated pack ice area as derived from both
algorithms would have on the breeding success of snow petrels the following summer. The
different proportions of MIZ and consolidated pack ice between algorithms affected the
inferences made from models tested even if trends were of the same sign. Given the sensitivity



of the relationships between ice types and breeding success of this species, caution is
warranted when doing this type of analysis as different relationships may emerge as a function
of which sea ice data set is used in the analysis. Further work is needed to validate the
accuracy of the ice types from passive microwave.
**Acknowledgements**
This work is funded under NASA Grant NNX14AH74G and NSF Grant PLR 1341548. We are grateful to
Sharon Stammerjohn for her helpful comments on the manuscript. Gridded fields of the different ice
classifications from both algorithms are available via ftp by contacting J. Stroeve. We thank all the
wintering fieldworkers involved in the collection of snow petrel data at Dumont d'Urville since more
than 50 years, as well as Institut Paul Emile Victor (program IPEV n°109, resp. H. Weimerskirch), Terres
Australes et Antarctiques Françaises and Zone Atelier Antarctique (CNRS-INEE) for support.





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

phytoplankton bloom in circum-Antarctic Polynyas, submitted to Geophysical Research
Letters, in revision.





# Tables


**Table 1.** Sea ice categories defined in this study.

| Region | Definition | Binary Classification Value |
|---|---|---|
| Outer MIZ | Outer region of sea ice with ice concentration between 15% and 80% | 16 |
| Inner Polynya | Region near the coast with concentration < 80% south of 80% concentration | 32 |
| Distant ice | Scattered sea ice regions north of MIZ, possibly islands or atmospheric storms | 48 |
| Pack Ice | Ice concentration > 80% | 80 |
| Inner open water | Concentration < 15% south of MIZ | 112 |
| Open pack ice | Concentration > 15% and < 80% within consolidated ice region | 128 |





**Table 2.** Monthly mean extents of the different ice classes. Values are only listed for the consolidated
pack ice, the marginal ice zone and the potential coastal polynya area. Values are listed in $10^6$ km$^2$.

| | NASA Team | | | Bootstrap | | |
|---|---|---|---|---|---|---|
| **Total Antarctic** | | | | | | |
| Month | MIZ | Polynya | Pack Ice | MIZ | Polynya | Pack Ice |
| January | 2.44 | 0.31 | 1.94 | 2.06 | 0.36 | 2.27 |
| February | 1.51 | 0.20 | 1.18 | 1.25 | 0.22 | 1.49 |
| March | 2.03 | 0.25 | 1.42 | 1.65 | 0.24 | 2.08 |
| April | 2.71 | 0.42 | 3.27 | 1.84 | 0.31 | 4.62 |
| May | 3.07 | 0.62 | 5.85 | 1.97 | 0.37 | 7.79 |
| June | 3.63 | 0.69 | 8.22 | 2.31 | 0.37 | 10.65 |
| July | 4.03 | 0.66 | 10.31 | 2.53 | 0.35 | 13.00 |
| August | 4.75 | 0.62 | 11.29 | 2.88 | 0.34 | 14.49 |
| September | 5.41 | 0.63 | 11.31 | 3.19 | 0.35 | 14.89 |
| October | 5.41 | 0.74 | 10.83 | 3.39 | 0.38 | 14.16 |
| November | 5.62 | 1.02 | 7.92 | 3.69 | 0.63 | 11.10 |
| December | 5.05 | 0.88 | 3.81 | 3.56 | 0.81 | 5.43 |
| Annual | 3.83 | 0.59 | 6.49 | 2.54 | 0.39 | 8.53 |
| **Ross Sea** | | | | | | |
| Month | MIZ | Polynya | Pack Ice | MIZ | Polynya | Pack Ice |
| January | 0.83 | 0.10 | 0.28 | 0.68 | 0.13 | 0.40 |
| February | 0.47 | 0.05 | 0.11 | 0.40 | 0.07 | 0.19 |
| March | 0.62 | 0.10 | 0.34 | 0.45 | 0.09 | 0.57 |
| April | 0.60 | 0.15 | 1.22 | 0.37 | 0.09 | 1.63 |
| May | 0.60 | 0.15 | 1.93 | 0.36 | 0.08 | 2.43 |
| June | 0.67 | 0.15 | 2.29 | 0.40 | 0.08 | 2.91 |
| July | 0.75 | 0.14 | 2.63 | 0.44 | 0.07 | 3.27 |
| August | 0.91 | 0.12 | 2.67 | 0.50 | 0.07 | 3.43 |
| September | 0.98 | 0.13 | 2.64 | 0.54 | 0.08 | 3.46 |
| October | 0.86 | 0.17 | 2.73 | 0.55 | 0.09 | 3.39 |
| November | 0.89 | 0.30 | 2.19 | 0.59 | 0.17 | 2.87 |
| December | 1.17 | 0.32 | 0.92 | 0.76 | 0.26 | 1.45 |
| Annual | 0.78 | 0.16 | 1.67 | 0.50 | 0.11 | 2.18 |
| **Bellinghausen/Amundsen Sea** | | | | | | |
| Month | MIZ | Polynya | Pack Ice | MIZ | Polynya | Pack Ice |
| January | 0.35 | 0.07 | 0.32 | 0.29 | 0.08 | 0.38 |
| February | 0.28 | 0.05 | 0.16 | 0.22 | 0.06 | 0.21 |
| March | 0.37 | 0.06 | 0.10 | 0.27 | 0.07 | 0.21 |
| April | 0.50 | 0.07 | 0.20 | 0.29 | 0.06 | 0.48 |
| May | 0.54 | 0.12 | 0.42 | 0.31 | 0.06 | 0.83 |
| June | 0.63 | 0.16 | 0.66 | 0.37 | 0.05 | 1.17 |
| July | 0.68 | 0.17 | 0.89 | 0.43 | 0.05 | 1.45 |
| August | 0.79 | 0.15 | 1.01 | 0.51 | 0.05 | 1.60 |
| September | 0.84 | 0.14 | 1.00 | 0.51 | 0.05 | 1.62 |
| October | 0.73 | 0.14 | 0.97 | 0.46 | 0.06 | 1.50 |
| November | 0.69 | 0.13 | 0.86 | 0.45 | 0.08 | 1.25 |
| December | 0.57 | 0.11 | 0.55 | 0.42 | 0.11 | 0.72 |
| Annual | 0.58 | 0.12 | 0.60 | 0.38 | 0.06 | 0.96 |
| **Weddell Sea** | | | | | | |
| Month | MIZ | Polynya | Pack Ice | MIZ | Polynya | Pack Ice |
| January | 0.72 | 0.12 | 0.93 | 0.60 | 0.11 | 1.07 |
| February | 0.37 | 0.08 | 0.70 | 0.30 | 0.06 | 0.84 |
| March | 0.47 | 0.06 | 0.87 | 0.38 | 0.04 | 1.07 |
| April | 0.69 | 0.07 | 1.49 | 0.46 | 0.05 | 1.87 |
| May | 0.82 | 0.10 | 2.53 | 0.54 | 0.06 | 3.04 |
| June | 0.96 | 0.10 | 3.62 | 0.64 | 0.06 | 4.21 |
| July | 1.08 | 0.08 | 4.51 | 0.65 | 0.05 | 5.16 |





| Month | MIZ | Polynya | Pack Ice | MIZ | Polynya | Pack Ice |
|---|---|---|---|---|---|---|
| August | 1.39 | 0.08 | 4.73 | 0.75 | 0.06 | 5.62 |
| September | 1.62 | 0.09 | 4.67 | 0.83 | 0.06 | 5.78 |
| October | 1.51 | 0.13 | 4.42 | 0.84 | 0.07 | 5.48 |
| November | 1.53 | 0.31 | 3.34 | 0.86 | 0.14 | 4.56 |
| December | 1.87 | 0.33 | 1.65 | 1.24 | 0.30 | 2.33 |
| Annual | 1.09 | 0.13 | 2.80 | 0.67 | 0.09 | 3.43 |
| **Indian Ocean** | | | | | | |
| **Month** | **MIZ** | **Polynya** | **Pack Ice** | **MIZ** | **Polynya** | **Pack Ice** |
| January | 0.26 | 0.01 | 0.16 | 0.23 | 0.02 | 0.18 |
| February | 0.15 | 0.01 | 0.06 | 0.14 | 0.01 | 0.08 |
| March | 0.24 | 0.01 | 0.03 | 0.24 | 0.02 | 0.06 |
| April | 0.43 | 0.01 | 0.16 | 0.35 | 0.05 | 0.30 |
| May | 0.57 | 0.13 | 0.55 | 0.43 | 0.08 | 0.80 |
| June | 0.75 | 0.14 | 1.04 | 0.53 | 0.08 | 1.40 |
| July | 0.82 | 0.13 | 0.59 | 0.54 | 0.07 | 2.05 |
| August | 0.87 | 0.11 | 2.09 | 0.57 | 0.06 | 2.59 |
| September | 1.03 | 0.12 | 2.24 | 0.67 | 0.07 | 2.81 |
| October | 1.33 | 0.15 | 2.02 | 0.87 | 0.08 | 2.71 |
| November | 1.62 | 0.18 | 1.10 | 1.13 | 0.13 | 1.75 |
| December | 0.94 | 0.07 | 0.37 | 0.74 | 0.09 | 0.55 |
| Annual | 0.75 | 0.10 | 0.96 | 0.54 | 0.06 | 1.29 |
| **Pacific Ocean** | | | | | | |
| **Month** | **MIZ** | **Polynya** | **Pack Ice** | **MIZ** | **Polynya** | **Pack Ice** |
| January | 0.28 | 0.01 | 0.24 | 0.25 | 0.02 | 0.26 |
| February | 0.23 | 0.01 | 0.14 | 0.19 | 0.02 | 0.17 |
| March | 0.34 | 0.02 | 0.10 | 0.31 | 0.03 | 0.15 |
| April | 0.51 | 0.05 | 0.20 | 0.38 | 0.06 | 0.34 |
| May | 0.54 | 0.11 | 0.43 | 0.35 | 0.10 | 0.67 |
| June | 0.61 | 0.14 | 0.62 | 0.38 | 0.11 | 0.93 |
| July | 0.70 | 0.14 | 0.73 | 0.45 | 0.10 | 1.10 |
| August | 0.81 | 0.14 | 0.79 | 0.54 | 0.09 | 1.19 |
| September | 0.93 | 0.14 | 0.76 | 0.63 | 0.10 | 1.17 |
| October | 0.96 | 0.14 | 0.71 | 0.68 | 0.09 | 1.08 |
| November | 0.88 | 0.10 | 0.44 | 0.66 | 0.11 | 0.70 |
| December | 0.49 | 0.05 | 0.30 | 0.41 | 0.06 | 0.38 |
| Annual | 0.61 | 0.09 | 0.46 | 0.44 | 0.07 | 0.69 |




**Table 3.** Comparison of trends in the marginal ice zone, polynyas and the consolidated pack ice for
March through November (1979 to 2013) for both the NASA Team and Bootstrap sea ice algorithms.
Trends are computed in km$^2$ per year. Statistical significance at the 90th, 95th and 99th percentiles are
denoted by +, ++ and +++, respectively. Results are only shown for March through November.

| | NASA Team | | | Bootstrap | | |
|---|---|---|---|---|---|---|
| **Total Antarctic** | | | | | | |
| Month | dMIZ/dt | dPoly/dt | dPack/dt | dMIZ/dt | dPoly/dt | dPack/dt |
| March | +2,900 | +700 | +14,300$^{+++}$ | +4,900 | -300 | +18,000$^{+++}$ |
| April | -8,200 | -500 | +29,600$^{+++}$ | -10,400 | -1000 | +38,000$^{+++}$ |
| May | -9,400 | -2,400 | +35,000$^{+++}$ | -8,500 | -2,200 | +41,300$^{+++}$ |
| June | -10,100 | -5,100 | +32,900$^{+++}$ | -9,200 | -2,400 | +52,400$^{+++}$ |
| July | -3,400 | -5,700 | +22,600$^{++}$ | -6,600 | -2,300 | +25,200$^{+++}$ |
| August | +3,700 | -3,600 | +11,900 | -6,200 | -1,500 | +31,800$^{+++}$ |
| September | +10,900$^{+}$ | -3,300 | +3,700 | -4,200 | -1,400 | +39,400$^{+++}$ |
| October | +9,600$^{+}$ | -4,900 | +7,300 | -4,300 | -2,900 | +25,200$^{+++}$ |
| November | +2,600 | -4,000 | +6,000 | -9,800 | -3,700 | +29,400$^{+++}$ |
| **Ross Sea** | | | | | | |
| Month | dMIZ/dt | dPoly/dt | dPack/dt | dMIZ/dt | dPoly/dt | dPack/dt |
| March | +2,800 | +300 | +4,100 | +1,500 | -100 | +7,700$^{++}$ |
| April | -1,400 | -1,500 | +12,400$^{++}$ | -2,700 | -1,400 | +14,600$^{+++}$ |
| May | +2,600$^{+}$ | -2,200 | +11,100$^{++}$ | -700 | -1,100 | +16,400$^{+++}$ |
| June | 0 | -1,200 | +12,700$^{++}$ | -2,000 | -800 | +18,600$^{+++}$ |
| July | +700 | -700 | +8,200$^{+}$ | -700 | -600 | +14,200$^{+++}$ |
| August | +6,900$^{+++}$ | -1,600 | +3,400 | +500 | -900 | +12.700$^{+++}$ |
| September | +4,800$^{++}$ | -1,200 | +1,800 | -700 | -700 | +15,100$^{+++}$ |
| October | +5,400$^{+++}$ | -2,300 | +7,300$^{+}$ | +1,100 | -1,300 | +17,600$^{+++}$ |
| November | +3,700$^{+}$ | -1,200 | +4,400 | -700 | -1,600 | +13,700$^{+++}$ |
| **Bellinghausen/Amundsen Sea** | | | | | | |
| Month | dMIZ/dt | dPoly/dt | dPack/dt | dMIZ/dt | dPoly/dt | dPack/dt |
| March | -7,500 | -1,500 | -2,800 | -2,400 | -1,700 | -7,500 |
| April | -8,600 | -800 | -3,100 | -3,100 | -900 | -7,700 |
| May | -8,600 | -1,200 | +2,800 | -2,100 | -800 | -4,600 |
| June | -6,800 | -2,600 | +8,500$^{+++}$ | -2,100 | -500 | +1,300 |
| July | -3,500 | -2,500 | +10,100$^{+++}$ | -700 | -700 | +4,000 |
| August | -1,200 | -700 | +7,000$^{+}$ | +500 | -200 | +2,700 |
| September | +2,600 | -500 | -300 | +1,500$^{+}$ | -200 | -100 |
| October | -800 | -200 | -1,100 | -300 | -200 | -1,800 |
| November | +2,600 | +1,000$^{++}$ | -1,400 | +1,600 | +600$^{+}$ | +300 |
| **Weddell Sea** | | | | | | |
| Month | dMIZ/dt | dPoly/dt | dPack/dt | dMIZ/dt | dPoly/dt | dPack/dt |
| March | +4,100$^{++}$ | +1,300$^{++}$ | +9,500$^{++}$ | +2,600$^{++}$ | +600$^{+}$ | +13,600$^{+++}$ |
| April | +1,700 | +400 | +12,000$^{++}$ | -2,000 | +200 | +19,200$^{+++}$ |
| May | -100 | -400 | +9,400$^{++}$ | -1,500 | -600 | +14,400$^{+++}$ |
| June | -2,300 | -900 | +100 | -4,800 | -600 | +8,800$^{++}$ |
| July | -2,900 | -1,100 | -4,800 | -4,200 | -400 | -100 |
| August | -1,700 | -700 | -5,100 | -3,500 | -100 | +600 |
| September | -200 | -600 | -100 | -2,900 | -200 | +4,900 |
| October | +4,300 | -1,400 | -8,800 | -3,700 | -700 | +3,400 |
| November | -2,100 | -3,500 | -4,700 | -6,300 | -2,200 | +700 |
| **Indian Ocean** | | | | | | |
| Month | dMIZ/dt | dPoly/dt | dPack/dt | dMIZ/dt | dPoly/dt | dPack/dt |
| March | +2,500$^{++}$ | +300$^{+}$ | +9,500$^{++}$ | +2,100$^{++}$ | +300$^{+}$ | +1,500$^{++}$ |
| April | +1,500$^{+}$ | +600$^{+}$ | +12,000$^{++}$ | -500 | +300 | +5,200$^{+++}$ |
| May | -200 | +600$^{+}$ | +9,400$^{++}$ | -1,400 | +100 | +7,700$^{+++}$ |
| June | +2,600$^{+}$ | -500 | +100 | +900 | -300 | +7,600$^{++}$ |
| July | +3,500$^{+}$ | -700 | -4,800 | +100 | -100 | +7,600$^{++}$ |
| August | +1,300 | -300 | -5,100 | -1,500 | 0 | +9,900$^{+++}$ |



| September | +4,600[+] | -900 | -100 | +400 | -100 | +6,700[++] |
| October | +1,900 | -900 | -8,800 | -200 | -400 | +8,600[++] |
| November | +2,000 | -200 | -4,700 | -500 | -400 | +8,700[++] |
| **Pacific Ocean** | | | | | | |
| **Month** | **dMIZ/dt** | **dPoly/dt** | **dPack/dt** | **dMIZ/dt** | **dPoly/dt** | **dPack/dt** |
| March | +1,100 | +400[+++] | +2,800[+++] | +1,100[++] | +600[+++] | +1,500[++] |
| April | -1,400 | +800[+++] | +5,600[+++] | -2,100 | +700[+++] | +5,200[+++] |
| May | -3,000 | +800[++] | +6,100[+++] | -2,800 | +300[+] | +7,700[+++] |
| June | -3,600 | +200 | +7,000[+++] | -1,200 | -300 | +7,600[++] |
| July | -1,300 | -700 | +5,700[++] | -100 | -400 | +7,600[++] |
| August | -1,500 | -300 | +2,200 | -2,200 | -300 | +9,900[+++] |
| September | -900 | -100 | +1,400 | -2,500 | -300 | +6,700[++] |
| October | -1,200 | 0 | +3,700[++] | -1,100 | -300 | +8,600[++] |
| November | -3,500 | -500 | +4,400[++] | -4,000 | -200 | +8,700[++] |





**Table 4.** Monthly latitude/longitude corners used for assessment of sea ice conditions on snow petrel
breeding success.

|  | April | May | June | July | August | September |
|---|---|---|---|---|---|---|
| Latitude$_1$ | -65 | -65 | -65 | -65 | -65 | -65 |
| Latitude$_2$ | -60 | -60 | -60 | -60 | -55 | -55 |
| Longitude$_1$ | 90 | 65 | 50 | 35 | 25 | 50 |
| Longitude$_2$ | 120 | 120 | 120 | 120 | 115 | 140 |


**Table 5. Results of model selection for the relationship between pack ice and MIZ on breeding success**
**of snow petrels. Model selection is based on the lowest AIC score, highlighted in gray. The slope of the**
**regression is also shown.**

| Model | Variable | AIC | Slope |
|---|---|---|---|
| Bootstrap | MIZ | 931.86 | -0.57544 |
| NASA Team | MIZ | 887.11 | -1.31416 |
| Bootstrap | Pack ice | 879.17 | -1.04223 |
| NASA Team | Pack ice | 927.8 | -0.41916 |





## List of Figures

**Figure 1.** Example of a radial profile from 50 to 90S at -11.60 degrees West on 3 September 1990, showing the different sea ice classifications found along this transect.

**Figure 2:** Samples of ice classification on day 70 (March) and day 273 (September) 2013. Results are shown for both the NASA Team (top) and Bootstrap (bottom) sea ice algorithms. The MIZ (red) represents regions of sea ice concentration between 15 and 80% from the outer ice edge, the pack ice is shown in light purple, representing regions of greater than 80% sea ice concentration. Orange regions within the pack ice represent coherent regions of less than 80% sea ice concentration, pink areas open water and green regions of less than 80% sea ice concentration near the Antarctic coastline. Dark blue represents the ocean mask applied to remove spurious ice concentrations beyond the ice edge.

**Figure 3.** Southern hemisphere regions as defined by *Parkinson and Cavalieri* [2012].

**Figure 4**. Location of the mean 1981-2010 outer marginal ice edge for both the NASA Team and Bootstrap algorithms.

**Figure 5.** Long-term (1979-2013) seasonal cycle in total Antarctic extent of the consolidated pack ice, the outer marginal ice zone, polynyas, open pack ice (or broken ice within the pack ice), and inner open water. There are essentially no scattered ice floes outside of the MIZ. NASA Team results are shown on the left and the Bootstrap on the right.

**Figure 6.** Long-term (1979-2013) seasonal cycle in regional sea ice extent of the consolidated pack ice, the outer marginal ice zone, polynyas, open pack ice (or broken ice within the pack ice), and inner open water. Results for the NASA Team algorithm are shown on the left and Bootstrap on the right, and for the Ross, Bellingshausen/Amundsen, Weddell, Indian and Pacific Oceans.

**Figure 7.** Expansion (red) or contraction (blue) of the outer ice edge (top), the width of the marginal ice zone (middle) and the width of the pack ice from 1979 to 2013 during the month of September.

**Figure 8.** Daily trends (1979 to 2013) in the consolidated pack ice, the outer MIZ and potential coastal polynyas for the entire Antarctic sea ice cover for the NASA Team (left) and Bootstrap (right) algorithms. Trends are provided in $10^6$ km$^2$ a$^{-1}$.

**Figure 9.** Daily (1979-2013) trends in regional sea ice extent of the consolidated pack ice (top), the outer marginal ice zone (middle) and potential coastal polynyas (bottom). Results for the NASA Team algorithm (left) and Bootstrap (right) are shown as a function of longitude. Trends are provided in $10^6$ km$^2$ a$^{-1}$. Note the difference in color bar scales.

**Figure 10.** Time-series of seasonal mean JJA (top), SON (middle) and MAM (bottom) marginal ice zone (left) and consolidated pack ice (right) for both sea ice algorithms; NASA Team is shown in red, Bootstrap in black. Shading represents one standard deviation. Note the difference in y-axis between the pack ice and the MIZ plots.





**Figure 11.** Breeding success of snow petrel (top) and effect of the Bootstrap pack ice on the breeding
success of snow petrels (bottom).




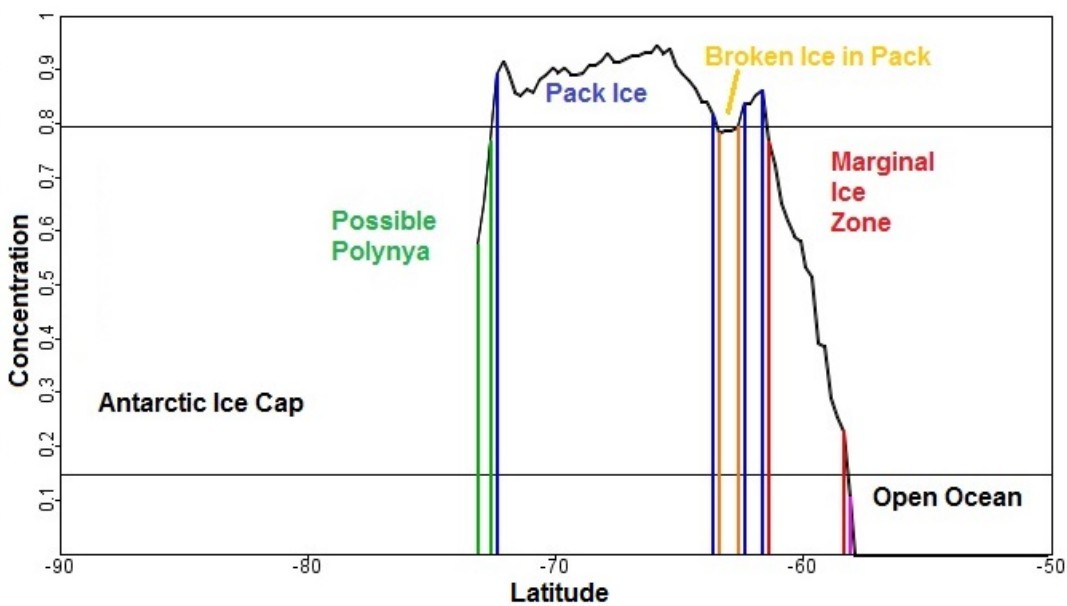


**Figure 1.** Example of a radial profile from 50 to 90S at -11.60 degrees West on 3 September 1990,
showing the different sea ice classifications found along this transect.





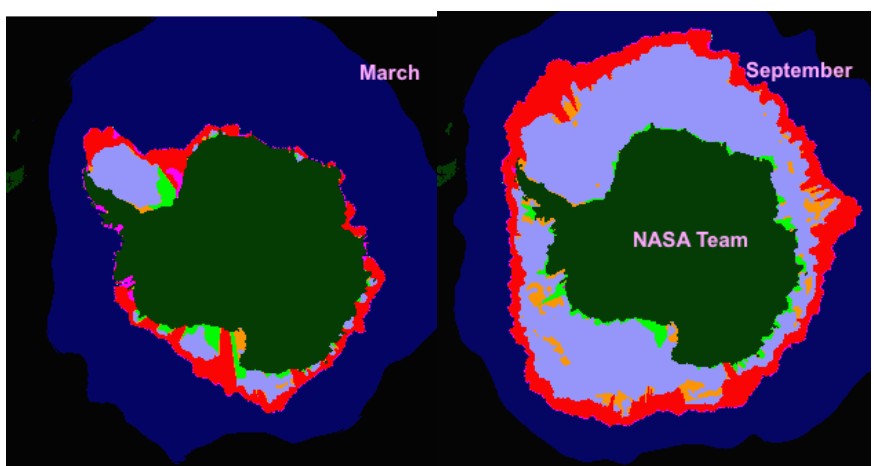


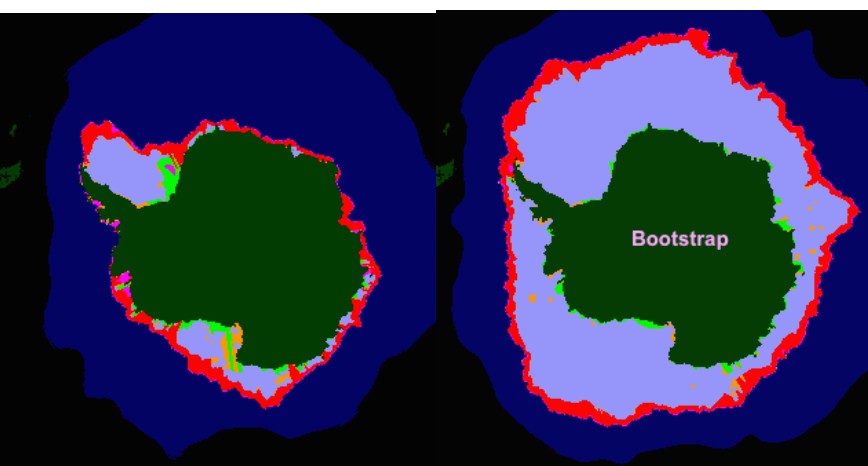


**Figure 2:** Samples of ice classification on day 70 (March) and day 273 (September) 2013. Results are shown for both the NASA Team (top) and Bootstrap (bottom) sea ice algorithms. The MIZ (red) represents regions of sea ice concentration between 15 and 80% south of the outer ice edge (defined by the ocean mask) and north of the pack ice. The pack ice is shown in light purple, representing regions of greater than 80% sea ice concentration. Orange regions within the pack ice (and away from the coastline) represent 'broken ice areas', coherent regions of less than 80% sea ice concentration. Pink areas are open water (SIC < 15%) areas detected south of the ocean mask but north of the coastline, and light green areas of less than 80% sea ice concentration extending from the Antarctic coastline are potential coastal polynyas. Dark blue represents the ocean mask applied to remove spurious ice concentrations at and beyond the ice edge.




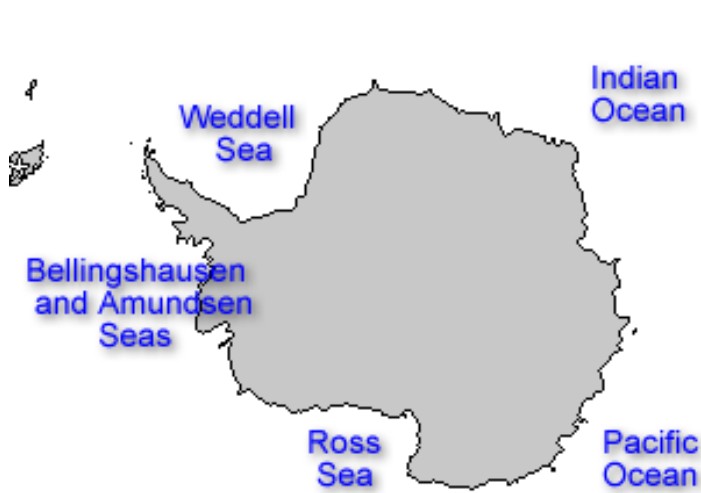


**Figure 3.** Southern hemisphere regions as defined by *Parkinson and Cavalieri* [2012].




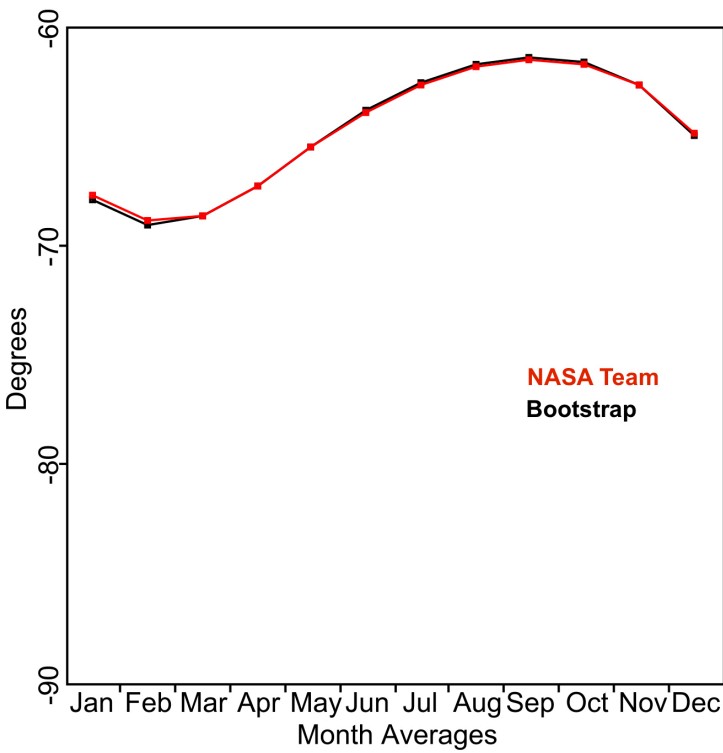


**Figure 4**. Location of the mean 1981-2010 outer marginal ice edge for both the NASA Team and
Bootstrap algorithms.





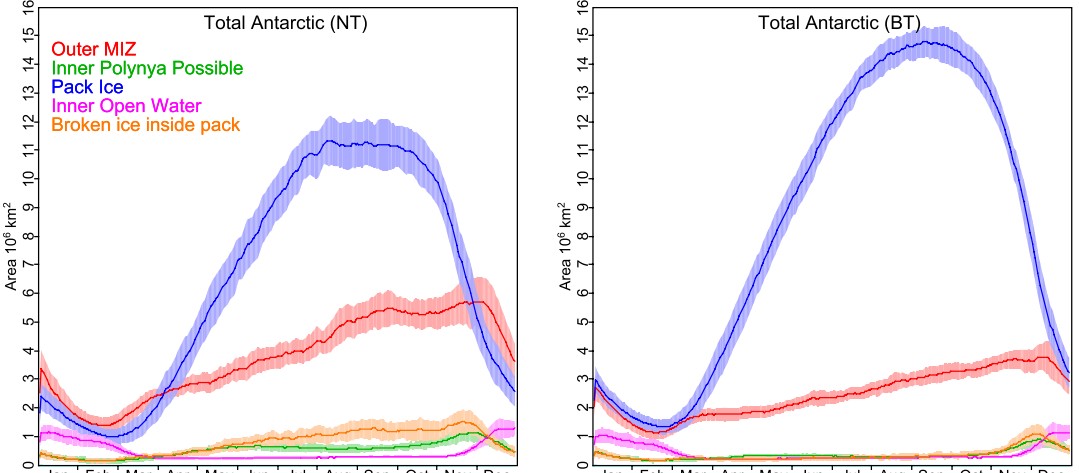

**Figure 5.** Long-term (1979-2013) seasonal cycle in total Antarctic extent of the consolidated pack ice, the outer marginal ice zone, polynyas, broken ice within the pack ice, and inner open water. There are essentially no scattered ice floes outside of the MIZ. NASA Team results are shown on the left and the Bootstrap on the right.





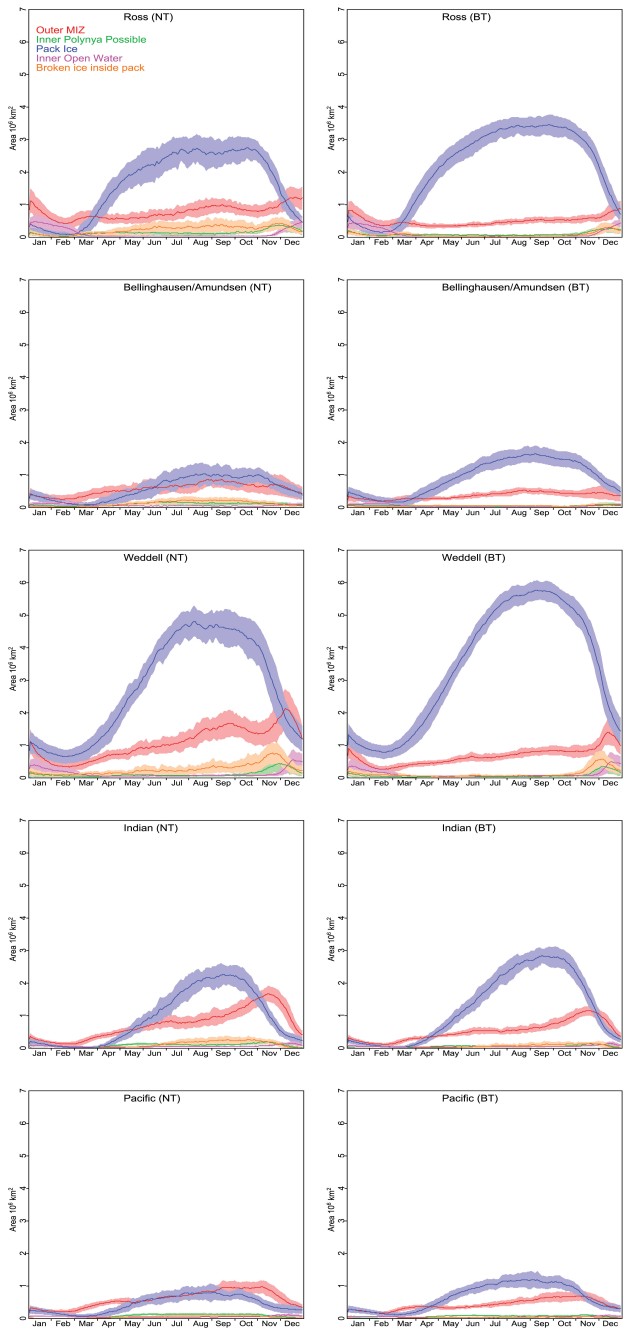

**Figure 6.** Long-term (1979-2013) seasonal cycle in regional sea ice extent of the consolidated pack ice, the outer marginal ice zone, polynyas, broken ice within the pack ice, and inner open water. Results for the NASA Team algorithm are shown on the left and Bootstrap on the right, and for the Ross, Bellinghausen/Amundsen, Weddell, Indian and Pacific Oceans.



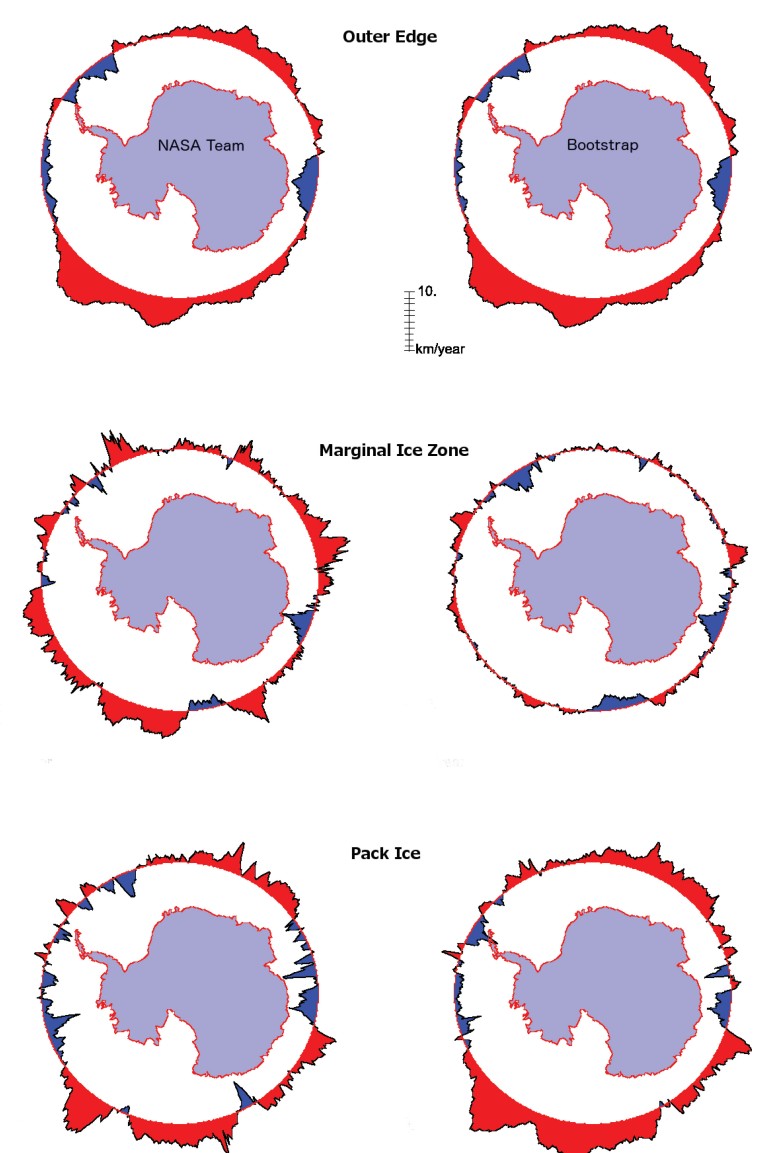

**Figure 7.** Expansion (red) or contraction (blue) of the outer ice edge (top), the width of the marginal ice
zone (middle) and the width of the pack ice from 1979 to 2013 during the month of September.



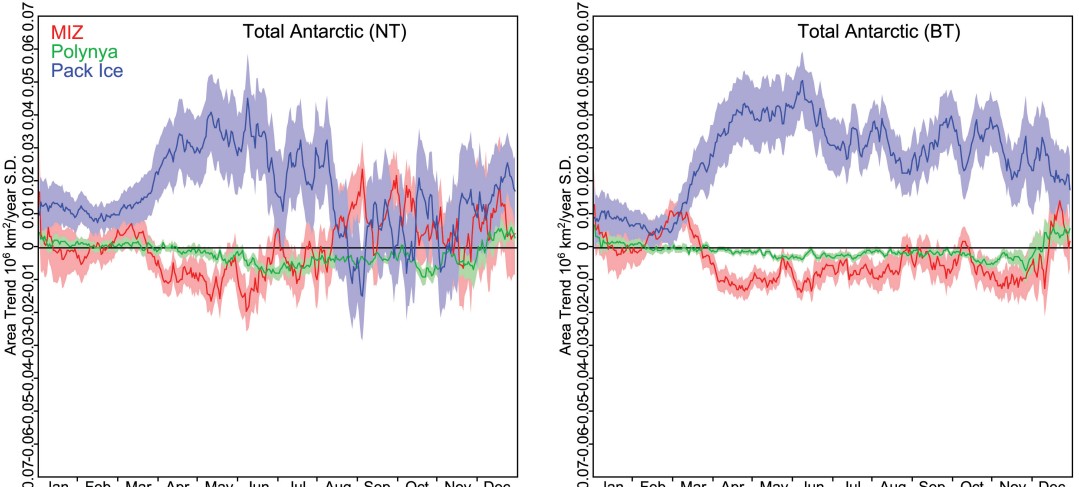

**Figure 8.** Daily trends (1979 to 2013) in the consolidated pack ice, the outer MIZ and potential coastal polynyas for the entire Antarctic sea ice cover for the NASA Team (left) and Bootstrap (right) algorithms. Trends are provided in $10^6$ km$^2$ a$^{-1}$.




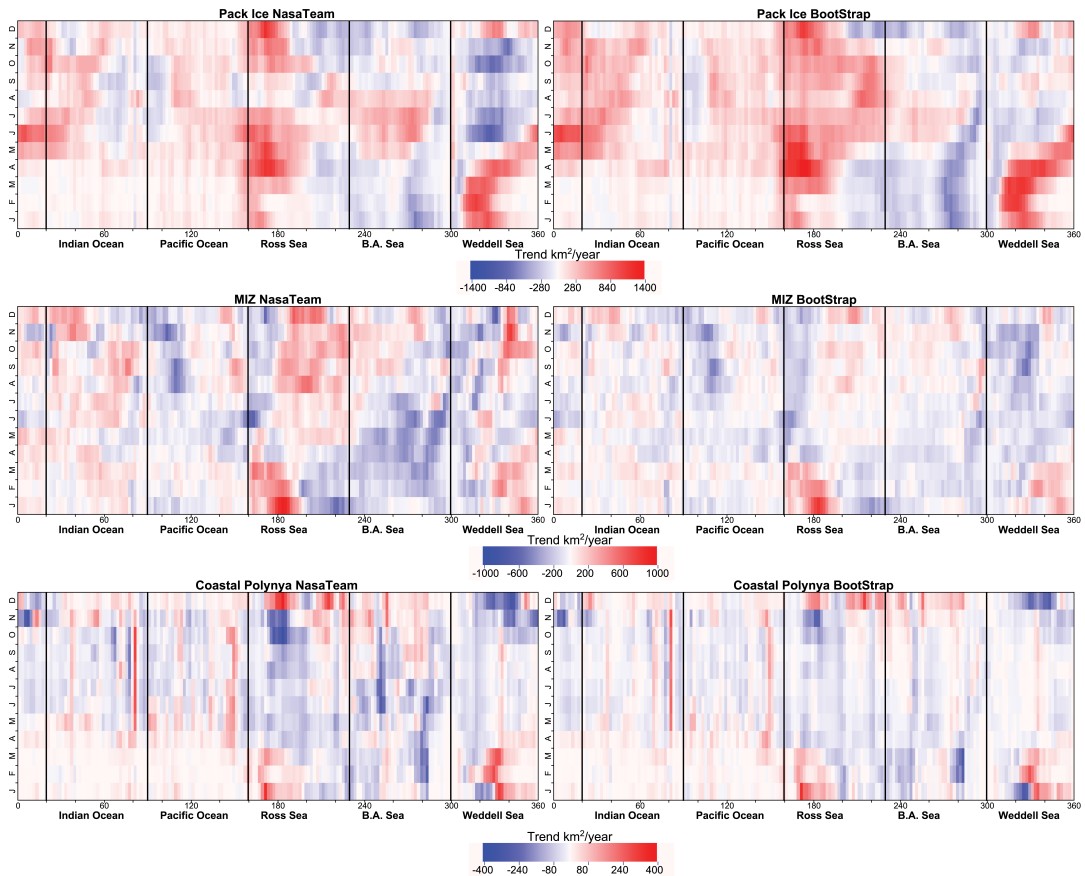


**Figure 9.** Daily (1979-2013) trends in regional sea ice extent of the consolidated pack ice (top), the outer
marginal ice zone (middle) and potential coastal polynyas (bottom). Results for the NASA Team
algorithm  (left) and Bootstrap (right) are shown as a function of longitude. Trends are provided in $10^6$
$km^2\ a^{-1}$. Note the difference in color bar scales.






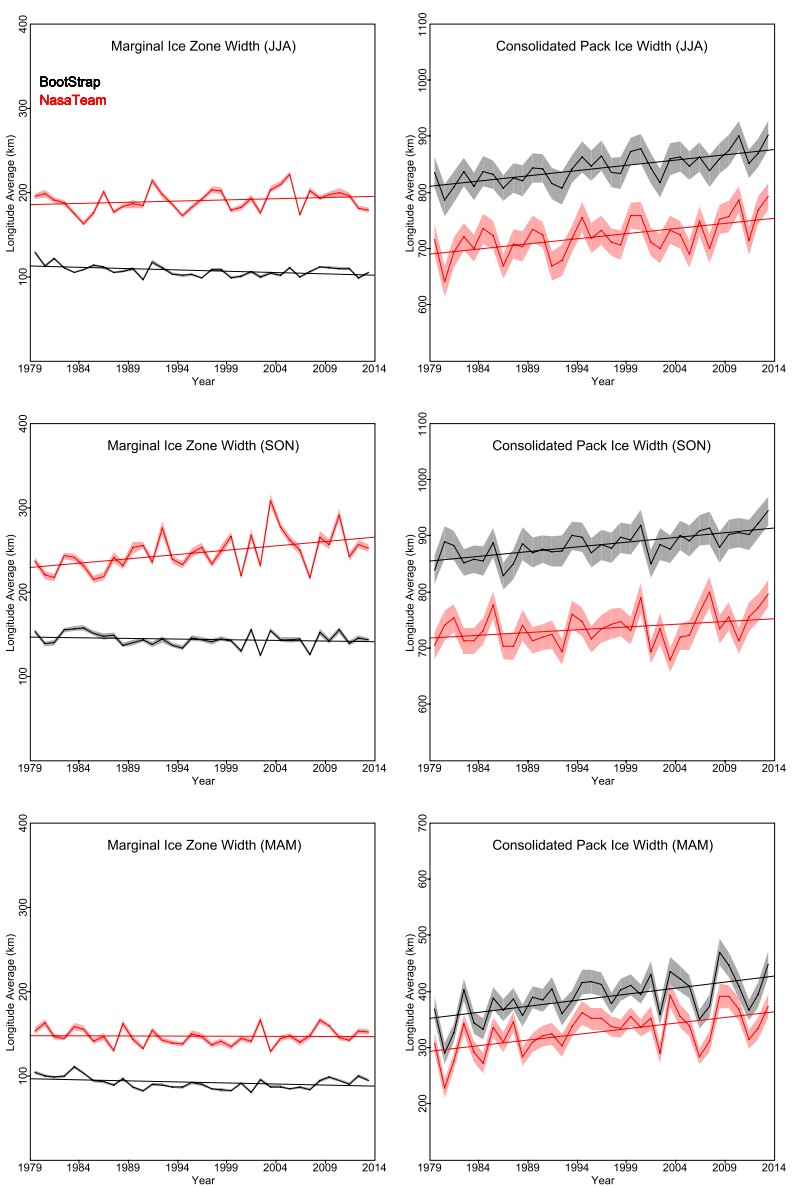

**Figure 10.** Time-series of seasonal mean JJA (top), SON (middle) and MAM (bottom) marginal ice zone (left) and consolidated pack ice (right) for both sea ice algorithms; NASA Team is shown in red, Bootstrap in black. Shading represents one standard deviation. Note the difference in y-axis between the pack ice and the MIZ plots.




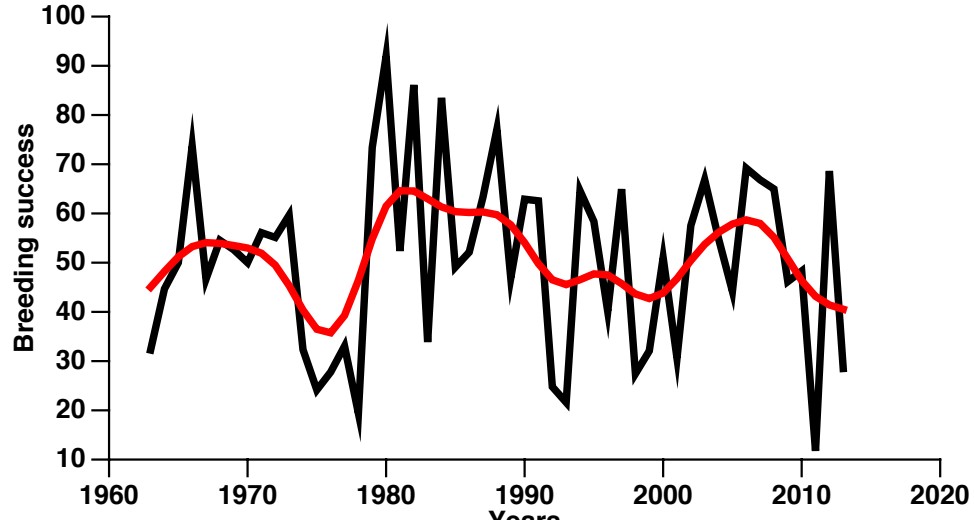


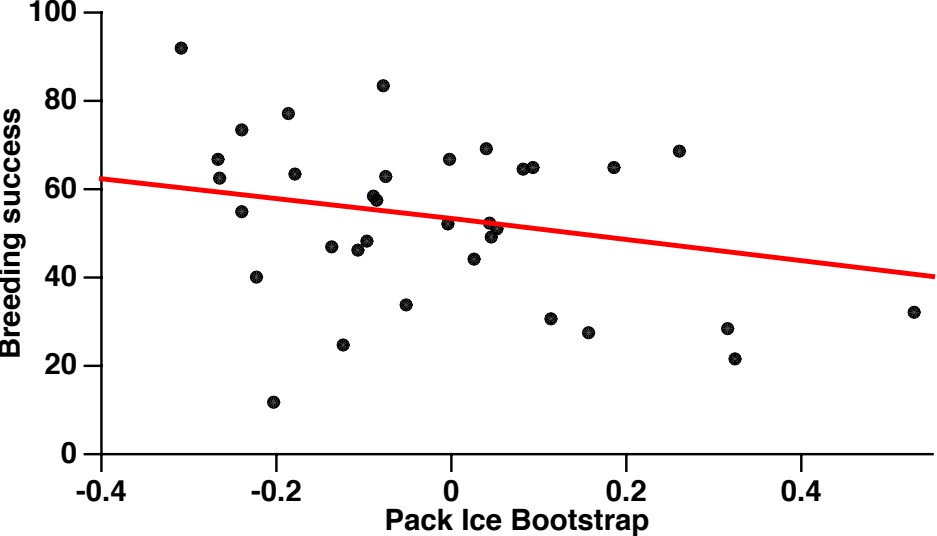


**Figure 11.** Breeding success of snow petrel (top) and effect of the Bootstrap pack ice on the breeding
success of snow petrels (bottom).

