# Peer review of "Mapping and Assessing Variability in the Antarctic Marginal Ice Zone, the Pack Ice and Coastal Polynyas"

_The Cryosphere, 2016_

## Referee Comment (RC1) · H. Flores (Referee) · 29 Mar 2016

General comments

Julienne Stroeve and her co-authors compare the results of two popular algorithms using passive microwave satellite data to classify ice types in the Antarctic sea ice zone, the NASA Team and the Bootsrap. They show that sea ice extent estimates are largely consistent between the two algorithms. They differ, however, in the proportion of consolidated pack-ice versus marginal ice zone (MIZ) and polynias, with greatest discrepancies in the contribution of the MIZ. When applied to biological datasets, in this case the breeding success of snow petrels Pagodroma nivea, however, these discrepancies can lead to opposite conclusions. This manuscript presents a highly desirable

critical cross-validation of satellite-derived data. In particular, the inclusion of a biological dataset adds high value to both physical and biological communities, highlighting the importance of such exercises in this under-studied cross section of disciplines. Furthermore, it was a pleasure reading this manuscript, since it manages to present a complex topic in an easily understandable language, even to non-physicists. Having said this, there are a few things that should be improved: 1) The results and discussion section are not well separated. To preserve the excellent flow of the manuscript, I recommend to merge them into a "Results and Discussion" section. If this is not possible, speculations and literature references should be consequently moved from the results section to the discussion. 2) In my view, the greatest weakness of the approach chosen is the definition of the 'polynia' ice type. Just using a proportional ice coverage of 0.8 near the coast as a criterion may incur confusion with other more open ice situations. It may well be that the two algorithms 'see' completely different things, and therefore result in a large difference in the seasonal occurrence of 'polynias'. 3) There is little information provided on the statistics of GLMs looking at the breeding success of snow petrels. A convincing statistical approach, including model selection, is the fundament of any conclusions concerning the seemingly opposite outcomes of the two algorithms.

Specific comments

Introduction Ll 44-66: When addressing regional variability, the strong decline in the WAP should be mentioned with respect to the final objective to discuss ecological implications. Also, previous declines in Antarctic SIE should be mentioned. For example, Flores et al. (2012) note that "This growth, however, has so far not compensated for a decline of the average sea ice coverage between 1973 and 1977, which accounted for $\sim 2 \times 10^6$ km2 (Cavalieri et al. 2003, Parkinson 2004). Reconstruct ions of the position of the ice edge in the pre-satellite era give strong evidence that the overall areal sea ice coverage in the Southern Ocean declined considerably during the second half of the 20th century (Turner et al. 2009a)".

M&M Ll151-152: Could be problematic with respect to ecological interpretations, because it also includes areas with loser pack ice and wakes behind icebergs that are not polynias and thus do not necessarily feature the same biological dynamics. Also: What is "near" the coast precisely?

Results Ll 413-526: More details on the model statistics are needed. Were the slope terms and intercepts significant? Using differences in AIC alone for model selection can be tricky. I recommend testing the 'better' model against the 'next worse' using ANOVA or F statistics, depending on the model applied. It would be useful to see the model validation plots in the supplementary material. To me it is not clear why the model using pack ice and BT is preferred.

Discussion L619: Another way to validate algorithms would be the ASPEcT ship observation data

Conclusions Ll643-646: I have the feeling that the results of the polynia estimation are blurry in both algorithms, resulting in this large variability of timing of polynya maxima. This may be due to inaccurate definition of polynyas in the analysis.

Technical corrections

Abstract L14: replace "biological" with "biologically"

Introduction Ll 40-43: Split this sentence in two. L59: I believe it should say "maxima" L94: Did you mean "continuously"? L114: replace "mattes" by "matters"

Results L197: Better say "Results and discussion"? L215: delete "(e.g. the shading)" L373: replace "significantly" by "significant L375; delete "(e.g. spring)" Figure 5: What is the variability measure here indicated by the shaded areas around the curves?

---

## Referee Comment (RC2) · S. Kern (Referee) · 6 Apr 2016

Review of

Mapping and assessing variability in the Antarctic marginal ice zone, the pack ice and coastal polynyas

by

Stroeve et al.

Summary: Sea ice concentration data sets obtained for 1979-2014 with two different algorithms (NASA Team and Comiso Bootstrap) from satellite microwave radiometry are

Interactive
comment

used to investigate the spatio-temporal evolution of a number of different ice regimes. These regimes are defined by sea ice concentration thresholds and by their position relative to the coast and/or open water along radial transects through the ice cover. Classified maps are subsequently used to compute the circum-Antarctic and regional extent of the ice regimes selected. The focus is laid onto ice regimes "marginal ice zone (MIZ)", "pack ice" and "possibly coastal polynya". The spatio-temporal distribution of these ice regimes is investigated and discussed with focus on the difference between both algorithms and on trends (and their significance) over the period investigated (1979-2014). At the end the paper tries to link the satellite observations with biological observations.

General comments: The reviewer is allover in line with the general content and also the main message of the paper. There are several issues, however, which the authors could improve and/or decide upon to substantially improve the paper. Points 1), 2) and 5) are those which made the reviewer to decide to rate rigour and impact as "fair" 1) One of the two algorithms used is - to the opinion of the reviewer - well known to provide more reliable results for Antarctic sea ice. The authors tried to avoid to state this upfront in their paper. Also in the discussion of their results the reviewer feels that the authors could have elaborated more on the known differences between the two algorithms and on their, partly known, reasons. The reviewer feels that this would come closer to the "Assessing Variability" part stated in the title of the paper. 2) The reviewer feels that the explanation of the methodology, its sensitivity to chosen parameter values, its limitations, and the relevance of the results could be improved: a) Some specific information about the methodology is missing (as is mentioned in the specific comments. b) An investigation about the sensitivity of the chosen threshold to separate MIZ from pack ice is missing. The authors might ask themselves how their results would look like if they would have used different sea ice concentration thresholds. c) The inclusion of ice regime "possibly coastal polynya" into the analysis seems not to be straightforward. The main reason for this is the fact that many polynyas are sub-grid scale features and therefore produced somewhat noisy signals in the analysis. Instead, as is suggested

by Figure 5, the ice regime "broken ice in the pack ice" might deserve more attention because this is on average the third-largest fraction the ice regimes chosen occupy in the NT data - at least with the chosen threshold of 80% sea ice concentration. In short: I would suggest to skip "possibly coastal polynya" and include "broken ice". d) As with regard to the relevance it would have been good to see how relevant results are. A trend in a regime of 10 km / decade manifests itself in 1-1.5 grid cells over the entire time period looked at. How sure are the authors that their results are solid? 3) The reviewer was confused a bit by the 3 or 4 different over-arching names the authors used for MIZ, pack ice, etc. Harmonizing these and not using "ice type" would certainly improve readability of the paper. 4) The text with regard to interpreting Figure 9 seemed a bit long and confused sometimes. Here the reviewer feels that focussing more on the easily to identify features would help the reader. Also, as detailed in the specific comments, a shift of the starting point from 0degW to 60degW would enhance interpretation of the figure. The authors might also want to think about difference maps between trends obtained for NT and BT. 5) While the reviewer applauds the inclusion of the biology at this stage as an external means to assess the satellite observations, the reviewer feels that a more detailed description is required to fully understand and evaluate the results presented in Figure 11 and Table 5. In the discussion the paper focusses more on climatic issues and potential linkages instead of discussing the potential limitations of the approach (see 2) and the added value included by the inter-disciplinarity. If this paper aims to advertise that through interdisciplinarity value is added to such kind of an analysis of a geophysical data set, then it has failed because it is neither visible from the title nor is this topic given enough weight, in comparison to the climate issues, in the discussion, and the future potential of using such inter-disciplinary approaches is also not discussed further. This part does not really seem to be integrated into the paper.

Specific comments: Abstract: First impression is that > 50% of the abstract do not reflect results of this study. Perhaps the sentence "Knowledge of the ... contraction in others." could be replaced by a sentence describing the methodology used in the

paper - which is currently missing in the abstract.

L49-51: In this context, the authors might want to refer to the work of Reid et al., and Simmonds, both in Annals of Glaciology, 56(69), 2015.

L67-75: The reviewer is wondering whether the authors might want to include also the work of Holland and Kwok, Nature Geoscience, 5, 2012, which nicely suggests the different effects and regions of predominantly dynamic control and thermo-dynamic control. Holland, The seasonality of Antarctic sea ice trends, Geophys. Res. Lett., 41, 2014, might also be a paper to take a look at and to include here.

L81: The reviewer suggests to refer also to the book edited by M.O. Jeffries: Antarctic Sea Ice, Physical processes, interaction and variability, Antarctic Research Series, 74, AGU, 1998.

L82: The authors write of a "dynamic MIZ". Is there also an "non-dynamic MIZ"? Comment: The authors define the MIZ basically via the dynamic processes and in a way refer to the action of waves to fracture the sea ice and generate smaller floes. While on can imagine that the width of the MIZ can be defined during on-ice wind or swell events by the penetration of swell into the sea ice The reviewer is wondering how one defines the MIZ during the other times, i.e. when there is no on-ice wind of swell, the pressure is released and the sea ice is following perhaps a divergent motion with a lot of openings which are refreezing. How does on define the MIZ here?

L83: "longer and larger ice-free summer". What is meant by "larger ice free"?

L84-94: What the reviewer is missing here is the special character of the MIZ in the Antarctic during sea ice growth / ice edge advance; the suggestion is to get back to the "good old pancake ice cycle" notation of Lange et al. 1989, Annals of Glaciology, 12, 92-96 and Lange and Eicken, 1991, Annals of Glaciology, 15, 210-215. Evven though these are old papers there is recent evidence that this is still valid: e.g. Ozsoy-Cicek et al., Deep Sea Research part II, 58(9-10).

L89: "ocean waves define ..." Is a repetition of L82.

L92-96: The polynya part comes a "bit naked" with regard to citations. Suggestion: Morales-Maqueda et al., 2004, Review of Geophysics, 42, and later on, when it comes to the role of ice production perhaps: Drucker et al., Geophys. Res. Lett., 38, 2011.

L119: At the end of this line one could add the reference of Comiso et al., 1997, Remote Sensing of Environment, 12 and of Comiso and Steffen, J. Geophys. Res. 106(C12), which both nicely illustrate to pros and cons of the two algorithms the authors are using in their study.

L126: The reviewer is not too happy with the term "ice type" as the authors use it here. Ice types are given in the WMO nomenclature of sea ice and are terms such as thin first-yer ice, shuga, nilas, grey ice, pancake ice, etc. What the authors define here in their work can perhaps be termed better by "sea ice regime" or "ice category" - a name the authors use for instance in L180 anyways, while in L187/188 the authors name it "ice classes". Perhaps sticking with one term would be a good idea. In case the authors decide to stay with "ice type" then the suggestion is to state clearly here that they are not referring to ice types in the classical sense but use the term "ice type" to differentiate between MIZ, pack ice, and polynya - and even more "broken ice in the pack".

L138/139: Here it might help to mention that the NSIDC has combined these two algorithm to build the first NSIDC sea ice concentration CDR (Peng et al., 2013, Earth System Science Data, 5)

L145: "heavily influenced" Could this statement be precised?

L148: The authors could note here that presumably this definition of the border between MIZ and pack ice is not connected to the other, more dynamic definition of the MIZ via the penetration depth of waves into the sea ice. Is that correct?

L152: Work carried out previously used different sea ice concentration tresholds: e.g.

70%, e.g. Parmiggiani et al., 2006, International J. Rem. Sens. 27(12) or Massom et al. 1998, Annals of Glaciology, 27, 420-426. The authors could comment on their choice of using 80%.

L153: What is the longitudinal sampling of the radial transects? Or in other words: How many transects did the authors use? The authors could perhaps also shed more light on how the transects were placed. The data are on a polar-stereographic grid. Therefore radial transects which have, e.g. a 2-pixel spacing at 55 deg South may overlap the same pixel further pole-ward. How is this treated in the algorithm? Is it correct to say that the automatic classification scheme is simply kind of convering a daily sea ice concentration map into a binary map where specific (which?) values are assigned to pixels of the respective ice category. In other words, the range of 0 ... 100 is condensed to 5 values (according to Figure 1 and Table 1). Hence there is no computation involved - in contrast to what is stated further down in L189. Is Table 1 really needed. Will the information given therein used later in the paper?

L172: Polynyas may also form offshore of a fast ice region. This occurs for instance in the Eastern Weddell Sea or the Indian Ocean sector. Are such areas counted under category "broken ice inside pack ice" or "inner open water"? What about the Cosmonaut polynya developing in the Indian Ocean sector? There are regions like closely east of the Antarctic Peninsula and in the Western Ross Sea where the radial transects do, when coming from the north, first transect sea ice / open water, then land, and then again sea ice / open water. How are these transects treated?

L173/Figure 2: Is there a change to instead list the colors in the caption annotate these in form of small boxes in or underneath the block of images? This way the authors dould avoid potential misinterpretation of the Figure by the individual way people allocate actual colors to the name given plus by the way printers interprete colors.

L174: Suggest to give the actual dates instead of the day of the year.

L189: How did the authors compute gridded fields and regional averages from binary,

i.e. classified maps? The reviewer's guess is, the gridded field is not computed but simply obtained by the classification process (see comment to L153). Subsequently, the classified gridded fields are used to compute the areas covered by the different ice regimes - for the entire southern ocean and the 5 regions - on daily temporal scale. The resulting time series of ice regime areas is then used to compute monthly mean ice regime areas. Is this correct? The reviewer had diffulties to understand this the way it is currently written and found myself thinking about how - in one grid cell - a temporal change between different ice regimes is treated. Which grid cell area file did the authors use to compute the ice regime areas? Since the authors did not use data gridded on the EASE grid, the grid-cell area at latitudes different from 70 deg South differ from 625 km^2.

L190: Figure 3 shows five instead of six regions. A typo?

L191 / Figure 3: Does figure 3 on purpose omit showing the borders between the regions? At least in my print out there are no border lines.

L192: Can the authors comment on the time period chosen? Why, if the sea ice concentration data sets are available from October 1978 until today they choose a shorter period? Sea ice concentrations based on SMMR, i.e. 1981-07/1987, are only observed every other day. Has this been taken into account in the averaging process?

Figure 4 and its discussion: Figure 4 seems not to using the space allocated for it efficiently. Suggestions: i) Start at a southern latitude of 80 degS or even 75 degS; ii) include the variability of the ice edge location either by using bars showing +/- on standard deviation or by a shading similar to Figure 5. The reader might be interested to see whether one algorithm varies more than the other one. Even though both algorithms seem very similar one can see a southward displacement of the Bootstrap ice egde during summer and a northward displacement of the Bootstrap ice edge during winter. The authors could detail a bit more how large this displacement is ... and perhaps write that this is of the order of one 25 km grid cell (which is how it looks like).

The authors state that the large emissivity difference between open water and sea ice drives the location of the ice edge. This is partly true only. Another reason why both algorithms show such close agreement in the ice edge location could be the application of a (the same?) weather filter which is known to cut off low ice concentrations [Ivanova et al., 2015].

L206: The authors speak about an extent, i.e. the sum of all grid cell areas within one ice categorie without any weighing carried out with the actual sea ice concentration, while the y-axis in Figure 5 denotes the quantity given as "Area". Would it make sense to, in order to avoid misunderstandings with the classical sea ice extent and sea ice area, also annotate the y-axes with "Extent"? - Another comment to Figure 5 - later in the paper it becomes even more clear that NT and BT are different particularly in terms of ice categories MIZ and pack ice. Not too much could be said about the ice category "possibly polynya" which results seem to be quite noise and also difficult to interpret given that these tend to be a sub-grid scale phenomenon during winter and only seem to become important during November/December. Here, in Figure 5 the 3rd largest difference between NT and BT is not observed for ice category "possibly polynya" but for "broken ice inside pack". Did the authors carried out similar analysis with this ice category like they did with "possibly polynya"? I could imagine that considering and looking at ice category "broken ice inside pack" could be more enlightening than "possible polynyas".

L214: "ice types" –> "ice categories"

L223: Seems the peak extent for NT is in July.

L224/225: This statement for BT is also for September?

L271: The Bellingshausen/Amundsen Sea is the only region where the maximum MIZ extent does NOT occur after the maximum pack ice extent during spring.

Figure 6: - Please correct "Bellinghausen" to "Bellingshausen". - The lines displaying

other ice categories than MIZ and pack ice are very difficult to delineate in the images; one could try to increase the size of the Figure and make it to extend over two pages or to leave these other 3 categories out.

L272-282: The authors could also mention the large interannual variability in the MIZ extent for NT compared to BT.

L283: The authors could add "and bottom" behind "4th row" because they are referring to both regions before "[Figure ...]".

L291-303: While the maximum polynya extents can be identified in Figure 6 while they occur during late spring or summer they cannot be identified when they occur during winter. The authors therefore indeed might want to increase the size of the images in Figure 6 and let it extend over two pages.

L306-317, Figure 7: Could the authors mention what the reference location / latitude is against which they define whether the ice edge expands or contracts over time? One could assume that this is the latitude given in Figure 4 but one cannot find an obvious hint towards this. Perhaps it is arbitrarily chosen? And an issue coming from personal taste: An expansion of the ice edge is usually associated with cold / colder conditions. And usually cold is linked to bluish colors while warm is linked to reddish colors. It is the other way round in Figure 7. - The authors could include into the caption that the bar in the range of the two upper maps give a reference for the trend in latitudinal movement of the extent of the respective ice category. This scale says 10 km / year which means that, e.g., in the Ross Sea there are regions where over the 30 years considered here the ice edge expanded by 300 km. Is that true? - The reviewer is surprized to see that even though we look at a 35-year long time series of data the trends in ice category expansion or contraction can be quite noisy - particularly for MIZ NT in the Indian Ocean sector north of Amery ice shelf or in the transition between Ross and Amundsen Sea. The same is true in a way to Pack ice NT. Can the authors comment on that? - Please increase the font size of the algorithm names in the two top

maps of Figure 7.

L318 ff and the middle and bottom row of Figure 7: Could the authors also here note what the reference extent is against which they plot the trend in the expansion / contraction?

L342-359 / Figure 8: - Please increase the font size and only show every second y-axis label to increase readability. - Following up with one of my early comments: The y-axis is again giving "Area" even though you mean "Extent". - Is there a reason why the authors switch between p-values and confidence level in percent? L356: Actually NT does not show positive MIZ extent trends anymore in November.

Figure 9: This is a very condense figure with regard to the annotation - particular of the y- and x-axes. The reviewer is wondering whether the authors would consider to either increase the size of the images themselves and therefore of the figure by itself or whether they find a different solution for this problem. The images themself are large enough to identify their message. - Is there a chance to highlight in the images which of the trends are significant? - The "possible polynya" cases so far have not been discussed that much and admittedly are also more noisy and difficult to interprete. How about the authors consider to skip that ice category completely in Figure 9 for the sake of presenting the other two more clearly? - The longitude 60W is a natural break point in the sea ice cover. How about the authors re-organize the images such that they start at 60W and end at 61W, i.e. in the Bellingshausen Sea? This way the authors could better visualize the eastward progression of the areas of positive pack ice trends with season in the Weddell Sea seen by both algorithms. This way the Weddell Sea would also not be split up. - Did the authors think to take a look at difference plots between the trends of the two algorithms? Such a difference plot might highlight the differences between the two algorithms even better.

L370/371: It would fit to add a sentence here which underlines that in the Ross Sea also in this view (i.e. the way presented in Figure 9) the trends in BT MIZ extent are

small (and not significant?).

L373: add "significant" after "statistically".

L379-382: The reviewer has difficulties to understand this argument. Southerly winds, i.e. winds blowing from the South to the North, cause the sea ice to be advected north - presumably with a substantial amount of lead formation. Openings generated by this more divergent type of sea ice motion should be seen by both algorithms in the same manner. Instead of voting for an additional oceanic influence (because this should again affect both algorithms the same way) the reviewer would vote for a smaller sensitivity to thin ice growing in openings and leads for BT than for NT. The authors could have a look into papers comparing these two algorithms. Also the paper by Holland and Kwok, 2012, mentioned further up, could shed some light on this issue.

L383-385: This statement holds up until May, yes, for both BT and NT and the pattern re-emerges in October in NT and December in BT

L390: Not even the large negative ones in the Weddell Sea in NT?

L396-398: Having read this one gets the impression that there is more broken sea ice left after summer melt in the region 30W to 0E. Also very interesting to see that the major positve trend in MAM (NT) or MAMJ (BT) is confined to a very narrow longitude band which moves to the East with progressing season. And it is this narrow longitude band where already in June negative trends pop up. Interesting.

L402/403: Suggestion to not comment on the intital retreat in September / October here; for ice category "pack ice" BT trends are almost zero and NT trends as well, they become larger in November / December.

L404: Which is an interesting notion given the fact that the eastern B.A. Seas have this finger-like structure of substantially negative trends extending from January well into July; NT pack ice trends are weaker here.

L405ff: The authors mention both algorithms for MIZ trends but in BT they are very

small except perhaps far east in May/June; in contrast there is a large blobb of negative MIZ extent trends visible for NT. The authors could re-write these sentences about what happens in the Bellingshausen/Amundsen Sea more clearly and perhaps also include in the advance-retreat cycling which has been mentioned and investigated by various authors for this region in particular (Stammerjohn is one candidate here, Harangozo another one).

L415-427: Of course, for completeness, these two sectors need to be discusses as well. However, the reviewer feels that what is written for these two regions could be condensed even more because i) the patterns are really very similar between NT and BT and ii) pack ice trends are generally positive, more in BT than NT, right, and iii) trends in MIZ extent basically vary around zero with exceptions during September through December in both regions and both algorithms. The reviewer found it hard to get into discussion of significance and the relation to SIE change here.

L426-427: This notion is not reflected by Figure 9. The reviewer again suggests to skip the ice category "polynya" in this figure.

L440-470 / Figure 10: - The images in Figure 10 could be re-ordered such that the topmost is MAM, then JJA and then SON - motivated by how the season progresses and motivated by the authors' order of describing the figure. - In JJA it is hard to see these trends and the fact that they oppose each other in Figure 10; maybe these are not that relevant? - in L451: The largest MIZ regime increase is 10 km / decade for NT ... that's not even half a grid cell ... and just between 1 and 1.5 grid cells in total over the 35-year period. - in L453: Perhaps referring to Figure 7 would be a good idea to underline this notion? - in L455: The authors could add that they are still referring to NT here. - in L467: This is an interesting observation and the reviewer is curious to see whether this weaker correlation has found its way into the discussion - it might be an effect of the larger sensitivity of NT to snow property variations particularly during late-winter / spring.

L472-481: This introduction into the seabird topic underlines my comment given with respect to Figure 5 that ice category "possible polynyas" is perhaps not that relevant for this study.

L502-504/ Table 4: The reviewer is a bit surprized that the number of chicks counted in Dumont D'Urville, which is situated at longitude 140degE can serve as a response variable for a region which in 5 of the 6 months taken starts at least 20 deg further West and which extents up to a quarter way around the continent towards the West. Is that site (Dumont D'Urville) really representative for the geographical region chosen? Why didn't the authors chose a region which is kind of centered at the chick collection site? Perhaps the authors could comment on that.

L507: What is "AIC"?

L513-526: - What is given at the x-axis of Figure 11. What is "pack ice bootstrap"? Are we looking at an anomaly? - The text (here and in the previous paragraph) speaks about AIC difference < 2 or > 2. Difference to what? What do the AIC values in Table 5 tell me? - The NT MIZ AIC value isn't that far away from BT pack ice AIC; why NT MIZ is not supported then? - The authors could use the same number of digits for all AIC values ... but could ask themselves how accurately AIC can be determined ... 1/100 or 1/10? - So what the authors wish to tell the reader here, in short, is that using BT pack ice fits best in relation to breeding success while using NT MIZ would have led to a wrong conclusion. NT pack ice and BT MIZ would have given in-different results. While the reviewer kind of likes the result (because it is known that BT provides more realistic sea ice concentration in the Southern Ocean than NT) the reviewer is wondering about i) how sensitive this results is with regard to application of a different model and ii) how the results would look like if the threshold sea ice concentration used to delineate MIZ and pack ice would have been 70% or even 60%?

L528-551: While what the authors write in these two paragraphs is for sure right the reviewer has difficulties to relate this to the main topic of the present paper - and in

particular to the link with biology.

L552-566: Here the authors try to establish the link to their paper. Suggestions are: - to clearly differentiate between sea ice extent and sea ice area. The latter includes sea ice concentration information and therefore is much better suited than sea ice extent to investigate changes and variability of the nature of a sea ice cover. - in L557: There are other studies which could be mentioned here, which focus much better on polynyas and their temporal development and/or associated ice production, e.g. Tamura et al., 2008; Kern, 2009; Nihashi and Ohshima, 2015 - in L564-566: Holland and Kwok, 2012, would give enlightening information.

L581-588: The reviewer suggests the authors make the notion that the BT used in the Arctic is not the same as is used in the Antarctic - nut just in terms of the tie points but in terms of the used frequency combination. The results are therefore not compatible 1-to-1.

L594-595: Such a statement might need a reference. Isn't also most of the Antarctic sea ice snow covered?

L595-600: As with regard to tie point selection the reviewer recommends to perhaps take another look into relevant literature. While it may be correct that NT uses predefined tie points it is also correct that the BT needs to define tie points which is done based on the actual observations, yes, but except in a very few cases or when applying the BT regionally as e.g. in the Southern Ross Sea where thin ice exported from the Ross Ice Shelf dominates the scene, it is likely that the sea ice is snow covered as well. In particular the sentence in L597-600 is then perhaps a sub-optimal statement. - The reviewer is wondering whether the authors can give a reference for the "seasonal variations in emissivity can be very large"?

L601-619: Finally the authors come up with at least one of the key papers of results from intercomparison of NT to BT sea ice concentrations in the Antarctic (e.g. Comiso et al., 1997). One could also take a glimpse into Comiso and Steffen, JGR-C, 2001.

One uncertainty factor for sea ice concentration retrieval is indeed the snow cover influence for which it is stated in the literature that the vertically stratified snow cover complicates sea ice concentration retrieval by the NT due to the stronger sensitivity of the 19 GHz channels to these effects - which is the main reason for the occasionally observed strong underestimation of the sea ice concentration by NT over pack ice compared to BT (Comiso and Steffen, 2001) and which is the main reason why the enhanced NT algorithm was developed (Markus and Cavalieri, 2000). The second factor is the different influence of thin ice on the sea ice concentration. Thin ice creates a negative bias in the sea ice concentration obtained as is illustrated, e.g. in Ivanova et al., 2015, and in Shokr and Kaleschke, 2012. On top of these the authors could mention other issues like gap layers, ice-snow interface flooding, formation of meteoric ice, snow metamorphism which all may or may not have an influence on the sea ice concentration which - to the reviewers' knowledge - nobody as quantified yet for Antarctic sea ice. - L606-610: This excursion to the Arctic seems confusing and could be deleted.

-L613-614: While one could agree to this notion one could also make the point that the definition of the MIZ via the sea ice concentration can be a very vague estimate. It can well be that even with 100reds of contemporary scenes with high-resolution optical imagery or SAR images one will not be able to better "validate" where the MIZ stops and pack ice starts. The reviewer has rather the feeling it is a matter of definition - and sea ice concentration might not be the ideal means for this.

References: Parkinson et al., 2012, in L53 not in refs. Bintanja et al., in L72 has year 2012 in text but 2013 in refs. Kohout et al. 2014, in L81 not in refs. Ferrari 2014 in L107 is Ferrari et al. 2014 in refs. Ivanova et al., 2015 in L113 is now in The Cryosphere Loeb et al. in L488 is 1993 in text but 1997 in refs. Polvani and Smith, 2013 in L534 not in refs. Hobbs et al., 2015, in L546 not in refs.

Ivanova et al. appears as Ivanova and others in the refs. Steig et al. appears as Steig et al. in the refs.

Kohout and Meylan, 2008 not used in text. Louzao et al. 2011 not used in text.

Typos: L114: "mattes" –> "matters"

---

## Referee Comment (RC3) · Anonymous Referee #3 · 19 Apr 2016

The manuscript examines the variability and trends of Antarctic sea ice in three categories: pack ice, marginal ice zone and coastal polynya derived by Bootstrap and NASA Team algorithms. Authors show that the differences in trends and variability between these two datasets are quite large for pack ice and marginal ice, even though the differences in the total extents are relative small. The details within the ice pack is quite essential for the atmosphere-ocean-sea ice interaction, as well as biological studies, such as the study on snow petrel presented here. The manuscript reveals an important fact that satellite observed sea ice concentration can contain large errors and biases, and should be used with caution. I think that the study contains valuable information for polar community. However, I see some places need to be clarified. In general, the

manuscript is rather tedious. Authors should reduce the discussions on insignificant results and focus on important and significant results. It is worth to be published after a minor revision.

The research represents an original effort to evaluate SIC quality by comparing two datasets in marginal ice zone and pack ice. It fits well the scope of TC. The methodology is sound and conclusions are sufficiently supported by analyses. However, the title is not quite accurate. The study cannot conclude on the trend and variability of MIZ since the discrepancies between two data products are quite large. The manuscript focuses on the discrepancies in MIZ, pack ice and coastal polynyas between data products. The title should reflect on that.

Technic/editorial issues.

Figure 3. It would be helpful to add longitude lines to separate the different regions. In addition, bird study areas also need to be marked in this figure.

Line 306-317. The figure 7 needs some clarification. Is the expansion and contracting of outer ice edge relative to a zonal mean? If it is the case, what is the zonal mean of MIZ, in consideration of that the zonal mean pack ice is the mean of 85% and outer edge is the mean of 15%?

Line342. Insert "area of the" or "extent of the" before pack ice.

Line 371, "While the sign of the Ross Sea sector trends from"

Figure 9 need some work. It would be very helpful if authors add contours of a confidence level in figure 9 so readers can see where and when significant trends occur. All x-axis and y-axis labels need to be enlarged, so they are visible.

Page 10, There are many detailed discussions and the key results don't stand out. Authors should focus on the significant trends and their implications.

Page 11, section 3.2.3. Authors need to explain how figure 10 is calculated. It should

be seasonal mean and zonal mean, right? In addition, it would be nice to comment on whether the results derived from the width are consistent with or different from the results from ice extent/area presented in the early section.

Line 507. Please define "AIC".

Page 13, Discussion. It is very clear that SIC derived from these two algorithms has large differences within the ice edge. To readers who use SIC for various studies, it is very important to know which product is more suitable for their need. It would be useful if authors could comment on whether there are other independent verifications? The snow petrel study is an excellent example. However, it is only related to the packed ice.

---

## Author Comment (AC1) · 17 May 2016

Response to reviewers' comments We are grateful to all reviewers for their thoughtful comments, which have strengthened our manuscript. Below we detail our responses to the reviewers' comments and changes made to the manuscript. Our responses are italicized and highlighted.

Reviewer #1 Interactive comment on "Mapping and Assessing Variability in the Antarctic Marginal Ice Zone, the Pack Ice and Coastal Polynyas" by Julienne C. Stroeve et al. H. Flores (Referee) Hauke.Flores@awi.de General comments Julienne Stroeve and her co-authors compare the results of two popular algorithms using passive microwave satellite data to classify ice types in the

Antarctic sea ice zone, the NASA Team and the Bootsrap. They show that sea ice extent estimates are largely consistent between the two algorithms. They differ, however, in the proportion of consolidated pack-ice versus marginal ice zone (MIZ) and polynias, with greatest discrepancies in the contribution of the MIZ. When applied to biological datasets, in this case the breeding success of snow petrels Pagodroma nivea, however, these discrepancies can lead to opposite conclusions. This manuscript presents a highly desirable critical cross-validation of satellite-derived data. In particular, the inclusion of a biological dataset adds high value to both physical and biological communities, highlighting the importance of such exercises in this under-studied cross section of disciplines. Furthermore, it was a pleasure reading this manuscript, since it manages to present a complex topic in an easily understandable language, even to non-physicists.

We thank the reviewer for their positive comments.

Having said this, there are a few things that should be improved: 1) The results and discussion section are not well separated. To preserve the excellent flow of the manuscript, I recommend to merge them into a "Results and Discussion" section. If this is not possible, speculations and literature references should be consequently moved from the results section to the discussion.

Given the length of the results and discussion it seems best to keep the Results and Discussion sections separated. Thus, we moved most of the speculations and literature references into the discussion section.

2) In my view, the greatest weakness of the approach chosen is the definition of the 'polynia' ice type. Just using a proportional ice coverage of 0.8 near the coast as a criterion may incur confusion with other more open ice situations. It may well be that the two algorithms 'see' completely different things, and therefore result in a large difference in the seasonal occurrence of 'polynas'.

Indeed, the notion of an 80% SIC threshold is somewhat arbitrary. In Li et al. (2016) we found that to match NASA Team and Bootstrap for polynya area, we needed to use a 75% SIC threshold for NASA Team and 85% for Bootstrap. This highlights the sensitivity of the algorithms. While we find good spatial distribution with polynyas reported in previous studies (e.g. Massom et al., 1998, Arrigo and van Dijken, 2003, Kern 2009, Arrigo et al., 2015) we are really focused on the initial stage of polynya development (before adjacent polynyas coalesce into a larger one). We have now specifically stated in the methods section:

We have previously tested mapping polynyas using a SIC threshold of 0.75 and 0.85 for the NASA Team and Bootstrap algorithms, respectively, and found that these thresholds provided consistent polynya areas between the two algorithms and matched other estimates of the spatial distribution of polynyas [see Li et al., 2016]. However, for this study we chose just one threshold, a compromise between the two algorithms, so that we can better determine the sensitivity of using the same threshold on polynya area and timing of formation.

3) There is little information provided on the statistics of GLMs looking at the breeding success of snow petrels. A convincing statistical approach, including model selection, is the fundament of any conclusions concerning the seemingly opposite outcomes of the two algorithms. We have added more to this section. We appreciate the reviewer finding this an important component of the paper. The concept of strength of evidence seems almost "new" in the life and social sciences. Traditional methods have focused on "testing" null hypotheses based on test statistics and their associated P values. From the P value comes an arbitrary judgment concerning "statistical significance" and dichotomous ruling about the rejection of, or failure to reject, the null hypothesis. For several reasons, P values do not constitute a basis for formal evidence (see Royall 1997). The new I-T methods are not a test in any sense; rather they represent a very different methodology for empirical science. The I-T methods provide a formal, fundamentally sound, approach of developing an a priori set of hypotheses and then a quantification of the data-based evidence for, and ranking of, each hypothesis. This is followed by interpretation of the results in the face of model selection uncertainty and this is one aspect of multimodel inference."

Specific comments Introduction Ll 44-66: When addressing regional variability, the strong decline in the WAP should be mentioned with respect to the final objective to discuss ecological implications. Done and we now reference Ducklow et al. 2012 and Smith and Stammerjohn 2001 about sea ice declines in the WAP.

Also, previous declines in Antarctic SIE should be mentioned. For example, Flores et al. (2012) note that "This growth, however, has so far not compensated for a decline of the average sea ice coverage between 1973 and 1977, which accounted for âĹij2 × 106 km2 (Cavalieri et al. 2003, Parkinson 2004). Reconstruct ions of the position of the ice edge in the pre-satellite era give strong evidence that the overall areal sea ice coverage in the Southern Ocean declined considerably during the second half of the 20th century (Turner et al. 2009a)". Done

M&M Ll151-152: Could be problematic with respect to ecological interpretations, because it also includes areas with loser pack ice and wakes behind icebergs that are not polynias and thus do not necessarily feature the same biological dynamics. Also: What is "near" the coast precisely? We agree that we are not necessarily resolving polynyas here with the coarse resolution satellite data. However, since our interest is open water areas near the coast that can be used by bird species for foraging, we feel that this metric is still important. We now make sure to say potential coastal polynya.

Results Ll 413-526: More details on the model statistics are needed. Were the slope terms and intercepts significant? Using differences in AIC alone for model selection can be tricky. I recommend testing the 'better' model against the 'next worse' using ANOVA or F statistics, depending on the model applied. It would be useful to see the model validation plots in the supplementary material. To me it is not clear why the model using pack ice and BT is preferred. For the reviewer concern, we think it is a statistical debate. We applied a state of the art approach that is commonly used in ecology. We try to make that point in the revision. However, the reviewer's comments seem to somewhat contradict each other. In the main comment it's stated " A convincing statistical approach, including model selection, is the fundament of any conclusions concerning the seemingly opposite outcomes of the two algorithms "which is what we did. But then "I recommend testing the 'better' model against the 'next worse' using ANOVA or F statistics, depending on the model applied", which is not the best approach for model selection.

We have added more details on the statistical model used. We applied the information-theoretic approach to valid inference that "replace the usual t tests and ANOVA tables that are so inferentially limited" (quote from the statistician Burnham et al. 2011 Behav Ecol Sociobiol (2011) 65:23–35 DOI 10.1007/s00265-010-1029-6). This is based on quantitative measures of the strength of evidence for each hypothesis (Hi) rather than on "testing" null hypotheses based on test statistics and their associated P values. Here, to quantify the strength of evidence for each hypothesis (Hi) , we used the common criteria AIC= - _2 log(L))+2K. The term -_2 log(L)) is the "deviance" of the model and log(L) the maximized log-likelihood. We select the model with the smallest AIC value as "best" in the sense of minimizing Kullback–Leibler (K-L) information loss, i.e. provide the highest strength of evidence for the given hypothesis. The section now reads as:

Effects of MIZ and pack ice area were analyzed using Generalized Linear Models (GLM) with logit-link functions and binomial errors fitted in R using the package glm. Specifically, the response variable is the number of chicks $C_t$ in a breeding season t, from 1979 to 2014 collected at Terre Adelie, Dumont D'Urville [Barbraud and Weimerskirch, 2001, Jenouvrier et al., 2005]. It follows a binomial distribution, such that $C_t \sim$ Bin(ïA■t,Nt), where Nt is the number of breeding pairs and ïA■t is the breeding success in year t. The breeding success is a function of the MIZ and pack ice covariates at time t (COV) such as: ïA■t = ïĄć0 + ïĄć1 COV(t)

To select the covariate that most impacts the breeding success of snow petrels, we applied the information-theoretic (I-T) approaches [Burnham et al., 2011]. This is based on quantitative measures of the strength of evidence for each hypothesis (Hi) rather than on "testing" null hypotheses based on test statistics and their associated P values. To quantify the strength of evidence for each hypothesis (Hi) – here the effect of each covariate on the breeding success- we used the common criteria AIC (the Akaike's Information Criteria), where AIC = - 2 log(L) + 2K [Akaike, 1973]. The term, -2 log(L), is the "deviance" of the model, with log(L) the maximized log-likelihood and K the total number of estimable parameters in the model. The chosen model being the one that minimizes the AIC, and the ability of two models to describe the data was assumed to be "not different" if the difference in their AIC was < 2 [Burnham and Anderson, 2002]. While non-linear models may be more appropriate as ecological system relationships are likely more complex than linear relationships, without a priori knowledge of the mechanisms that could lead to such non-linear relationships, it is extremely difficult to set meaningful hypothesis to be included in the model selection. Discussion L619: Another way to validate algorithms would be the ASPEcT ship observation data. Yes that could be useful and something to consider for future work.

Conclusions Ll643-646: I have the feeling that the results of the polynia estimation are blurry in both algorithms, resulting in this large variability of timing of polynya maxima. This may be due to inaccurate definition of polynyas in the analysis. We agree that precise polynya delineations are problematic at this resolution, but as we previously state, our interest is really in potential open water near the coast. We added another sentence in the conclusions to highlight this: While we do not precisely resolve polynyas, these potential coastal polynyas (i.e. open water areas near the coast) are important foraging sites for sea birds.

Technical corrections Abstract L14: replace "biological" with "biologically" Done

Introduction Ll 40-43: Split this sentence in two. Done

L59: I believe it should say "maxima" Corrected

L94: Did you mean "continuously"? Corrected

L114: replace "mattes" by "matters" Done

Results L197: Better say "Results and discussion"? We felt it best to keep them separate

L215: delete "(e.g. the shading)" Done

L373: replace "significantly" by "significant Done

L375; delete "(e.g. spring)" Done

Figure 5: What is the variability measure here indicated by the shaded areas around the curves? This is the standard deviation from the long-term mean. We now state that in the figure caption.

Reviewer #2: Stefan Kern Summary: Sea ice concentration data sets obtained for 1979-2014 with two different al- gorithms (NASA Team and Comiso Bootstrap) from satellite microwave radiometry are used to investigate the spatio temporal evolution of a number of different ice regimes. These regimes are defined by sea ice concentration thresholds and by their position relative to the coast and/or open water along radial transects through the ice cover. Classified maps are subsequently used to compute the circum-Antarctic and regional extent of the ice regimes selected. The focus is laid onto ice regimes "marginal ice zone (MIZ)", "pack ice" and "possibly coastal polynya". The spatio-temporal distribu- tion of these ice regimes is investigated and discussed with focus on the difference between both algorithms and on trends (and their significance) over the period investigated (1979-2014). At the end the paper tries to link the satellite observations with biological observations.

General comments: The reviewer is allover in line with the general content and also the main message of the paper. There are several issues, however, which the authors could improve and/or decide upon to substantially improve the paper. Points 1), 2) and 5) are those which made the reviewer to decide to rate rigour and impact as "fair" 1) One of the two algorithms used is - to the opinion of the reviewer - well known to provide more reliable results for Antarctic sea ice. The authors tried to avoid to state this upfront in their paper. Also in the discussion of their results the reviewer feels that the authors could have elaborated more on the known differences between the two algorithms and on their, partly known, reasons. The reviewer feels that this would come closer to the "Assessing Variability" part stated in the title of the paper. We thank the reviewer for their very in depth review. We agree with the reviewer that the Bootstrap algorithm has been more systematically used for Antarctic studies. However, we find in the literature that the NASA Team algorithm is also used in many studies, even in a recent study by myself and colleagues, as we wanted to use the most up-to-date data set and the NASA Team algorithm is produced in near-real-time. We have purposefully steered away of recommending one algorithm over another as we feel that in order to do so requires more validation studies, which we plan to do using visible imagery this year. We already spend three paragraphs on the known differences between the algorithms and impacts on the ice concentrations. We feel that this is sufficient given the lack of validation of each data set in the Antarctic. However, we add yet another reference from observations in the Weddell Sea that suggest sea ice concentrations are likely closer to the Bootstrap algorithm than the NASA team, though these were made in the interior of the ice pack. An in depth validation effort is outside the scope of this paper but will be performed this summer and submitted to a remote sensing journal.

2) The reviewer feels that the explanation of the methodology, its sensitivity to chosen parameter values, its limitations, and the relevance of the results could be improved: a) Some specific information about the methodology is missing (as is mentioned in the specific comments. b) An investigation about the sensitivity of the chosen threshold to separate MIZ from pack ice is missing. The authors might ask themselves how their results would look like if they would have used different sea ice concentration thresholds. c) The inclusion of ice regime "possibly coastal polynya" into the analysis seems not to be straightforward. The main reason for this is the fact that many polynyas are sub-grid scale features and therefore produced somewhat noisy signals in the analysis. Instead, as is suggested by Figure 5, the ice regime "broken ice in the pack ice"

might deserve more attention because this is on average the third-largest fraction the ice regimes chosen occupy in the NT data - at least with the chosen threshold of 80% sea ice concentration. In short: I would suggest to skip "possibly coastal polynya" and include "broken ice". d) As with regard to the relevance it would have been good to see how relevant results are. A trend in a regime of 10 km / decade manifests itself in 1-1.5 grid cells over the entire time period looked at. How sure are the authors that their results are solid? The choice of the MIZ threshold was based on the paper by Strong and Rigor for the Arctic, and this was validated with National Ice Center Charts. Thus, we chose to remain consistent with the definition of the MIZ used in that study. As mentioned above, we do plan a validation this summer with visible imagery. We agree with the reviewer that the trends in the MIZ are rather small (1-1.5 grid cells) when averaged over the entire SH sea ice, but regionally there are larger trends. The reason we used possible coastal polynya as we have already published a paper in GRL this year using the NASA Team and Bootstrap data for mapping polynyas and comparing with the timing of phytoplankton blooms. We agree with the reviewer that polynyas are generally sub-grid scale features and not straightforward to detect with the coarse resolution passive microwave data. However, it is not our intent to accurately map polynyas at this resolution but rather areas where they may exist and therefore areas that are biologically active regions. In this use, and the reason we use the word, 'possible coastal polynyas'is for the biological relevance and the fact that birds can forage in these open water areas. We make this clear now in the methods section and state: We have previously tested mapping polynyas using a SIC threshold of 0.75 and 0.85 for the NASA Team and Bootstrap algorithms, respectively, and found that these thresholds provided consistent polynya areas between the two algorithms and matched other estimates of the spatial distribution of polynyas [see Li et al., 2016]. However, for this study we chose just one threshold, a compromise between the two algorithms, so that we can better determine the sensitivity of using the same threshold on polynya area and timing of formation.

3) The reviewer was confused a bit by the 3 or 4 different over-arching names the authors used for MIZ, pack ice, etc. Harmonizing these and not using "ice type" would certainly improve readability of the paper. We are not entirely sure what the reviewer means by different names for the MIZ and pack ice, but we have removed the use of ice types to address the reviewers concern. When instead say ice category.

4) The text with regard to interpreting Figure 9 seemed a bit long and confused sometimes. Here the reviewer feels that focussing more on the easily to identify features would help the reader. Also, as detailed in the specific comments, a shift of the starting point from 0degW to 60degW would enhance interpretation of the figure. The authors might also want to think about difference maps between trends obtained for NT and BT. We simplified the discussion but chose not to show difference maps as they are harder to interpret.

5) While the reviewer applauds the inclusion of the biology at this stage as an external means to assess the satellite observations, the reviewer feels that a more detailed description is required to fully understand and evaluate the results presented in Figure 11 and Table 5. In the discussion the paper focusses more on climatic issues and potential linkages instead of discussing the potential limitations of the approach (see 2) and the added value included by the inter-disciplinarity. If this paper aims to advertise that through interdisciplinarity value is added to such kind of an analysis of a geophysical data set, then it has failed because it is neither visible from the title nor is this topic given enough weight, in comparison to the climate issues, in the discussion, and the future potential of using such inter-disciplinary approaches is also not discussed further. This part does not really seem to be integrated into the paper. We have attempted to better integrate the biology aspect of the paper. We have also changed the title to: Mapping and Assessing Variability in the Antarctic Marginal Ice Zone, the Pack Ice and Coastal Polynyas in two Sea Ice Algorithms with implications on Breeding Success of Snow Petrels. We have additionally included more information on the biological method that we hope allows readers to fully understand the results presented in Figure 11 and Table 5.

Specific comments: Abstract: First impression is that > 50% of the abstract do not reflect results of this study. Perhaps the sentence "Knowledge of the ... contraction in others." could be replaced by a sentence describing the methodology used in the paper - which is currently missing in the abstract. We slightly tweaked a sentence in the abstract to include the methods used: This study uses two popular passive microwave sea ice algorithms, the NASA Team and Bootstrap, and applies thresholds to the sea ice concentrations to evaluate the distribution and variability in the MIZ, the consolidated pack ice and coastal polynyas. We also added a bit more of the results into the abstract: Results reveal that the seasonal cycle in the MIZ and pack ice is generally similar between both algorithms, yet the NASA Team algorithm has on average twice the MIZ and half the consolidated pack ice area as the Bootstrap algorithm. Trends also differ, with the Bootstrap algorithm suggesting statistically significant trends towards increased pack ice area and no significant trends in the MIZ. The NASA Team algorithm on the other hand indicates significant positive trends in the MIZ during spring (September, October and November). Potential coastal polynya area is also larger in the NASA Team algorithm, and the timing of maximum polynya area may differ by as much as 5 months between algorithms. These differences lead to different relationships between sea ice characteristics and biological processes, as illustrated here with the breeding success of an Antarctic seabird.

L49-51: In this context, the authors might want to refer to the work of Reid et al., and Simmonds, both in Annals of Glaciology, 56(69), 2015. Done

L67-75: The reviewer is wondering whether the authors might want to include also the work of Holland and Kwok, Nature Geoscience, 5, 2012, which nicely suggests the different effects and regions of predominantly dynamic control and thermo-dynamic control. Holland, The seasonality of Antarctic sea ice trends, Geophys. Res. Lett., 41, 2014, might also be a paper to take a look at and to include here. Done

L81: The reviewer suggests to refer also to the book edited by M.O. Jeffries: Antarctic Sea Ice, Physical processes, interaction and variability, Antarctic Research Series, 74,

AGU, 1998. Thank you for pointing out this reference, we were unaware of this book.

L82: The authors write of a "dynamic MIZ". Is there also an "non-dynamic MIZ"? Comment: The authors define the MIZ basically via the dynamic processes and in a way refer to the action of waves to fracture the sea ice and generate smaller floes. While on can imagine that the width of the MIZ can be defined during on-ice wind or swell events by the penetration of swell into the sea ice The reviewer is wondering how one defines the MIZ during the other times, i.e. when there is no on-ice wind of swell, the pressure is released and the sea ice is following perhaps a divergent motion with a lot of openings which are refreezing. How does on define the MIZ here? This is discussed further in the methods section. As we state we chose the threshold based on previous studies of mapping the MIZ in the Arctic.

L83: "longer and larger ice-free summer". What is meant by "larger ice free"? Longer refers to the time, and larger referred to spatial area. We found the sentence was not really needed so we removed it.

L84-94: What the reviewer is missing here is the special character of the MIZ in the Antarctic during sea ice growth / ice edge advance; the suggestion is to get back to the "good old pancake ice cycle" notation of Lange et al. 1989, Annals of Glaciology, 12, 92-96 and Lange and Eicken, 1991, Annals of Glaciology, 15, 210-215. Even though these are old papers there is recent evidence that this is still valid: e.g. Ozsoy-Cicek et al., Deep Sea Research part II, 58(9-10). We are not entirely sure what the reviewer is looking for. The aim of the paper is not to go into detailed processes of the formation of the MIZ (e.g. pancake ice formation) but rather give a brief description of the MIZ and its importance for biological activities. Since the paper is already rather long, to go into the detail requested by the reviewer will only add to the length to the paper but also likely confuse readers as to what the study is really about, and that is to compare two different sea ice algorithms and interpret how the use of the different algorithms would impact uses of the data for biological applications.

[Figure]

L89: "ocean waves define ..." Is a repetition of L82. Fixed

L92-96: The polynya part comes a "bit naked" with regard to citations. Suggestion: Morales-Maqueda et al., 2004, Review of Geophysics, 42, and later on, when it comes to the role of ice production perhaps: Drucker et al., Geophys. Res. Lett., 38, 2011. Done

L119: At the end of this line one could add the reference of Comiso et al., 1997, Remote Sensing of Environment, 12 and of Comiso and Steffen, J. Geophys. Res. 106(C12), which both nicely illustrate to pros and cons of the two algorithms the authors are using in their study. We feel those references are a bit out of place in that sentence but we refer to them in the discussion.

L126: The reviewer is not too happy with the term "ice type" as the authors use it here. Ice types are given in the WMO nomenclature of sea ice and are terms such as thin first-yer ice, shuga, nilas, grey ice, pancake ice, etc. What the authors define here in their work can perhaps be termed better by "sea ice regime" or "ice category" - a name the authors use for instance in L180 anyways, while in L187/188 the authors name it "ice classes". Perhaps sticking with one term would be a good idea. In case the authors decide to stay with "ice type" then the suggestion is to state clearly here that they are not referring to ice types in the classical sense but use the term "ice type" to differentiate between MIZ, pack ice, and polynya - and even more "broken ice in the pack". We now say ice category if we do not specifically refer to the MIZ, pack ice or polynyas.

L138/139: Here it might help to mention that the NSIDC has combined these two algorithm to build the first NSIDC sea ice concentration CDR (Peng et al., 2013, Earth System Science Data, 5) Good idea, now mentioned.

L145: "heavily influenced" Could this statement be precised? We feel this is not necessary

L148: The authors could note here that presumably this definition of the border be-
tween MIZ and pack ice is not connected to the other, more dynamic definition of the
MIZ via the penetration depth of waves into the sea ice. Is that correct? It is difficult
to determine precisely how far waves penetrate without doing wave analysis as well,
which is outside the scope of this paper. Our definition of the MIZ follows that from
Arctic studies and should reflect most of the wave/ice interactions.

L152: Work carried out previously used different sea ice concentration tresholds: e.g.
C5 70%, e.g. Parmiggiani et al., 2006, International J. Rem. Sens. 27(12) or Massom
et al. 1998, Annals of Glaciology, 27, 420-426. The authors could comment on their
choice of using 80%. As we stated in the manuscript, we followed the analysis by
Strong and Rigor, thresholds that were previously validated in the Arctic. We feel this
is justified for this paper without additional validation with other satellite data.

L153: What is the longitudinal sampling of the radial transects? Or in other words:
How many transects did the authors use? The authors could perhaps also shed more
light on how the transects were placed. The data are on a polar-stereographic grid.
Therefore radial transects which have, e.g. a 2-pixel spacing at 55 deg South may
overlap the same pixel further pole-ward. How is this treated in the algorithm? Is it
correct to say that the automatic classification scheme is simply kind of convering a
daily sea ice concentration map into a binary map where specific (which?) values are
assigned to pixels of the respective ice category. In other words, the range of 0 ... 100
is condensed to 5 values (according to Figure 1 and Table 1). For the classification
of the daily 25 km grid cells, transects (stepping north to south) was performed at 1/2
degree steps. Yes, there is some noise in the edge on a daily basis. Some missing
data will filled in by values from adjacent days. The classification scheme converts
the raw concentration data into 5 classes: MIZ, PACK, Polynya, Ocean and Interior
broken sea ice within the pack ice region. But it is using the relative positions of open
water, MIZ and Pack ice to lead to some understanding of the time variation of sea
ice. Ultimately one has a 3 dimensional array of sea ice concentrations (lat, lon, day).

We are classifying it to better understand what is happening as the seasons and years change. Other classifications (for instance different thresholds or different geometry) might tease out other general features of the changes. We can only say that this classification improved our understanding of sea ice changes. In this paper we are trying to communicate the general features we found.

For the area average seasonal cycles and trends the actual area of each grid cell was used. This smoothed out the rough edges. If one looks at figure 8, the daily annual cycle of trends, some noise is leaking through. Similarly some of the small scale bumps between different longitudes in figure 7 are not important.

Hence there is no computation involved - in contrast to what is stated further down in L189. Is Table 1 really needed. Will the information given therein used later in the paper? We prefer to keep Table 1 in as it also shows the SIC thresholds for broken ice and open ice in the pack ice.

L172: Polynyas may also form offshore of a fast ice region. This occurs for instance in the Eastern Weddell Sea or the Indian Ocean sector. Are such areas counted under category "broken ice inside pack ice" or "inner open water"? What about the Cosmonaut polynya developing in the Indian Ocean sector? We are really focused on the polynyas near the coast, and depending on how far these polynyas are from the coast they may or may not be captured by this method. Depends on whether or not they are captured in the coarse resolution of the satellite pixel. However, given the coarse resolution and the SIC thresholds we use, they are likely captured.

There are regions like closely east of the Antarctic Peninsula and in the Western Ross Sea where the radial transects do, when coming from the north, first transect sea ice / open water, then land, and then again sea ice / open water. How are these transects treated? For the limited range of longitudes that experience this problem, we stepped from the north to the first ice edge. If you look carefully at all the data, there are a few anomalous results near the peninsula and some areas with islands. There are 1%

errors in the classification. That is in keeping with the SSMI concentration accuracy.

L173/Figure 2: Is there a change to instead list the colors in the caption annotate these in form of small boxes in or underneath the block of images? This way the authors dould avoid potential misinterpretation of the Figure by the individual way people allocate actual colors to the name given plus by the way printers interprete colors. Done

L174: Suggest to give the actual dates instead of the day of the year. Done

L189: How did the authors compute gridded fields and regional averages from binary, i.e. classified maps? The reviewer's guess is, the gridded field is not computed but simply obtained by the classification process (see comment to L153). Subsequently, the classified gridded fields are used to compute the areas covered by the different ice regimes - for the entire southern ocean and the 5 regions - on daily temporal scale. The resulting time series of ice regime areas is then used to compute monthly mean ice regime areas. Is this correct? The reviewer had diffulties to understand this the way it is currently written and found myself thinking about how - in one grid cell - a temporal change between different ice regimes is treated. Which grid cell area file did the authors use to compute the ice regime areas? Since the authors did not use data gridded on the EASE grid, the grid-cell area at latitudes different from 70 deg South differ from 625 kmËĘ2. You are correct that we provide binary gridded fields at 25 km spatial resolution with a flag for different ice categories (as seen in Figure 2). We use the area per pixel to compute the true extent of each ice category. Since the reviewer found our description confusing we made the following changes: Using the binary classification scheme, daily gridded fields at each 25 km pixel are obtained. Using this gridded data set we then obtain regional averages for five different regions as defined previously by Parkinson and Cavalieri [2012]. These regions are shown in Figure 3 for reference. Climatological mean daily and monthly time-series spanning 1981 to 2010 are computed for each of the five sub-regions, as well as the entire circumpolar region, and for each ice classification together with the +/- one standard deviation (1ïĄş). Monthly trends over the entire time-series are computed by first averaging the daily fields into monthly

Interactive
comment
values and then using a standard linear least squares, with statistical significance evaluated at the 90th, 95th and 99th percentiles using a student t-test.

L190: Figure 3 shows five instead of six regions. A typo? Corrected

L191 / Figure 3: Does figure 3 on purpose omit showing the borders between the regions? At least in my print out there are no border lines. We have added the border between regions

L192: Can the authors comment on the time period chosen? Why, if the sea ice concentration data sets are available from October 1978 until today they choose a shorter period? Sea ice concentrations based on SMMR, i.e. 1981-07/1987, are only observed every other day. Has this been taken into account in the averaging process? At the time we did the study, the Bootstrap data was only available through 2013 and yes we take into consideration that SMMR is only observed every other day. We updated results through 2014 as the Bootstrap data set was updated.

Figure 4 and its discussion: Figure 4 seems not to using the space allocated for it efficiently. Suggestions: i) Start at a southern latitude of 80 degS or even 75 degS; ii) include the variability of the ice edge location either by using bars showing +/- on standard deviation or by a shading similar to Figure 5. The reader might be interested to see whether one algorithm varies more than the other one. Even though both algorithms seem very similar one can see a southward displacement of the Bootstrap ice egde during summer and a northward displacement of the Bootstrap ice edge during winter. The authors could detail a bit more how large this displacement is ... and perhaps write that this is of the order of one 25 km grid cell (which is how it looks like). The errors are small for Figure 4 because this is an average over 35 years (we have now added error bars on the figure). The idea was to show that NASA Team SIE = Bootstrap SIE. Details like the MIZ and polynya are different because the concentrations are different, especially around the 80% SIC threshold. We do now mention the small southward displacement of the Bootstrap ice edge in summer but the northward displacement overlaps with the error bars. The authors state that the large emissivity difference between open water and sea ice drives the location of the ice edge. This is partly true only. Another reason why both algorithms show such close agreement in the ice edge location could be the application of a (the same?) weather filter which is known to cut off low ice concentrations [Ivanova et al., 2015]. The weather filters are different between the NASA Team and Bootstrap algorithms and thus this is not a factor here.

L206: The authors speak about an extent, i.e. the sum of all grid cell areas within one ice categorie without any weighing carried out with the actual sea ice concentration, while the y-axis in Figure 5 denotes the quantity given as "Area". Would it make sense to, in order to avoid misunderstandings with the classical sea ice extent and sea ice area, also annotate the y-axes with "Extent"? We can see the reviewers point about a confusion between the extent vs. area used with sea ice concentrations. However, extent in the context of sea ice is the outer boundary of the ice cover. The area of the holes are not subtracted out. In Figure 5 for example, the areas account for different interior ice types.

Another comment to Figure 5 - later in the paper it becomes even more clear that NT and BT are different particularly in terms of ice categories MIZ and pack ice. Not too much could be said about the ice category "possibly polynya" which results seem to be quite noise and also difficult to interpret given that these tend to be a sub-grid scale phenomenon during winter and only seem to become important during November/December. Here, in Figure 5 the 3rd largest difference between NT and BT is not observed for ice category "possibly polynya" but for "broken ice inside pack". Did the authors carried out similar analysis with this ice category like they did with "possibly polynya"? I could imagine that considering and looking at ice category "broken ice inside pack" could be more enlightening than "possible polynyas". We are unclear what the review is after here as we do discuss the broken pack ice inside the pack ice in the final paragraph in this subsection.

L214: "ice types" –> "ice categories" Done

L223: Seems the peak extent for NT is in July. While it does appear so in the Figure, it is actually quite flat from July to October, but as you can see in Table 2 the peak does indeed occur in September.

L224/225: This statement for BT is also for September? Yes

L271: The Bellingshausen/Amundsen Sea is the only region where the maximum MIZ extent does NOT occur after the maximum pack ice extent during spring. Good point, this has now been noted

Figure 6: - Please correct "Bellinghausen" to "Bellingshausen". Done

The lines displaying other ice categories than MIZ and pack ice are very difficult to delineate in the images; one could try to increase the size of the Figure and make it to extend over two pages or to leave these other 3 categories out. We agree it is a bit hard to view the other ice categories and we tried different sizes, which during publication should resolve these features better than how we displayed the data in the word doc.

L272-282: The authors could also mention the large interannual variability in the MIZ extent for NT compared to BT. Done

L283: The authors could add "and bottom" behind "4th row" because they are referring to both regions before "[Figure ...]". Done

L291-303: While the maximum polynya extents can be identified in Figure 6 while they occur during late spring or summer they cannot be identified when they occur during winter. The authors therefore indeed might want to increase the size of the images in Figure 6 and let it extend over two pages. We can request the figure extends two pages during publication.

L306-317, Figure 7: Could the authors mention what the reference location / latitude is against which they define whether the ice edge expands or contracts over time?

One could assume that this is the latitude given in Figure 4 but one cannot find an obvious hint towards this. Perhaps it is arbitrarily chosen? It is 60S, note this figure is for illustrative purposes, Figure 9 shows the trends in all months/longitudes

And an issue coming from personal taste: An expansion of the ice edge is usually associated with cold / colder conditions. And usually cold is linked to bluish colors while warm is linked to reddish colors. It is the other way round in Figure 7. We are keeping it the way we have it as it is consistent with how NSIDC presents ice concentration anomalies (blue for negative, red for positive).

The authors could include into the caption that the bar in the range of the two upper maps give a reference for the trend in latitudinal movement of the extent of the respective ice category. This scale says 10 km / year which means that, e.g., in the Ross Sea there are regions where over the 30 years considered here the ice edge expanded by 300 km. Is that true? - The reviewer is surprized to see that even though we look at a 35-year long time series of data the trends in ice category expansion or contraction can be quite noisy - particularly for MIZ NT in the Indian Ocean sector north of Amery ice shelf or in the transition between Ross and Amundsen Sea. The same is true in a way to Pack ice NT. Can the authors comment on that? - Please increase the font size of the algorithm names in the two top maps of Figure 7. Figure 7 is primarily for illustrative purposes only as we found this a nice way to highlight differences between the two algorithms. We are not entirely sure what the reviewer means by noisy? Perhaps the alternating locations of negative trends rather than spatial consistency? It is true that there is an expansion on the order of ∼200-300km. For example, we see at longitude 215, the expansion is 200km, at 219.5 degrees it is 232 km. The physical processes for that are outside the scope of the present paper but would make a good follow on study.

L318 ff and the middle and bottom row of Figure 7: Could the authors also here note what the reference extent is against which they plot the trend in the expansion / contraction? To make this figure, first a sequence of monthly north longitudes for each 2

degree latitude bin was prepared. Then trends in that longitude were calculated. This is listed in km/yr, implying that the ice edge is moving north or south with time. The pack ice latitude is the monthly mean first occurrence of pack ice stepping north to south. Note however this is really for visual illustration to guide the discussion of where expansion and contraction occur in longitude.

L342-359 / Figure 8: - Please increase the font size and only show every second y-axis label to increase readability. - Following up with one of my early comments: The y-axis is again giving "Area" even though you mean "Extent". - Is there a reason why the authors switch between p-values and confidence level in percent? L356: Actually NT does not show positive MIZ extent trends anymore in November. Done

Figure 9: This is a very condense figure with regard to the annotation - particular of the y- and x-axes. The reviewer is wondering whether the authors would consider to either increase the size of the images themselves and therefore of the figure by itself or whether they find a different solution for this problem. The images themself are large enough to identify their message. - Is there a chance to highlight in the images which of the trends are significant? - The "possible polynya" cases so far have not been discussed that much and admittedly are also more noisy and difficult to interprete. How about the authors consider to skip that ice category completely in Figure 9 for the sake of presenting the other two more clearly? - The longitude 60W is a natural break point in the sea ice cover. How about the authors re-organize the images such that they start at 60W and end at 61W, i.e. in the Bellingshausen Sea? This way the authors could better visualize the eastward progression of the areas of positive pack ice trends with season in the Weddell Sea seen by both algorithms. This way the Weddell Sea would also not be split up. - Did the authors think to take a look at difference plots between the trends of the two algorithms? Such a difference plot might highlight the differences between the two algorithms even better. We have changed the figure to increase both the size of the figure and also note which trends are statistically significant by using small circles to denote the areas with trends not statistically significant. We kept the longitudinal range however from 0-360. We have also changed our discussion of the figure to be more succinct.

L370/371: It would fit to add a sentence here which underlines that in the Ross Sea also in this view (i.e. the way presented in Figure 9) the trends in BT MIZ extent are small (and not significant?). Done

L373: add "significant" after "statistically". Done

L379-382: The reviewer has difficulties to understand this argument. Southerly winds, i.e. winds blowing from the South to the North, cause the sea ice to be advected north - presumably with a substantial amount of lead formation. Openings generated by this more divergent type of sea ice motion should be seen by both algorithms in the same manner. Instead of voting for an additional oceanic influence (because this should again affect both algorithms the same way) the reviewer would vote for a smaller sensitivity to thin ice growing in openings and leads for BT than for NT. The authors could have a look into papers comparing these two algorithms. Also the paper by Holland and Kwok, 2012, mentioned further up, could shed some light on this issue. Ok

L383-385: This statement holds up until May, yes, for both BT and NT and the pattern re-emerges in October in NT and December in BT Ok

L390: Not even the large negative ones in the Weddell Sea in NT? Corrected with regards to Figure 9. When averaged over the entire region then it is true that the trends are not statistically significant (Table 3).

L396-398: Having read this one gets the impression that there is more broken sea ice left after summer melt in the region 30W to 0E. Also very interesting to see that the major positve trend in MAM (NT) or MAMJ (BT) is confined to a very narrow longitude band which moves to the East with progressing season. And it is this narrow longitude band where already in June negative trends pop up. Interesting. We agree this is interesting and added a comment following the reviewer.

L402/403: Suggestion to not comment on the intital retreat in September / October here; for ice category "pack ice" BT trends are almost zero and NT trends as well, they become larger in November / December. Agreed

L404: Which is an interesting notion given the fact that the eastern B.A. Seas have this finger-like structure of substantially negative trends extending from January well into July; NT pack ice trends are weaker here. We realize this sentence was confusing as it was pertaining to the Trends in Table 3 rather than Figure 9. We have rewritten to focus the discussion on Figure 9 and following reviewer 1, much of the discussion is now left for the conclusions.

L405ff: The authors mention both algorithms for MIZ trends but in BT they are very small except perhaps far east in May/June; in contrast there is a large blobb of negative MIZ extent trends visible for NT. The authors could re-write these sentences about what happens in the Bellingshausen/Amundsen Sea more clearly and perhaps also include in the advance-retreat cycling which has been mentioned and investigated by various authors for this region in particular (Stammerjohn is one candidate here, Harangozo another one). The entire section has been rewritten.

L415-427: Of course, for completeness, these two sectors need to be discusses as well. However, the reviewer feels that what is written for these two regions could be condensed even more because i) the patterns are really very similar between NT and BT and ii) pack ice trends are generally positive, more in BT than NT, right, and iii) trends in MIZ extent basically vary around zero with exceptions during September through December in both regions and both algorithms. The reviewer found it hard to get into discussion of significance and the relation to SIE change here. Done

L426-427: This notion is not reflected by Figure 9. The reviewer again suggests to skip the ice category "polynya" in this figure. As requested by the reviewer we have removed polynyas and the discussion from Figure 9.
L440-470 / Figure 10: - The images in Figure 10 could be re-ordered such that the topmost is MAM, then JJA and then SON - motivated by how the season progresses and motivated by the authors' order of describing the figure. Done

In JJA it is hard to see these trends and the fact that they oppose each other in Figure 10; maybe these are not that relevant? We felt it was important to keep the y-axis the same for each seasonal mean, which is why it is hard to see the trend in the MIZ in JJA since it is so small. Since it is not statistically significant in regards to the MIZ it is not all the relevant anyway in L451: The largest MIZ regime increase is 10 km / decade for NT ... that's not even half a grid cell ... and just between 1 and 1.5 grid cells in total over the 35-year period. Point noted in L453: Perhaps referring to Figure 7 would be a good idea to underline this notion? Done in L455: The authors could add that they are still referring to NT here. – We did not find that necessary in L467: This is an interesting observation and the reviewer is curious to see whether this weaker correlation has found its way into the discussion - it might be an effect of the larger sensitivity of NT to snow property variations particularly during late-winter / spring. We are not entirely clear at this point why there is a difference in correlation strength for SON. It may indeed be a result of snow properties as the reviewer suggests but without data to assess that (i.e. snow data), it is difficult to speculate with any certainty.

L472-481: This introduction into the seabird topic underlines my comment given with respect to Figure 5 that ice category "possible polynyas" is perhaps not that relevant for this study. While it is not entirely relevant for this study as we are focusing on a different sea bird, others who do studies on other bird species may find this discussion relevant. Thus, we have decided to keep this in.

L502-504/ Table 4: The reviewer is a bit surprized that the number of chicks counted in Dumont D'Urville, which is situated at longitude 140degE can serve as a response variable for a region which in 5 of the 6 months taken starts at least 20 deg further West and which extents up to a quarter way around the continent towards the West. Is that site (Dumont D'Urville) really representative for the geographical region chosen? Why didn't the authors chose a region which is kind of centered at the chick collection site? Perhaps the authors could comment on that. We used the distribution at sea of snow petrels recorded from miniaturized saltwater immersion geolocators during winter to define our area. Please refer to Delord et al. (2016) and their Figure 2 with the mean latitudes and longitudes of snow petrel during the non-breeding period recorded (see also our Table 4). We have now added more information on legend of Table 4 to clarify that point. "These areas were defined from the distribution at sea of snow petrels recorded from miniaturized saltwater immersion geolocators during winter (Delord et al. 2016)." Then, we study the carry over effect of winter conditions on the breeding success of snow petrel. Indeed, breeding success of snow petrels depends on sufficient body condition of the females, which in part reflects favorable environmental and foraging conditions prior to the breeding season [Barbraud and Chastel, 1999]. This is the first time that appropriate areas of the observed foraging range are used to study the carry over effect of winter conditions on the breeding performance of snow petrel, as this information did not existed previously.

L507: What is "AIC"? the Akaike's Information Criteri, this is now noted in the text

L513-526: - What is given at the x-axis of Figure 11. What is "pack ice bootstrap"? Are we looking at an anomaly? - The text (here and in the previous paragraph) speaks about AIC difference < 2 or > 2. Difference to what? What do the AIC values in Table 5 tell me? - The NT MIZ AIC value isn't that far away from BT pack ice AIC; why NT MIZ is not supported then? - The authors could use the same number of digits for all AIC values ... but could ask themselves how accurately AIC can be determined ... 1/100

or 1/10? - So what the authors wish to tell the reader here, in short, is that using BT pack ice fits best in relation to breeding success while using NT MIZ would have led to a wrong conclusion. NT pack ice and BT MIZ would have given in-different results. While the reviewer kind of likes the result (because it is known that BT provides more realistic sea ice concentration in the Southern Ocean than NT) the reviewer is wondering about i) how sensitive this results is with regard to application of a different model and ii) how the results would look like if the threshold sea ice concentration used to delineate MIZ and pack ice would have been 70% or even 60%? Yes the x - axis on figure 11 is the area of pack ice calculated with the bootstrap algorithm, and expressed as proportional anomalies relative to the mean over the observation period (1979- 2013). We have now specified that in the legend of Figure 11 and added line 506 that " sea ice covariates (MIZ and pack ice areas) were expressed as proportional anomalies to the mean, with xa(t)=(x(t)-xm)/xm; with xm the average value from 1979 to 2013. "

The Akaike Information Criterion (AIC) is a way of selecting a model from a set of models based on information theory [Burnham and Anderson, 2002]. AIC is largely used in the biological sciences but it has remained largely unused outside of this fields. The few examples that can be found in other fields, include the pharmacological sciences and social sciences (see the application to wines and policy by Snipes and Taylor, 2014). The chosen model with the lowest AIC is the one that minimizes the Kullback-Leibler distance between the model and the truth. It is defined as: AIC = -2 ( ln ( likelihood )) + 2 K where likelihood is the probability of the data given a model and K is the number of parameters in the model. AIC scores are often shown as $\Delta$AIC scores, or difference between the best model (smallest AIC) and each model (so the best model has a $\Delta$AIC of zero). We have now added in the legend of Table 5 the following sentence to clarify the model selection: "AIC scores are often interpreted as difference between the best model (smallest AIC) and each model referred as $\Delta$AIC. According to information theory, models with $\Delta$AIC<2 are both likely [Burnham and Anderson, 2002] but if a model shows a $\Delta$AIC>4 it is unlikely in comparison with the best model (smallest AIC). In our following model section, the model with the lowest AIC (highligthed in gray)

includes Pack ice calculated with the Bootstrap algorithm as a sea ice covariate. If AIC are sorted from lowest to highest value, the next model includes the sea ice covariate MIZ calculated with the NASA algorithm. It shows a $\Delta$AIC $\sim$8 from the best model, and thus is not well supported by the data in comparison to the best model. "

We have explored the various combinations of sea ice covariates, so our model selection is complete and robust. We could have explored non linear relationship, but we do not have a priori knowledge of the mechanisms that could lead to such non linear relationship. We could have implemented several covariates in the same models. However, it is not statistically appropriate in our case because sea ice covariates are strongly correlated. We have not evaluated how sensitive the results would be if the SIC for the MIZ changes.

L528-551: While what the authors write in these two paragraphs is for sure right the reviewer has difficulties to relate this to the main topic of the present paper - and in particular to the link with biology. We have edited to make more relevant but we wanted the discussion to also focus on other aspects that may be of broader interest.

L552-566: Here the authors try to establish the link to their paper. Suggestions are: - to clearly differentiate between sea ice extent and sea ice area. The latter includes sea ice concentration information and therefore is much better suited than sea ice extent to investigate changes and variability of the nature of a sea ice cover. Our values for the different ice categories are area as open water is accounted for when computing the area averages.

L557: There are other studies which could be mentioned here, which focus much better on polynyas and their temporal development and/or associated ice production, e.g. Tamura et al., 2008; Kern, 2009; Nihashi and Ohshima, 2015 - in L564-566: Holland and Kwok, 2012, would give enlightening information. We prefer not to go into that line of detail as it is outside the scope of the present paper.

L581-588: The reviewer suggests the authors make the notion that the BT used in the

Arctic is not the same as is used in the Antarctic - nut just in terms of the tie points but in terms of the used frequency combination. The results are therefore not compatible 1-to-1. I do not believe that is the case as tie points are computed from the BT algorithm on a daily basis using scatter plots of Tbs.

L594-595: Such a statement might need a reference. Isn't also most of the Antarctic sea ice snow covered? Yes most of the Antarctic sea ice is snow covered, but it is also prone to flooding, which makes the emissivity different than for dry snow on sea ice. The NT algorithm selects different tie points for each hemisphere which assumes a specific type of sea ice (generally snow covered). However work we have done shows the SIC is underestimated for thin ice, or bare ice.

L595-600: As with regard to tie point selection the reviewer recommends to perhaps take another look into relevant literature. While it may be correct that NT uses prede-fined tie points it is also correct that the BT needs to define tie points which is done based on the actual observations, yes, but except in a very few cases or when applying the BT regionally as e.g. in the Southern Ross Sea where thin ice exported from the Ross Ice Shelf dominates the scene, it is likely that the sea ice is snow covered as well. The BT algorithm selects tie points each day based on the distribution of the brightness temperatures, so they change day to day, whereas the NT tie points are fixed.

In particular the sentence in L597-600 is then perhaps a sub-optimal statement. - The reviewer is wondering whether the authors can give a reference for the "seasonal variations in emissivity can be very large"? We added 3 references

L601-619: Finally the authors come up with at least one of the key papers of results from intercomparison of NT to BT sea ice concentrations in the Antarctic (e.g. Comiso et al., 1997). One could also take a glimpse into Comiso and Steffen, JGR-C, 2001. Done One uncertainty factor for sea ice concentration retrieval is indeed the snow cover influence for which it is stated in the literature that the vertically stratified snow cover complicates sea ice concentration retrieval by the NT due to the stronger sensitivity of the 19 GHz channels to these effects - which is the main reason for the occasionally observed strong underestimation of the sea ice concentration by NT over pack ice compared to BT (Comiso and Steffen, 2001) and which is the main reason why the enhanced NT algorithm was developed (Markus and Cavalieri, 2000). The second factor is the different influence of thin ice on the sea ice concentration. Thin ice creates a negative bias in the sea ice concentration obtained as is illustrated, e.g. in Ivanova et al., 2015, and in Shokr and Kaleschke, 2012. On top of these the authors could mention other issues like gap layers, ice-snow interface flooding, formation of meteoric ice, snow metamorphism which all may or may not have an influence on the sea ice concentration which - to the reviewers' knowledge - nobody as quantified yet for Antarctic sea ice. We now mention these factors L606-610: This excursion to the Arctic seems confusing and could be deleted. We were trying to relate this to the reason why the MIZ differences between algorithms is larger in the Arctic than in the Antarctic. We made this statement clearer..

-L613-614: While one could agree to this notion one could also make the point that the definition of the MIZ via the sea ice concentration can be a very vague estimate. It can well be that even with 100reds of contemporary scenes with high-resolution optical imagery or SAR images one will not be able to better "validate" where the MIZ stops and pack ice starts. The reviewer has rather the feeling it is a matter of definition - and sea ice concentration might not be the ideal means for this.

References: Parkinson et al., 2012 in L53 not in refs. Should be Parkinson and Cavelieri, corrected.

Bintanja et al., in L72 has year 2012 in text but 2013 in refs. Should be 2013

Kohout et al. 2014, in L81 not in refs. Added

Ferrari 2014 in L107 is Ferrari et al. 2014 in refs. Changed

Ivanova et al., 2015 in L113 is now in The Cryosphere We could not find the final published version doing a google search, all results come as as Cryosphere Discussion.

Loeb et al. in L488 is 1993 in text but 1997 in refs. Should be 1997

Polvani and Smith, 2013 in L534 not in refs. Added

Hobbs et al., 2015, in L546 not in refs. Added

Ivanova et al. appears as Ivanova and others in the refs.

Steig et al. appears as Steig et al. in the refs. It is because there are more than 20 authors, if TC needs them all listed I can do that, but generally with that many authors, et al., is ok in the references. Kohout and Meylan, 2008 not used in text. Louzao et al. 2011 not used in text. Typos: L114: "mattes" –> "matters" Thanks for catching these, they have been corrected.

Anonymous Referee #3 The manuscript examines the variability and trends of Antarctic sea ice in three cat- egories: pack ice, marginal ice zone and coastal polynya derived by Bootstrap and NASA Team algorithms. Authors show that the differences in trends and variability be- tween these two datasets are quite large for pack ice and marginal ice, even though the differences in the total extents are relative small. The details within the ice pack is quite essential for the atmosphere-ocean-sea ice interaction, as well as biological studies, such as the study on snow petrel presented here. The manuscript reveals an important fact that satellite observed sea ice concentration can contain large errors and biases, and should be used with caution. I think that the study contains valuable information for polar community. However, I see some places need to be clarified. In general, the manuscript is rather tedious. Authors should reduce the discussions on insignificant results and focus on important and significant results. It is worth to be published after a minor revision. We thank the reviewer for their helpful comments. We agree at times the manuscript is tedious and it has been a difficult balance to highlight the importance of the results while also quantifying them. While Reviewer 1 found the paper quite easy to read for a non-specialist we also see this Reviewers point. We have tried to stream-line the discussion as much as possible in our revision. The research represents an original effort to evaluate SIC quality by comparing two datasets in marginal ice zone and pack ice. It fits well the scope of TC. The methodology is sound and conclusions are sufficiently supported by analyses. However, the title is not quite accurate. The study cannot conclude on the trend and variability of MIZ since the discrepancies be-tween two data products are quite large. The manuscript focuses on the discrepancies in MIZ, pack ice and coastal polynyas between data products. The title should reflect on that. We thank the reviewer for their comment. While we see the reviewers point, we feel the title remains appropriate given that we are assessing their variability. We added to the title the mention of looking at two sea ice algorithms. The title now reads: Mapping and Assessing Variability in the Antarctic Marginal Ice Zone, the Pack Ice and Coastal Polynyas in two Sea Ice Algorithms with implications on Breeding Success of Snow Petrels Technic/editorial issues. Figure 3. It would be helpful to add longitude lines to separate the different regions. In addition, bird study areas also need to be marked in this figure. Done

Line 306-317. The figure 7 needs some clarification. Is the expansion and contracting of outer ice edge relative to a zonal mean? If it is the case, what is the zonal mean of MIZ, in consideration of that the zonal mean pack ice is the mean of 85% and outer edge is the mean of 15%? It is relative to 60S. We have added that to the caption. Line342. Insert "area of the" or "extent of the" before pack ice. done Line 371, "While the sign of the Ross Sea sector trends from" done Figure 9 need some work. It would be very helpful if authors add contours of a confidence level in figure 9 so readers can see where and when significant trends occur. All x-axis and y-axis labels need to be enlarged, so they are visible. We have updated Figure 9 to have enlarged x and y axis and included the confidence level Page 10, There are many detailed discussions and the key results don't stand out. Authors should focus on the significant trends and their implications. We thank the reviewer for their comment but not entirely sure what the reviewer is after from this comment. Nevertheless, we have made some modifications that we hope clarify and add some more quantitative analysis to this section and add some significance of the results. Page 11, section 3.2.3. Authors need to explain how figure 10 is calculated. It should be seasonal mean and zonal mean, right? In addition, it would be nice to comment on whether the results derived from the width are consistent with or different from the results from ice extent/area presented in the early section. Figure 10 was calculated as follows: Step 1: We classify each 25 km pixel according to its class: MIZ, Pack Ice, Interior broken, Polynya. Step 2: Scan from north to south until the transitions between ocean to MIZ, MIZ to Pack Ice, Pack to Polynya or coast. Step 3: Record the latitude of the transition for each day. Step 4: Compute average latitude for each month at particular longitudes. Step 5: Convert monthly mean latitude boundaries to width in km. Step 6: Sum up for all longitudes. Step 7: Compute seasonal means.

We updated the text to read: Finally, we compute the overall width of the MIZ and pack ice following Strong and Rigor [2013] and produce seasonal means. Briefly, following the classification of each ice type, latitude boundaries are computed for each longitude and each day. These are averaged for each month to provide monthly mean latitude boundaries at each longitude. The boundaries are subsequently converted to width in km, and averaged for all longitudes. Finally, seasonal means are derived. Line 507. Please define "AIC". It is Akaike's Information Criterion and a common metric used in ecology for model selection. We have added this information to the paper. Page 13, Discussion. It is very clear that SIC derived from these two algorithms has large differences within the ice edge. To readers who use SIC for various studies, it is very important to know which product is more suitable for their need. It would be useful if authors could comment on whether there are other independent verifications? The snow petrel study is an excellent example. However, it is only related to the packed ice. We agree that it is important to know which product is more suitable for their need and we are not aware of any validation studies addressing these different ice types. However, we plan to do a validation exercise with visible imagery this year but it is outside the scope of this current paper and will likely be submitted to a remote sensing journal. Instead in this paper we focus on the fact that different results are obtained based on which algorithm is used, which is important to keep in mind when doing biological studies like this.

Please also note the supplement to this comment:
http://www.the-cryosphere-discuss.net/tc-2016-26/tc-2016-26-AC1-supplement.pdf

**Supplement:**

**Mapping and Assessing Variability in the Antarctic Marginal Ice Zone, the Pack Ice and Coastal Polynyas in two Sea Ice Algorithms with implications on Breeding Success of Snow Petrels**

Julienne C. Stroeve[1,2], Stephanie Jenouvrier[3,4], G. Garrett Campbell[1], Christophe Barbraud[4] and Karine Delord[4]

[1]National Snow and Ice Data Center, Cooperative Institute for Research in Environmental Sciences, University of Colorado, Boulder, CO, USA

[2]Center for Polar Observation and Modelling, University College London, London, UK

[3]Woods Hole Oceanographic Institution, Woods Hole, MA, USA

[4]Centre d'Etudes Biologiques de Chizé, UMR 7372 CNRS, 79360 Villiers en Bois, France

**Abstract**

Sea ice variability within the marginal ice zone (MIZ) and polynyas plays an important role for phytoplankton productivity and krill abundance. Therefore, mapping their spatial extent, seasonal and interannual variability is essential for understanding how current and future changes in these biologically active regions may impact the Antarctic marine ecosystem. Knowledge of the distribution of MIZ, consolidated pack ice and coastal polynyas to the total Antarctic sea ice cover may also help to shed light on the factors contributing towards recent expansion of the Antarctic ice cover in some regions and contraction in others. The long-term passive microwave satellite data record provides the longest and most consistent record for assessing the proportion of the sea ice cover that is covered by each of these ice categories. However, estimates of the amount of MIZ, consolidated pack ice and polynyas depends strongly on what sea ice algorithm is used. This study uses two popular passive microwave sea ice algorithms, the NASA Team and Bootstrap, and applies the same thresholds to the sea ice concentrations to evaluate the distribution and variability in the MIZ, the consolidated pack ice and coastal polynyas. Results reveal that the seasonal cycle in the MIZ and pack ice is generally similar between both algorithms, yet the NASA Team algorithm has on average twice the MIZ and half the consolidated pack ice area as the Bootstrap algorithm. Trends also differ, with the Bootstrap algorithm suggesting statistically significant trends towards increased pack ice area and no statistically significant trends in the MIZ. The NASA Team algorithm on the other hand indicates statistically significant positive trends in the MIZ during spring. Potential coastal polynya area and broken ice within the consolidated ice pack is also larger in the NASA Team algorithm. The timing of maximum polynya area may differ by as much as 5 months between algorithms. These differences lead to different relationships between sea ice characteristics and biological processes, as illustrated here with the breeding success of an Antarctic seabird.

**1. Introduction**

Changes in the amount of the ocean surface covered by sea ice play an important role in the global climate system. For one, sea ice and its snow cover have a high surface reflectivity, or albedo, reflecting the majority of the sun's energy back to space. This helps to keep the polar regions cool and moderates the global climate. When sea ice melts or retreats, the darker (lower albedo) ocean is exposed, allowing the ocean to absorb solar energy and warm, which in turn melts more ice, creating a positive feedback loop. During winter, sea ice helps to insulate the ocean from the cold atmosphere, influencing the exchange of heat and moisture to the atmosphere with impacts on cloud cover, pressure distribution and precipitation. These in turn can lead to large-scale atmospheric changes, affecting global weather patterns [e.g. *Jaiser et al.*, 2012]. Sea ice also has important implications for the entire polar marine ecosystem, including sea ice algae, phytoplankton, crustaceans, fish, seabirds, and marine mammals, all of which depend on the seasonal cycle of ice formation in winter and ice melt in summer. For example, sea ice melt stratifies the water column, producing optimal light conditions for stimulating bloom conditions. Antarctic sea birds rely upon the phytoplankton bloom for their breeding success and survival [e.g. *Park et al.*, 1999].

In stark contrast to the Arctic, which is undergoing a period of accelerated ice loss [e.g. *Stroeve et al.*, 2012; *Serreze and Stroeve*, 2015], the Antarctic is witnessing a modest increase in total sea ice extent [*Parkinson and Cavalieri*, 2012; *Simmonds et al.*, 2015]. Sea ice around Antarctica reached another record high extent in September 2014, recording a maximum extent of more than 20 million km$^2$ for the first time since the modern passive microwave satellite data record began in October 1978. This follows previous record maxima in 2012 and 2013 [*Reid et al.*, 2015], resulting in an overall increase in Antarctic September sea ice extent of 1.1% per decade since 1979. While the observed increase is statistically significant, Antarctic's sea ice extent (SIE) is also highly variable from year to year and region to region [e.g. *Maksym et al.*, 2012; *Parkinson and Cavalieri*, 2012; *Stammerjohn et al.*, 2012]. For example, around the West Antarctic Peninsula (WAP), there have been large decreases in sea ice extent and sea ice duration [e.g. *Ducklow et al.*, 2012; *Smith and Stammerjohn*, 2001], coinciding with rapid warming since 1950 [*Ducklow et al.*, 2012].

[revised manuscript text omitted]

Using these SIC fields, we define six binary categories of sea ice based on different SIC thresholds [**Table 1**]. Because the marginal ice zone is highly dynamic in time and space, it is difficult to precisely define this region of the ice cover. *Wadhams* [1986] defined the MIZ as that part of the ice cover close enough to the open ocean boundary to be impacted by its presence, e.g. by waves. Thus the MIZ is typically defined as the part of the sea ice that is close enough to the open ocean to be heavily influenced by waves, and it extends from the open ocean to the dense pack ice. In this study, we define the MIZ as extending from the outer sea ice/open ocean boundary (defined by SIC $\geq$ 0.15 ice fraction) to the boundary of the consolidated pack ice (defined by SIC = 0.80). This definition was previously used by *Strong and Rigor* [2013] to assess MIZ changes in the Arctic and matches the upper SIC limit used by the National Ice Center in mapping the Arctic MIZ. The consolidated ice pack is then defined as the area south of the MIZ with ice fractions between $0.80 \leq$ SIC $\leq 1.0$. Potential coastal polynyas are defined as regions near the coast that have SIC < 0.80.

To automate the mapping of different ice categories, radial transects from 50 to 90S are individually selected to construct one-dimensional profiles [**Figure 1**]. The algorithm first steps from the outer edge until the 0.15 SIC is detected, providing the latitude of the outer MIZ edge. Next, the algorithm steps from the outer MIZ edge until either the 0.80 SIC is encountered, or the continent is reached. Data points along the transect between these SIC thresholds are flagged as the MIZ. In this way, the MIZ includes an outer band of low sea ice concentrations that surrounds a band of inner consolidated pack ice, *but* sometimes the MIZ also extends all the way to the Antarctic coastline (as sometimes observed in summer). South of the MIZ, the consolidated ice pack ($0.80 \leq$ SIC $\leq 1.0$) is encountered; however, low sea ice concentrations can appear near the coast inside the pack ice region as well. These are areas of potential coastal polynyas. While it is difficult to measure the fine scale location of a polynya at 25km spatial resolution, the lower sea ice concentrations provide an indication of some open water near the coast, which for sea birds provides a source of open water for foraging. We have previously tested mapping polynyas using a SIC threshold of 0.75 and 0.85 for the NASA Team and Bootstrap algorithms, respectively, and found that these thresholds provided consistent polynya areas between the two algorithms and matched other estimates of the spatial distribution of polynyas [see *Li et al.,* 2016]. However, for this study we chose just one threshold, a compromise between the two algorithms, so that we can better determine the sensitivity of using the same threshold on polynya area and timing of formation.

Using our method of radial transects, the algorithm then steps from the coast northward and flags pixels with < 0.80 SIC until a 0.80 SIC pixel appears and defines that region as a potential coastal polynya. Within the consolidated pack ice (and away from the coast), it is also possible to encounter instances where $0.15 < SIC < 0.80$ or $SIC < 0.15$. These are flagged as open pack ice and open water areas within the consolidated pack ice, respectively. Finally, an ocean mask derived from climatology and distributed by NSIDC was applied to remove spurious ice concentrations at the ice edge as a result of weather effects.

**Figure 2** shows sample images of the classification scheme as applied to the NASA Team and Bootstrap algorithms on days 70 (March 11) and 273 (September 30), respectively, in 2013. During the fall and winter months when the ice cover is expanding there is a well-established consolidated pack ice region, surrounded by the outer MIZ. Coastal polynyas are also found surrounding the continent in both algorithms. The BT algorithm tends to show a larger consolidated ice pack than NT, particularly during the timing of maximum extent. During the melt season there is mixing of low and high ice concentrations, leading to mixtures of different categories, which is still seen to some extent in the March images. However, during March areas of polynyas (green), open water (pink) and open pack ice (orange) appear to extend from the coastline in some areas (e.g. southern Weddell and Ross seas). While any pixel with SIC < 0.8 adjacent to the coastal boundary is flagged as potential polynya when stepping northwards, if a pixel is already flagged as MIZ or consolidated pack ice when stepping southwards, it remains flagged as MIZ or pack ice. After that analysis, a check for pixels with SICs less than 0.8 is done to flag for broken ice or open water. Thus, during these months (e.g. December to February or March), the physical interpretation of the different ice classes may be less useful.

Using the binary classification scheme, daily gridded fields at each 25 km pixel are obtained. Using this gridded data set we then obtain regional averages for five 
[revised manuscript text omitted]

pack ice trends are spatially consistent between both algorithms, though not all trends are
statistically significant, particularly for the NT algorithm. The largest consistency occurs in the
the western Ross Sea, where positive trends are seen in both algorithms, statistically significant
from March to November (p<0.01) in the BT algorithm, and from January to July and October to
November in the NT algorithm. Note also that both algorithms show statistically significant
positive trends in the MIZ from January to March in the western Ross Sea and generally negative
trends in the eastern Ross Sea. This pattern switches from June to December, with mostly
negative MIZ trends in the western Ross Sea and positive trends in the eastern Ross Sea. In
particular, the statistically significant positive trends in the MIZ in the NT algorithm occur at the
time of year with the largest overall trends in the SIE in this region. This would suggest perhaps
different interpretation of processes impacting the overall ice expansion in the Ross Sea
depending on which algorithm is used.
In the B/A Sea, statistically significant positive trends in pack ice are limited to May through
August in the NT algorithm and June and July in the BT algorithm. The positive NT pack ice
trends are offset by negative trends in the NT MIZ. Both algorithms exhibit negative pack ice
trends during other months that are consistent between the algorithms, though larger in
magnitude for the BT algorithm. This is generally compensated by statistically significant
negative trends in the NT MIZ to give an overall negative decline of total extent.
Trends in the pack ice are also consistent between algorithms in the Weddell Sea, with
statistically significant trends generally occurring at the same longitude and during the same
months. The positive pack ice trends in MAM (NT) or MAMJ (BT) are confined to a very
narrow longitude band which moves to the east with progressing season. Then in June, and
continuing for several months, negative pack ice trends occur. For both algorithms, trends in the
MIZ are generally not statistically significant, except for some positive trends in the eastern

Deleted: , the largest regional positive trends in total SIE are found at a rate of 119,000 km² per decade [e.g. *Turner et al.*, 2015], accounting for about 60% of the circumpolar ice extent increase. In the BT algorithm this is entirely a result of
Deleted: While the sign of the Ross Sea sector trends from the NT algorithm are spatially consistent with the pack ice trends shown in the BT algorithm, trends are only statistically from April to June (p<0.05). Instead,

Weddell Sea from January to March and negative trends mostly from June to November near 330 degrees longitude.

Finally, in the Pacific and Indian Oceans we again see spatial consistency in pack ice and MIZ trends for both algorithms, with generally larger (smaller) pack ice (MIZ) trends for the BT algorithm, though trends are closer in magnitude in the Pacific sector from March to July. Pack ice trends are generally positive, more in BT than NT and trends in MIZ extent basically vary around zero with exceptions during August through December in both algorithms in the Pacific Ocean.

In summary, while the magnitude of trends differs between both algorithms, there is general spatial consistency in the patterns of positive and negative trends in the consolidated pack ice and the MIZ. Results suggest that positive trends in total SIE are generally a result of statistically significant positive trends in the consolidated pack ice in the BT algorithm in all sectors of the Antarctic, except for the Bellingshausen/Amundsen Sea sector and the Weddell Sea during ice retreat. The NT algorithm on the other hand suggests more instances of statistically significant positive trends in the MIZ, though this is highly regionally dependent.

**3.2.3 Seasonal Trends in MIZ and Pack Ice Width**

Finally, we compute the overall width of the MIZ and pack ice following *Strong and Rigor* [2013] and produce seasonal means. Briefly, following the classification of each ice category, latitude boundaries are computed for each longitude and each day. These are averaged for each month to provide monthly mean latitude boundaries at each longitude. The boundaries are subsequently converted to width in km, and averaged for all longitudes. Finally, seasonal means are derived.

Time-series of seasonal means of the circumpolar MIZ width and pack ice width are shown in **Figure 10** for all seasons except summer when the results are noisy. As we may expect following the previous results, the NT MIZ width is larger and the pack ice width is smaller than the seen in the BT algorithm. During autumn (MAM) however, the differences in widths for both the MIZ and the pack ice between the algorithms are largely reduced compared to the other seasons. For example the difference in 1979-2013 pack ice width between the algorithms during MAM is 60 km, 121 km in JJA and 139 km in SON. Similarly, the long-term mean MIZ width differences are 54 km (MAM), 74 km (JJA) and 83 km (SON). In addition, during autumn, trends in the MIZ and pack ice are largely consistent between the two algorithms, with no trend in the MIZ and increases in the pack ice on the order of 21.2 km dec$^{-1}$ and 20.0 km dec$^{-1}$ (p<0.01) for the BT and NT algorithms, respectively. This is the season with the largest trends in the pack ice width, representing a 21% widening over the satellite record.

[revised manuscript text omitted]

In the following, we expect that an extensive pack ice during winter may reduce breeding success the following breeding season by protecting the under ice community from predation, while an extensive MIZ may increase breeding success by providing easier access to foraging. With the classifications as defined by both algorithms we calculated the MIZ and pack ice area in a wide rectangular sector defined by the migration route of the snow petrel [*Delord et al.,* 2016] from April to September [see **Table 4** for latitude and longitude limits]. This is the first time that appropriate areas of the observed foraging range are used to study the carry over effect of winter conditions on the breeding performance of snow petrel, as this information did not existed previously. Using these locations, we averaged the MIZ and pack ice extents over the entire winter from April to September. We next employed a logistic regression approach to study the effects of MIZ and pack ice area within this sector and evaluate the impacts on breeding success the following summer. The response variable was the number of chicks $C_t$ in a breeding season $t$, from 1979 to 2014 collected at Terre Adélie, Dumont D'Urville [*Barbraud and Weimerskirch*, 2001, *Jenouvrier et al.*, 2005].

Effects of MIZ and pack ice area were analyzed using Generalized Linear Models (GLM) with logit-link functions and binomial errors fitted in R using the package glm. Specifically, the response variable is the number of chicks $C_t$ in a breeding season $t$, from 1979 to 2014 collected at Terre Adelie, Dumont D'Urville [*Barbraud and Weimerskirch*, 2001, *Jenouvrier et al.*, 2005]. It follows a binomial distribution, such that $C_t \sim Bin(\mu_t, N_t)$, where $N_t$ is the number of breeding pairs and $\mu_t$ is the breeding success in year $t$. The breeding success is a function of the MIZ and pack ice covariates at time $t$ (COV) such as:

$$\mu_t = \beta_0 + \beta_1 COV_{(t)}$$

To select the covariate that most impacts the breeding success of snow petrels, we applied the information-theoretic (I-T) approaches [*Burnham et al.*, 2011]. This is based on quantitative measures of the strength of evidence for each hypothesis (Hi) rather than on "testing" null hypotheses based on test statistics and their associated P values. To quantify the strength of evidence for each hypothesis (Hi) – here the effect of each covariate on the breeding success- we used the common criteria AIC (the Akaike's Information Criteria), where AIC = - 2 log(L) + 2K [*Akaike*, 1973]. The term, -2 log(L), is the "deviance" of the model, with log(L) the maximized log-likelihood and K the total number of estimable parameters in the model. The chosen model is the one that minimizes the AIC, in orther words, minimizes the Kullback-Leibler distance between the model and truth. The ability of two models to describe the data was assumed to be "not different" if the difference in their AIC was < 2 [*Burnham and Anderson*, 2002]. Note the AIC is a way of selecting a model from a set of models based on information theory [*Burnham and Anderson*, 2002], and is largely used in biological sciences. While non-linear models may be more appropriate as ecological system relationships are likely more complex than linear relationships, without *a priori* knowledge of the mechanisms that could lead to such non-linear relationships, it is extremely difficult to set meaningful hypothesis to be included in the model selection.

**Table 5** summarizes model selection. The model with the lowest AIC (highlighted in gray) suggests the BT pack ice as a sea ice covariate. If AIC are sorted from lowest to highest value, the next model includes the sea ice covariate MIZ calculated with the NASA algorithm. However, it shows a ΔAIC ~8 from the best model, and thus the NT MIZ is not well supported by the data in comparison to the best model. The relationship between BT pack ice and breeding success is negative [**Figure 11**]. In other words, a more extensive consolidated pack ice during winter tends to reduce breeding success the following summer by limiting foraging opportunities. The effect of the MIZ however was uncertain, contrary to what one may expect given the increased opportunities for foraging within the MIZ. However, if we had only used ice classifications based on the NASA Team algorithm, the model with the lowest AIC would have suggested an importance of the MIZ. We would have then concluded a negative effect of the MIZ on the breeding success of snow petrels, contrary to what one may expect given that the MIZ is the main feeding habitat of the species. By using both algorithms, we instead conclude that the breeding success of snow petrels is negatively affected by the pack ice area as calculated
with the Bootstrap algorithm.

**5. Discussion**

While the main purpose for doing the classification of different ice categories is for
interdisciplinary studies of sea bird breeding success, the results may also be useful for
attribution of the observed sea ice changes. The positive trends in Antarctic sea ice extent are
currently poorly understood and are at odds with climate model forecasts that suggest the sea ice
should be declining in response to increasing greenhouse gases and stratospheric ozone depletion
[e.g. *Turner et al.*, 2013; *Bitz and Polvani*, 2012; *Sigmond and Fyfe*, 2010]. However, several
modeling studies, such as those used in the phase 5 Coupled Model Intercomparison Project
(CMIP5), have suggested that the sea ice increase over the last 36 years remains within the range
of intrinsic of internal variability [e.g. *Bitz and Polvani*, 2012; *Turner et al.*, 2013; *Mahlstein et
al.*, 2013; *Polvani and Smith,* 2013; *Swart and Fyfe*, 2013]. Earlier satellite from the 1960s and
1970s and from ship observations suggest periods of high and low sea ice extent, and thus high
natural variability [*Meier et al.*, 2013; *Gallaher et al.*, 2014]. Further evidence comes from ice
core climate records, which suggest that the climate variability observed in the Antarctic during
the last 50 years remains within the range of natural variability seen over the last several hundred
to thousands of years [*Thomas et al.*, 2013; *Steig et al.*, 2013]. Thus, we may require much
longer records to properly assess Antarctic sea ice trends in contrast to the Arctic, where negative
trends are outside the range of natural variability and are consistent with those simulated from
climate models.
While many assessments of how Antarctic sea ice trends and variability compare with
climate models have focused on the net circumpolar sea ice extent, it is the regional variability
that becomes more important. For example, *Hobbs et al.* [2015] argue that when viewing trends
on a regional basis, the observed summer and autumn trends fall outside of the range of natural
variability as simulated by present-day climate models, with the signal dominated by opposing
trends in the Ross Sea and the Bellingshausen/Amundsen seas. These results have questioned the
ability of climate models to correctly simulate processes at the regional level and within the
southern ocean-atmosphere-sea ice coupled system.
The net take-away point from these studies is that the net circumpolar changes in sea ice
extent do not enhance our understanding of how the Antarctic sea ice is changing. Instead our
focus should be on what drives regional and seasonal sea ice changes, including feedbacks and
competing mechanisms. The results of this study may help to better understand regional and total
changes in Antarctic sea ice by focusing not only on the total ice area, but also on how the
consolidated pack ice, the marginal ice zone and coastal polynyas are changing. Differences in
climatologies and trends of the different ice classes may suggest different processes are likely
contributing to their seasonal and interannual variability. In addition, the different contributions
of ice categories towards the overall expansion of the Antarctic sea ice cover between algorithms
may in turn influence attribution of the observed increase in SIE. For example, within the highly
dynamic MIZ region, intense atmosphere-ice-ocean interactions take place [e.g. *Lubin and
Massom*, 2006] and thus an expanding or shrinking MIZ may help to shed light on the relative
importance of atmospheric or oceanic processes impacting the observed trends in total SIE.
Another issue is whether or not new ice is forming along the outer edge of the pack ice or if it is
all being dynamically transported from the interior.

However, a complication exists, what sea ice algorithm should be used for such assessments? In this study we focused on using passive microwave satellite data for defining the different ice categories used here as it is the longest time-series available and is not limited by polar darkness or clouds. However, results are highly dependent on which sea ice algorithm is used to look at the variability in these ice classes, which will also be important in assessing processes contributing to these changes as well as implications of these changes to the polar marine ecosystem. In this study, the positive trends in circumpolar sea ice extent over the satellite data record are primarily driven by statistically significant trends ($p<0.05$) in expansion of the consolidated pack ice in both sea ice algorithms. However, an exception occurs in the NASA Team sea ice algorithm after the ice pack reaches its seasonal maximum extent when the positive trends in the pack ice are no longer as large, nor statistically significant. Instead, positive trends in the MIZ dominate during September and October ($p<0.10$). This is in stark contrast to the Bootstrap algorithm, which shows a declining MIZ area from March through November.

The algorithms also give different proportions of how much the total ice cover consists of consolidated ice, MIZ or polynya area. In some regions, such as the Pacific Ocean sector, the NT algorithm suggests the MIZ is the dominant ice category whereas in the BT algorithm, the pack ice is dominant, which is true for all sectors analyzed in the Bootstrap algorithm. Considering the circumpolar ice cover, the MIZ in the NASA Team algorithm is on average twice as large as in the Bootstrap algorithm. In the Arctic, *Strong and Rigor* [2013] found the NASA Team algorithm gave about three times wider MIZ than the Bootstrap algorithm. In this case, the Bootstrap results agreed more with MIZ widths obtained from the National Ice Center (NIC).

While we find consistency in trends in pack ice and the MIZ, there are some important differences that may influence interpretation of processes governing sea ice changes. For example, in the Ross Sea, the largest regional positive trends in total SIE are found at a rate of 119,000 km$^2$ per decade [e.g. *Turner et al.*, 2015], accounting for about 60% of the circumpolar ice extent increase. This is entirely a result of large positive trends in the pack ice in the BT algorithm from March to November ($p<0.01$) whereas the NT algorithm shows statistically significant increases in the MIZ. Several studies have suggested a link between sea ice anomalies in the Ross Sea and the wind-field associated with the Amundsen Sea Low (ASL) [e.g. *Fogt et al.*, 2012; *Hosking et al.*, 2013; *Turner et al.*, 2012]. The strengthened southerly winds over the Ross Sea cause a more compacted and growing consolidated ice cover in the BT algorithm at the expense of a shrinking MIZ, whereas in the NT algorithm the area of the MIZ is increasing more than the pack ice during autumn, which may suggest a smaller sensitivity to thin ice growing in openings and leads for BT than for NT. While this is true as averaged over the entire Ross Sea sector, Figure 9 highlights that the area-averaged trends hide important spatial variability.

In the Weddell Sea, expansion of the overall ice cover is only statistically significant during the autumn months (MAM) [e.g. *Turner et al.*, 2015]. During this time-period, both algorithms agree on statistically significant positive trends in the pack ice area, that extend through May for NT ($p<0.05$) and through June for BT ($p<0.05$). Statistically significant trends are also seen during March in the MIZ, with larger trends in the NT algorithm ($p<0.01$). Thus, overall expansion of sea ice in the Weddell during autumn is in part driven by expansion of the MIZ early in the season, after which it is controlled by further expansion of the consolidated pack.

In contrast, the B/A Sea is a region undergoing declines in the overall ice cover [e.g. *Parkinson and Cavalieri*, 2012; *Stammerjohn et al.*, 2012]. Separating out trends for both the pack ice and the MIZ reveals positive trends during winter (JJA), and negative trends in the consolidated pack ice during the start of ice expansion in March and April. However, when averaging over the entire region, the trends are generally not statistically significant except for positive trends during winter in the NT algorithm. This is the only region where the BT algorithm does not show statistically significant trends in the pack ice. In the NT algorithm, the overall sea ice decline is largely a result of negative trends in the MIZ, consistent with the observation that the SIE trends in the Bellingshausen/Amundsen Sea are largely wind-driven, so it would be expected that the wind-driven compaction would lead to decreased MIZ and increased pack ice. In regards to potential coastal polynyas, the largest expansion of polynya area is found in the Bellingshausen/Amundsen Sea during November, whereas small increases in polynya area are found in both the Indian and Pacific sector during the ice expansion phase. Outside of these regions/months, no significant changes in coastal polynya area are observed.

Differences between the algorithms are not entirely surprising as the two algorithms use different channel combinations with different sensitivities to changes in physical temperature [*Comiso et al.,* 1997; *Comiso and Steffen,* 2001]. In addition, the NT uses previously defined tie points for passive microwave radiances over known ice-free ocean, and ice types, defined as type A and B in the Antarctic, as the radiometric signature between first-year and multiyear ice in the Antarctic is lost. The ice is assumed to be snow-covered when selecting the tie points, which can result in an underestimation of sea ice concentration if the ice is not snow covered [e.g. *Cavalieri et al.*, 1990]. While large-scale validation studies are generally lacking, a recent study of the interior of the ice pack in the Weddell Sea in winter suggested that the Bootstrap algorithm shows a better fit to upward looking sonar data [*Connolley,* 2005]. This suggests that broken water inside the pack ice as recorded by the NASA Team algorithm during winter may be erroneously detected.

However, another complication is that seasonal variations in sea ice and snow emissivity can be very large, leading to seasonal biases in either algorithm [e.g. *Andersen et al.,* 2007; *Willmes et al.*, 2014; *Gloersen and Cavalieri,* 1986]. In addition, ice-snow interface flooding, formation of meteoric ice and snow metamorphism all impact sea ice concentrations, which have not been quantified yet for Antarctic sea ice, and trends in brightness temperatures found in the Weddell Sea may reflect increased melt rates or changes in the melt season [*Willmes et al.,* 2014]. The advantage of the Bootstrap algorithm is that the ice concentration can be derived without an *a priori* assumption about ice type, though consolidated ice data points are sometimes difficult to distinguish from mixtures of ice and open ocean due to the presence of snow cover, flooding or roughness effects.

While one may expect the Bootstrap algorithm to provide more accurate results than the NASA Team algorithm, near the coast the BT algorithm has been shown to have difficulties when temperatures are very cold. Because the NT algorithm uses brightness temperature ratios it is largely temperature independent. During summer or for warmer temperatures, the NT algorithm may indeed be biased towards lower sea ice concentrations whereas the BT algorithm may be biased towards higher ice concentrations [e.g. *Comiso et al.,* 1997]. This will result in different proportions of MIZ and consolidated pack ice. In the Arctic, the MIZ is not only driven by wave mechanics and flow breaking (dynamic origin), but also by melt pond processes in summer (thermodynamic origin) [*Arnsten et al.*, 2015]. Thus, larger sensitivity of the NT algorithm to melt processes may be one reason for the larger discrepancy observed in the MIZ between the algorithms the Arctic. Interestingly, the BT algorithm shows less interannual variability in the MIZ, consolidated pack ice and potential coastal polynyas compared to NT (as shown by the smaller standard deviations). This would in turn influence assessments of atmospheric or oceanic conditions driving observed changes in the ice cover.

What is clear is that more validation is needed to assess the accuracy of these data products,
especially for discriminating the consolidated pack ice from the MIZ. Errors likely are larger in
the MIZ because of the coarse spatial resolution of the satellite sensors. The MIZ is very
dynamic in space and time, making it challenging to provide precise delimitations using sea ice
concentrations that are in turn sensitive to melt processes and surface conditions. Another
concern is that mapping of the consolidated ice pack does not always mean a compact ice cover.
The algorithms may indicate 100% sea ice concentration (e.g. a consolidated pack ice), when in
reality the ice consists of mostly brash ice and small ice floes more representative of the MIZ.
Future work will focus on validation with visible imagery.

**Conclusions**

Antarctic sea ice plays an important role in the polar marine ecosystem. While total Antarctic
sea ice cover is expanding in response to atmospheric and oceanic variability that remains to be
fully understood, one may expect that these increases would also be manifested in either
equatorward progression of the MIZ or the consolidated pack ice or both, that in turn would
impact the entire trophic web, from primary productivity, to top predator species, such as
seabirds. In this study we identified several different ice categories using two different sets of
passive microwave sea ice concentration data sets. The algorithms are in agreement as to the
location of the northern edge of the total sea ice cover, but differ in regards to how much of the
ice cover consists of the marginal ice zone, the consolidated ice pack, the size of potential
polynyas as well as the amount of broken ice within the consolidated ice pack. Here we use sea ice concentration thresholds of $0.15 \leq SIC < 0.80$ to define the width of the MIZ
and $0.80 \leq SIC \leq 1.0$ to define the consolidated pack ice. Yet applying the same thresholds for
both sea ice algorithms results in a MIZ from the NASA Team algorithm that is on average twice
as large as in the Bootstrap algorithm and considerably more broken ice within the consolidated
pack ice. Total potential coastal polynya areas ($SIC \leq 0.80$) also differ between the algorithms,
though differences are generally smaller than for the MIZ and the consolidated pack ice. While
we do not precisely resolve polynyas, these potential coastal polynyas (i.e. open water areas near
the coast) are important foraging sites for sea birds.
While the spatial extents of the different ice classes may differ, the seasonal cycle is
generally consistent between both algorithms. Climatologically, the advance of the consolidated
ice pack happens over a much longer period (~7-8 months) than the retreat (~4-5 months), while
the MIZ exhibits a longer advance period (~8-10 months). This seasonal cycle in
expansion/contraction of the ice cover is in general agreement with results by *Stammerjohn et al.*
[2008] who showed sea ice retreat begins in September at the outer most edge of the sea ice and
continues poleward over the next several months. However, what these results show is that while
the pack ice starts to retreat around September, this in turn results in a further expansion of the
MIZ, the amount of which is highly dependent on which algorithm is used. The timing of when
the maximum polynya extent is reached however can differ by several months between the
algorithms in regions such as the Bellingshausen/Amundsen Sea and the Pacific Ocean.
Since the MIZ is an important region for phytoplankton biomass and productivity [e.g. *Park*
*et al.,* 1999], mapping seasonal and interannual changes in the MIZ is important for
understanding changes in top predator populations and distributions. However, as we show in
this study, results are highly dependent on which sea ice algorithm is used for delineating the
MIZ, which may result in different conclusions when using this data in ecosystem models. To
* * *
**Moved (insertion) [2]**
* * *
**Moved (insertion) [1]**

**Moved up [2]:** Furthermore, accurately mapping the extent of the MIZ from coarse resolution satellite data such as that from passive microwave sensors remains problematic. The MIZ is very dynamic in space and time, making it challenging to provide precise delimitations using sea ice concentrations that are in turn sensitive to melt processes and surface conditions.

hightlight this sensitivity, we examined the impact the winter MIZ and consolidated pack ice
area as derived from both algorithms would have on the breeding success of snow petrels the
following summer. The different proportions of MIZ and consolidated pack ice between
algorithms affected the inferences made from models tested even if trends were of the same sign.
Given the sensitivity of the relationships between the consolidated pack ice/MIZ and breeding
success of this species, caution is warranted when doing this type of analysis as different
relationships may emerge as a function of which sea ice data set is used in the analysis. Further
work is needed to validate the accuracy of the distribution of the MIZ and consolidated pack ice
from passive microwave so that the data will be more useful for future biological and ecosystem
studies.
**Acknowledgements**

This work is funded under NASA Grant NNX14AH74G and NSF Grant PLR 1341548. We are
grateful to Sharon Stammerjohn for her helpful comments on the manuscript. Gridded fields of
the different ice classifications from both algorithms are available via ftp by contacting J.
Stroeve. We thank all the wintering fieldworkers involved in the collection of snow petrel data at
Dumont d'Urville since more than 50 years, as well as Institut Paul Emile Victor (program IPEV
n°109, resp. H. Weimerskirch), Terres Australes et Antarctiques Françaises and Zone Atelier
Antarctique (CNRS-INEE) for support.

[revised manuscript text omitted]

**Table 5.** Results of model selection for the relationship between pack ice and MIZ on breeding
success of snow petrel. The model with the lowest AIC is highlighted in gray. AIC scores are
often interpreted as difference between the best model (smallest AIC) and each model referred as
$\Delta$AIC. According to information theory, models with $\Delta$AIC < 2 are both likely [*Burnham and*
*Anderson*, 2002] but if a model shows a $\Delta$AIC > 4 it is unlikely in comparison with the best
model (smallest AIC).

[revised manuscript text omitted]

Finally, the largest expansion of polynya area is found in the Bellingshausen/Amundsen Sea during November, whereas small increases in polynya area are found in both the Indian and Pacific sector during the ice expansion phase. Outside of these regions/months, no significant changes in coastal polynya area are observed.

| Page 12: [5] Formatted | Microsoft Office User | 5/13/16 4:17 PM |
|---|---|---|

Font:(Default) Times New Roman, Not Highlight

| Page 12: [5] Formatted | Microsoft Office User | 5/13/16 4:17 PM |
|---|---|---|

Font:(Default) Times New Roman, Not Highlight

| Page 12: [5] Formatted | Microsoft Office User | 5/13/16 4:17 PM |
|---|---|---|

Font:(Default) Times New Roman, Not Highlight

| Page 12: [5] Formatted | Microsoft Office User | 5/13/16 4:17 PM |
|---|---|---|

Font:(Default) Times New Roman, Not Highlight

| Page 12: [5] Formatted | Microsoft Office User | 5/13/16 4:17 PM |
|---|---|---|

Font:(Default) Times New Roman, Not Highlight

| Page 12: [5] Formatted | Microsoft Office User | 5/13/16 4:17 PM |
|---|---|---|

Font:(Default) Times New Roman, Not Highlight

| Page 12: [5] Formatted | Microsoft Office User | 5/13/16 4:17 PM |
|---|---|---|

Font:(Default) Times New Roman, Not Highlight

| Page 12: [5] Formatted | Microsoft Office User | 5/13/16 4:17 PM |
|---|---|---|

Font:(Default) Times New Roman, Not Highlight

| Page 12: [5] Formatted | Microsoft Office User | 5/13/16 4:17 PM |
|---|---|---|

Font:(Default) Times New Roman, Not Highlight

| Page 12: [5] Formatted | Microsoft Office User | 5/13/16 4:17 PM |
|---|---|---|

Font:(Default) Times New Roman, Not Highlight

| Page 12: [5] Formatted | Microsoft Office User | 5/13/16 4:17 PM |
|---|---|---|

Font:(Default) Times New Roman, Not Highlight

| Page 12: [5] Formatted | Microsoft Office User | 5/13/16 4:17 PM |
|---|---|---|

Font:(Default) Times New Roman, Not Highlight

| Page 12: [6] Formatted | Microsoft Office User | 5/13/16 4:19 PM |
|---|---|---|

Font:(Default) Times New Roman

| Page 12: [6] Formatted | Microsoft Office User | 5/13/16 4:19 PM |
|---|---|---|

Font:(Default) Times New Roman

| Page 12: [6] Formatted | Microsoft Office User | 5/13/16 4:19 PM |
|---|---|---|

Font:(Default) Times New Roman

| Page 12: [6] Formatted | Microsoft Office User | 5/13/16 4:19 PM |
|---|---|---|

Font:(Default) Times New Roman

| Page 12: [6] Formatted | Microsoft Office User | 5/13/16 4:19 PM |

Font:(Default) Times New Roman

| Page 12: [6] Formatted | Microsoft Office User | 5/13/16 4:19 PM |

Font:(Default) Times New Roman

| Page 12: [6] Formatted | Microsoft Office User | 5/13/16 4:19 PM |

Font:(Default) Times New Roman

| Page 12: [6] Formatted | Microsoft Office User | 5/13/16 4:19 PM |

Font:(Default) Times New Roman

| Page 12: [6] Formatted | Microsoft Office User | 5/13/16 4:19 PM |

Font:(Default) Times New Roman

| Page 12: [6] Formatted | Microsoft Office User | 5/13/16 4:19 PM |

Font:(Default) Times New Roman

| Page 12: [6] Formatted | Microsoft Office User | 5/13/16 4:19 PM |

Font:(Default) Times New Roman

| Page 12: [6] Formatted | Microsoft Office User | 5/13/16 4:19 PM |

Font:(Default) Times New Roman

| Page 12: [6] Formatted | Microsoft Office User | 5/13/16 4:19 PM |

Font:(Default) Times New Roman

| Page 12: [6] Formatted | Microsoft Office User | 5/13/16 4:19 PM |

Font:(Default) Times New Roman

| Page 12: [6] Formatted | Microsoft Office User | 5/13/16 4:19 PM |

Font:(Default) Times New Roman

| Page 12: [6] Formatted | Microsoft Office User | 5/13/16 4:19 PM |

Font:(Default) Times New Roman

| Page 12: [6] Formatted | Microsoft Office User | 5/13/16 4:19 PM |

Font:(Default) Times New Roman

| Page 12: [6] Formatted | Microsoft Office User | 5/13/16 4:19 PM |

Font:(Default) Times New Roman

| Page 12: [6] Formatted | Microsoft Office User | 5/13/16 4:19 PM |

Font:(Default) Times New Roman

| Page 12: [6] Formatted | Microsoft Office User | 5/13/16 4:19 PM |

Font:(Default) Times New Roman

| Page 12: [6] Formatted | Microsoft Office User | 5/13/16 4:19 PM |

Font:(Default) Times New Roman

| | | |
|---|---|---|
| **Page 12: [6] Formatted** | **Microsoft Office User** | **5/13/16 4:19 PM** |

Font:(Default) Times New Roman

| | | |
|---|---|---|
| **Page 12: [6] Formatted** | **Microsoft Office User** | **5/13/16 4:19 PM** |

Font:(Default) Times New Roman

| | | |
|---|---|---|
| **Page 12: [6] Formatted** | **Microsoft Office User** | **5/13/16 4:19 PM** |

Font:(Default) Times New Roman

| | | |
|---|---|---|
| **Page 12: [7] Formatted** | **Microsoft Office User** | **5/13/16 4:19 PM** |

Font:(Default) Times New Roman

| | | |
|---|---|---|
| **Page 12: [7] Formatted** | **Microsoft Office User** | **5/13/16 4:19 PM** |

Font:(Default) Times New Roman

| | | |
|---|---|---|
| **Page 12: [7] Formatted** | **Microsoft Office User** | **5/13/16 4:19 PM** |

Font:(Default) Times New Roman

| | | |
|---|---|---|
| **Page 12: [7] Formatted** | **Microsoft Office User** | **5/13/16 4:19 PM** |

Font:(Default) Times New Roman

| | | |
|---|---|---|
| **Page 12: [7] Formatted** | **Microsoft Office User** | **5/13/16 4:19 PM** |

Font:(Default) Times New Roman

| | | |
|---|---|---|
| **Page 12: [8] Deleted** | **Microsoft Office User** | **5/11/16 10:00 AM** |

 model according to the information criteria AIC, t

| | | |
|---|---|---|
| **Page 12: [8] Deleted** | **Microsoft Office User** | **5/11/16 10:00 AM** |

 model according to the information criteria AIC, t

| | | |
|---|---|---|
| **Page 12: [9] Formatted** | **Microsoft Office User** | **5/13/16 4:19 PM** |

Font:(Default) Times New Roman

| | | |
|---|---|---|
| **Page 12: [9] Formatted** | **Microsoft Office User** | **5/13/16 4:19 PM** |

Font:(Default) Times New Roman

| | | |
|---|---|---|
| **Page 12: [9] Formatted** | **Microsoft Office User** | **5/13/16 4:19 PM** |

Font:(Default) Times New Roman

| | | |
|---|---|---|
| **Page 12: [10] Formatted** | **Microsoft Office User** | **5/13/16 4:36 PM** |

Font:(Default) Times New Roman

| | | |
|---|---|---|
| **Page 12: [10] Formatted** | **Microsoft Office User** | **5/13/16 4:36 PM** |

Font:(Default) Times New Roman

| | | |
|---|---|---|
| **Page 12: [10] Formatted** | **Microsoft Office User** | **5/13/16 4:36 PM** |

Font:(Default) Times New Roman

| | | |
|---|---|---|
| **Page 12: [10] Formatted** | **Microsoft Office User** | **5/13/16 4:36 PM** |

Font:(Default) Times New Roman

| | | |
|---|---|---|
| **Page 12: [11] Formatted** | **Microsoft Office User** | **5/13/16 4:38 PM** |

Font:(Default) Times New Roman

| Page 12: [11] Formatted | Microsoft Office User | 5/13/16 4:38 PM |

Font:(Default) Times New Roman

| Page 16: [12] Deleted | Microsoft Office User | 5/13/16 4:48 PM |

In the Weddell Sea, expansion of the overall ice cover is only statistically significant during the autumn months (MAM) [e.g. *Turner et al.*, 2015]. During this time-period, both algorithms agree on statistically significant positive trends in the pack ice area, that extend through May for the NT algorithm ($p<0.05$) and through June for the BT algorithm ($p<0.05$). Statistically significant trends are also seen during March in the MIZ and polynya area ($p<0.05$), with larger trends in the NT algorithm ($p<0.01$). Thus, overall expansion of sea ice in the Weddell during autumn is in part driven by expansion of the MIZ early in the season, after which it is controlled by further expansion of the consolidated pack.

| Page 16: [13] Deleted | Microsoft Office User | 5/14/16 6:20 PM |

Furthermore, accurately mapping the extent of the MIZ from coarse resolution satellite data such as that from passive microwave sensors remains problematic. The MIZ is very dynamic in space and time, making it challenging to provide precise delimitations using sea ice concentrations that are in turn sensitive to melt processes and surface conditions. NeverthelessFurthermore, accurately mapping the extent of the MIZ from coarse resolution satellite data such as that from passive microwave sensors remains problematic. The MIZ is very dynamic in space and time, making it challenging to provide precise delimitations using sea ice concentrations that are in turn sensitive to melt processes and surface conditions.

---

## Author Comment (AC2) · 17 May 2016

The comment was uploaded in the form of a supplement:
http://www.the-cryosphere-discuss.net/tc-2016-26/tc-2016-26-AC2-supplement.pdf
* * *

---

## Author Comment (AC3) · 17 May 2016

**Mapping and Assessing Variability in the Antarctic Marginal Ice Zone, the Pack Ice and Coastal Polynyas in two Sea Ice Algorithms with implications on Breeding Success of Snow Petrels**

Julienne C. Stroeve[1,2], Stephanie Jenouvrier[3,4], G. Garrett Campbell[1], Christophe Barbraud[4] and Karine Delord[4]

[1]National Snow and Ice Data Center, Cooperative Institute for Research in Environmental Sciences, University of Colorado, Boulder, CO, USA

[2]Center for Polar Observation and Modelling, University College London, London, UK

[3]Woods Hole Oceanographic Institution, Woods Hole, MA, USA

[4]Centre d'Etudes Biologiques de Chizé, UMR 7372 CNRS, 79360 Villiers en Bois, France

**Abstract**

Sea ice variability within the marginal ice zone (MIZ) and polynyas plays an important role for phytoplankton productivity and krill abundance. Therefore, mapping their spatial extent, seasonal and interannual variability is essential for understanding how current and future changes in these biologically active regions may impact the Antarctic marine ecosystem. Knowledge of the distribution of MIZ, consolidated pack ice and coastal polynyas to the total Antarctic sea ice cover may also help to shed light on the factors contributing towards recent expansion of the Antarctic ice cover in some regions and contraction in others. The long-term passive microwave satellite data record provides the longest and most consistent record for assessing the proportion of the sea ice cover that is covered by each of these ice categories. However, estimates of the amount of MIZ, consolidated pack ice and polynyas depends strongly on what sea ice algorithm is used. This study uses two popular passive microwave sea ice algorithms, the NASA Team and Bootstrap, and applies the same thresholds to the sea ice concentrations to evaluate the distribution and variability in the MIZ, the consolidated pack ice and coastal polynyas. Results reveal that the seasonal cycle in the MIZ and pack ice is generally similar between both algorithms, yet the NASA Team algorithm has on average twice the MIZ and half the consolidated pack ice area as the Bootstrap algorithm. Trends also differ, with the Bootstrap algorithm suggesting statistically significant trends towards increased pack ice area and no statistically significant trends in the MIZ. The NASA Team algorithm on the other hand indicates statistically significant positive trends in the MIZ during spring. Potential coastal polynya area and broken ice within the consolidated ice pack is also larger in the NASA Team algorithm. The timing of maximum polynya area may differ by as much as 5 months between algorithms. These differences lead to different relationships between sea ice characteristics and biological processes, as illustrated here with the breeding success of an Antarctic seabird.

**1. Introduction**

Changes in the amount of the ocean surface covered by sea ice play an important role in the global climate system. For one, sea ice and its snow cover have a high surface reflectivity, or albedo, reflecting the majority of the sun's energy back to space. This helps to keep the polar regions cool and moderates the global climate. When sea ice melts or retreats, the darker (lower
albedo) ocean is exposed, allowing the ocean to absorb solar energy and warm, which in turn
melts more ice, creating a positive feedback loop. During winter, sea ice helps to insulate the
ocean from the cold atmosphere, influencing the exchange of heat and moisture to the
atmosphere with impacts on cloud cover, pressure distribution and precipitation. These in turn
can lead to large-scale atmospheric changes, affecting global weather patterns [e.g. *Jaiser et al.*,
2012]. Sea ice also has important implications for the entire polar marine ecosystem, including
sea ice algae, phytoplankton, crustaceans, fish, seabirds, and marine mammals, all of which
depend on the seasonal cycle of ice formation in winter and ice melt in summer. For example,
sea ice melt stratifies the water column, producing optimal light conditions for stimulating bloom
conditions. Antarctic sea birds rely upon the phytoplankton bloom for their breeding success and
survival [e.g. *Park et al.*, 1999].
In stark contrast to the Arctic, which is undergoing a period of accelerated ice loss [e.g.
*Stroeve et al.*, 2012; *Serreze and Stroeve*, 2015], the Antarctic is witnessing a modest increase in
total sea ice extent [*Parkinson and Cavalieri*, 2012; *Simmonds et al.*, 2015]. Sea ice around
Antarctica reached another record high extent in September 2014, recording a maximum extent
of more than 20 million km$^2$ for the first time since the modern passive microwave satellite data
record began in October 1978. This follows previous record maxima in 2012 and 2013 [*Reid et
al.*, 2015], resulting in an overall increase in Antarctic September sea ice extent of 1.1% per
decade since 1979. While the observed increase is statistically significant, Antarctic's sea ice
extent (SIE) is also highly variable from year to year and region to region [e.g. *Maksym et al.*,
2012; *Parkinson and Cavalieri*, 2012; *Stammerjohn et al.*, 2012]. For example, around the West
Antarctic Peninsula (WAP), there have been large decreases in sea ice extent and sea ice duration
[e.g. *Ducklow et al.*, 2012; *Smith and Stammerjohn*, 2001], coinciding with rapid warming since
1950 [*Ducklow et al.*, 2012].

[revised manuscript text omitted]
record (CDR) [*Meier et al.,* 2013].
Using these SIC fields, we define six binary categories of sea ice based on different SIC
thresholds [**Table 1**]. Because the marginal ice zone is highly dynamic in time and space, it is
difficult to precisely define this region of the ice cover. *Wadhams* [1986] defined the MIZ as that
part of the ice cover close enough to the open ocean boundary to be impacted by its presence,
e.g. by waves. Thus the MIZ is typically defined as the part of the sea ice that is close enough to
the open ocean to be heavily influenced by waves, and it extends from the open ocean to the
dense pack ice. In this study, we define the MIZ as extending from the outer sea ice/open ocean
boundary (defined by SIC $\geq$ 0.15 ice fraction) to the boundary of the consolidated pack ice
(defined by SIC = 0.80). This definition was previously used by *Strong and Rigor* [2013] to
assess MIZ changes in the Arctic and matches the upper SIC limit used by the National Ice
Center in mapping the Arctic MIZ. The consolidated ice pack is then defined as the area south of
the MIZ with ice fractions between $0.80 \leq$ SIC $\leq 1.0$. Potential coastal polynyas are defined as
regions near the coast that have SIC < 0.80.
To automate the mapping of different ice categories, radial transects from 50 to 90S are
individually selected to construct one-dimensional profiles [**Figure 1**]. The algorithm first steps
from the outer edge until the 0.15 SIC is detected, providing the latitude of the outer MIZ edge.
Next, the algorithm steps from the outer MIZ edge until either the 0.80 SIC is encountered, or the
continent is reached. Data points along the transect between these SIC thresholds are flagged as
the MIZ. In this way, the MIZ includes an outer band of low sea ice concentrations that
surrounds a band of inner consolidated pack ice, *but* sometimes the MIZ also extends all the way
to the Antarctic coastline (as sometimes observed in summer). South of the MIZ, the
consolidated ice pack ($0.80 \leq$ SIC $\leq 1.0$) is encountered; however, low sea ice concentrations can
appear near the coast inside the pack ice region as well. These are areas of potential coastal
polynyas. While it is difficult to measure the fine scale location of a polynya at 25km spatial
resolution, the lower sea ice concentrations provide an indication of some open water near the
coast, which for sea birds provides a source of open water for foraging. We have previously
tested mapping polynyas using a SIC threshold of 0.75 and 0.85 for the NASA Team and
Bootstrap algorithms, respectively, and found that these thresholds provided consistent polynya
areas between the two algorithms and matched other estimates of the spatial distribution of
polynyas [see *Li et al.,* 2016]. However, for this study we chose just one threshold, a compromise between the two algorithms, so that we can better determine the sensitivity of using
the same threshold on polynya area and timing of formation.
Using our method of radial transects, the algorithm then steps from the coast northward and
flags pixels with < 0.80 SIC until a 0.80 SIC pixel appears and defines that region as a potential
coastal polynya. Within the consolidated pack ice (and away from the coast), it is also possible to
encounter instances where 0.15 < SIC < 0.80 or SIC < 0.15. These are flagged as open pack ice
and open water areas within the consolidated pack ice, respectively. Finally, an ocean mask
derived from climatology and distributed by NSIDC was applied to remove spurious ice
concentrations at the ice edge as a result of weather effects.
**Figure 2** shows sample images of the classification scheme as applied to the NASA Team
and Bootstrap algorithms on days 70 (March 11) and 273 (September 30), respectively, in 2013.
During the fall and winter months when the ice cover is expanding there is a well-established
consolidated pack ice region, surrounded by the outer MIZ. Coastal polynyas are also found
surrounding the continent in both algorithms. The BT algorithm tends to show a larger
consolidated ice pack than NT, particularly during the timing of maximum extent. During the
melt season there is mixing of low and high ice concentrations, leading to mixtures of different
categories, which is still seen to some extent in the March images. However, during March areas
of polynyas (green), open water (pink) and open pack ice (orange) appear to extend from the
coastline in some areas (e.g. southern Weddell and Ross seas). While any pixel with SIC < 0.8
adjacent to the coastal boundary is flagged as potential polynya when stepping northwards, if a
pixel is already flagged as MIZ or consolidated pack ice when stepping southwards, it remains
flagged as MIZ or pack ice. After that analysis, a check for pixels with SICs less than 0.8 is done
to flag for broken ice or open water. Thus, during these months (e.g. December to February or
March), the physical interpretation of the different ice classes may be less useful.
Using the binary classification scheme, daily gridded fields at each 25 km pixel are obtained.

[revised manuscript text omitted]

pack ice trends are spatially consistent between both algorithms, though not all trends are
statistically significant, particularly for the NT algorithm. The largest consistency occurs in the
the western Ross Sea, where positive trends are seen in both algorithms, statistically significant
from March to November (p<0.01) in the BT algorithm, and from January to July and October to
November in the NT algorithm. Note also that both algorithms show statistically significant
positive trends in the MIZ from January to March in the western Ross Sea and generally negative
trends in the eastern Ross Sea. This pattern switches from June to December, with mostly
negative MIZ trends in the western Ross Sea and positive trends in the eastern Ross Sea. In
particular, the statistically significant positive trends in the MIZ in the NT algorithm occur at the
time of year with the largest overall trends in the SIE in this region. This would suggest perhaps
different interpretation of processes impacting the overall ice expansion in the Ross Sea
depending on which algorithm is used.
In the B/A Sea, statistically significant positive trends in pack ice are limited to May through
August in the NT algorithm and June and July in the BT algorithm. The positive NT pack ice
trends are offset by negative trends in the NT MIZ. Both algorithms exhibit negative pack ice
trends during other months that are consistent between the algorithms, though larger in
magnitude for the BT algorithm. This is generally compensated by statistically significant
negative trends in the NT MIZ to give an overall negative decline of total extent.
Trends in the pack ice are also consistent between algorithms in the Weddell Sea, with
statistically significant trends generally occurring at the same longitude and during the same
months. The positive pack ice trends in MAM (NT) or MAMJ (BT) are confined to a very
narrow longitude band which moves to the east with progressing season. Then in June, and
continuing for several months, negative pack ice trends occur. For both algorithms, trends in the
MIZ are generally not statistically significant, except for some positive trends in the eastern

Weddell Sea from January to March and negative trends mostly from June to November near
330 degrees longitude.
Finally, in the Pacific and Indian Oceans we again see spatial consistency in pack ice and
MIZ trends for both algorithms, with generally larger (smaller) pack ice (MIZ) trends for the BT
algorithm, though trends are closer in magnitude in the Pacific sector from March to July. Pack
ice trends are generally positive, more in BT than NT and trends in MIZ extent basically vary
around zero with exceptions during August through December in both algorithms in the Pacific
Ocean.
In summary, while the magnitude of trends differs between both algorithms, there is general
spatial consistency in the patterns of positive and negative trends in the consolidated pack ice
and the MIZ. Results suggest that positive trends in total SIE are generally a result of statistically
significant positive trends in the consolidated pack ice in the BT algorithm in all sectors of the
Antarctic, except for the Bellingshausen/Amundsen Sea sector and the Weddell Sea during ice
retreat. The NT algorithm on the other hand suggests more instances of statistically significant
positive trends in the MIZ, though this is highly regionally dependent.

### 3.2.3 Seasonal Trends in MIZ and Pack Ice Width
Finally, we compute the overall width of the MIZ and pack ice following *Strong and Rigor*
[2013] and produce seasonal means. Briefly, following the classification of each ice category,
latitude boundaries are computed for each longitude and each day. These are averaged for each
month to provide monthly mean latitude boundaries at each longitude. The boundaries are
subsequently converted to width in km, and averaged for all longitudes. Finally, seasonal means
are derived.
Time-series of seasonal means of the circumpolar MIZ width and pack ice width are shown
in **Figure 10** for all seasons except summer when the results are noisy. As we may expect
following the previous results, the NT MIZ width is larger and the pack ice width is smaller than
the seen in the BT algorithm. During autumn (MAM) however, the differences in widths for both
the MIZ and the pack ice between the algorithms are largely reduced compared to the other
seasons. For example the difference in 1979-2014 pack ice width between the algorithms during
MAM is 60 km, 121 km in JJA and 139 km in SON. Similarly, the long-term mean MIZ width
differences are 54 km (MAM), 74 km (JJA) and 83 km (SON). In addition, during autumn,
trends in the MIZ and pack ice are largely consistent between the two algorithms, with no trend
in the MIZ and increases in the pack ice on the order of 21.2 km dec$^{-1}$ and 20.0 km dec$^{-1}$
(p<0.01) for the BT and NT algorithms, respectively. This is the season with the largest trends in
the pack ice width, representing a 21% widening over the satellite record.
During winter (JJA) and spring (SON) however, the NT and BT algorithms exhibit opposing
trends in the MIZ with the NT algorithm indicating an increase, and the BT a decrease. The
largest positive trend in the MIZ width occurs during spring at a rate of +10.3 km dec$^{-1}$ (p<0.01)
in the NT algorithm, indicating a 6% widening since 1979. This widening is a result of the MIZ
moving slightly equatorward rather than expanding southwards (as also seen in Figure 7).
However, this is an increase of only about 1 to 1.5 grid cells over the entire data record, and
despite a statistically significant trend, there remains substantial interannual variability in the
SON MIZ width, with the maximum width recorded in 2003 (310 km) and the minimum in 1985
(217 km), with a mean SON MIZ width of 248 km. The trend during winter is considerably
smaller at +2.7 km dec$^{-1}$, as a result of expansion both equatorward and southwards, yet it is not
statistically significant.

For the pack ice, both sea ice algorithms show statistically significant positive trends towards increased width of the pack ice, which are also nearly identical during winter at +18.7 and +18.1 km dec$^{-1}$ (p<0.01) for the BT and NT algorithms, respectively. This represents a widening of the pack ice of approximately 11% from 1979 to 2014 during winter. As one may expect, differences in the pack ice width between the algorithms are largely found in spring as a result of the MIZ expanding in the NT algorithm. Therefore, during SON the trends in the width of the NT pack ice are smaller, with trends of +10.0 (p<0.05) km dec$^{-1}$ compared to +16.7 (p<0.01) for the BT algorithm.

Finally it is important to point out that the interannual variability in the pack ice is similar between both data sets despite differences in magnitude. Correlations between the two algorithms are: 0.96 (MAM), 0.92 (JJA) and 0.77 (SON). The reason for the weaker correlation in SON is not entirely clear. For the MIZ, interannual variability is generally about twice as large in the NASA Team algorithm and the two data sets are not highly correlated except for autumn, with correlations of 0.67 (MAM), 0.39 (JJA) and 0.43 (SON).

**4. Implications for a Seabird**

Here we use data on the MIZ and the consolidated ice pack from both algorithms to understand the role of sea ice habitat on breeding success of a seabird, the snow petrel *Pagodroma nivea*. As mentioned in the introduction, the MIZ is a biologically important region because it is an area of high productivity and provides access to food resources needed by seabirds [*Ainley et al.*, 1992]. During winter, productivity is reduced at the surface in open water, while it is concentrated within the ice habitat, especially within the ice floes [*Ainley et al.*, 1986]. This patchy distribution of food availability within the MIZ and pack ice provides feeding opportunities for seabirds such as the snow petrel. Observations suggest that the snow petrel forages more successfully in areas close to the ice edge and within the MIZ than in consolidated ice conditions [*Ainley et al.*, 1984, 1992].

Breeding success of snow petrels depends on sufficient body condition of the females, which in part reflects favorable environmental and foraging conditions prior to the breeding season. Indeed, female snow petrels in poor early body condition are not able to build up the necessary body reserves for successful breeding [*Barbraud and Chastel*, 1999]. Breeding success was found to be higher during years with extensive sea ice cover during the preceding winter [*Barbraud and Weimerskirch*, 2001]. This is in part because winters with extensive sea ice are associated with higher krill abundance the following summer [*Flores et al.*, 2012; *Loeb et al.*, 1997; *Atkinson et al.*, 2004], thereby increasing the resource availability during the breeding season. However, extensive winter sea ice may protect the under ice community from predation and thus reduce food availability, in turn affecting breeding success [*Olivier et al.*, 2005]. By distinguishing between the areas of MIZ and pack ice, we can expect a better understanding of the role of sea ice on food availability and hence breeding success of snow petrels.

In the following, we expect that an extensive pack ice during winter may reduce breeding success the following breeding season by protecting the under ice community from predation, while an extensive MIZ may increase breeding success by providing easier access to foraging. With the classifications as defined by both algorithms we calculated the MIZ and pack ice area in a wide rectangular sector defined by the migration route of the snow petrel [*Delord et al.*, 2016] from April to September [see **Table 4** for latitude and longitude limits]. This is the first time that appropriate areas of the observed foraging range are used to study the carry over effect of winter conditions on the breeding performance of snow petrel, as this information did not
existed previously. Using these locations, we averaged the MIZ and pack ice extents over the
entire winter from April to September. We next employed a logistic regression approach to study
the effects of MIZ and pack ice area within this sector and evaluate the impacts on breeding
success the following summer. The response variable was the number of chicks $C_t$ in a breeding
season $t$, from 1979 to 2014 collected at Terre Adélie, Dumont D'Urville [*Barbraud and*
*Weimerskirch*, 2001, *Jenouvrier et al.*, 2005].
Effects of MIZ and pack ice area were analyzed using Generalized Linear Models (GLM)
with logit-link functions and binomial errors fitted in R using the package glm.
Specifically, the response variable is the number of chicks $C_t$ in a breeding season $t$, from 1979 to
2014 collected at Terre Adelie, Dumont D'Urville [*Barbraud and Weimerskirch*, 2001,
*Jenouvrier et al.*, 2005]. It follows a binomial distribution, such that $C_t \sim Bin(\mu_t, N_t)$, where $N_t$ is
the number of breeding pairs and $\mu_t$ is the breeding success in year $t$. The breeding success is a
function of the MIZ and pack ice covariates at time $t$ (COV) such as:

$$\mu_t = \beta_0 + \beta_1 COV_{(t)}$$

To select the covariate that most impacts the breeding success of snow petrels, we applied the
information-theoretic (I-T) approaches [*Burnham et al.*, 2011]. This is based on quantitative
measures of the strength of evidence for each hypothesis (Hi) rather than on "testing" null
hypotheses based on test statistics and their associated P values. To quantify the strength of
evidence for each hypothesis (Hi) – here the effect of each covariate on the breeding success-
we used the common criteria AIC (the Akaike's Information Criteria), where AIC = - 2 log(L) +
2K [*Akaike,* 1973]. The term, -2 log(L), is the "deviance" of the model, with log(L) the
maximized log-likelihood and K the total number of estimable parameters in the model. The
chosen model is the one that minimizes the AIC, in orther words, minimizes the Kullback-
Leibler distance between the model and truth. The ability of two models to describe the data was
assumed to be "not different" if the difference in their AIC was < 2 [*Burnham and Anderson*,
2002]. Note the AIC is a way of selecting a model from a set of models based on information
theory [*Burnham and Anderson*, 2002], and is largely used in biological sciences. While non-
linear models may be more appropriate as ecological system relationships are likely more
complex than linear relationships, without *a priori* knowledge of the mechanisms that could lead
to such non-linear relationships, it is extremely difficult to set meaningful hypothesis to be
included in the model selection.
**Table 5** summarizes model selection. The model with the lowest AIC (highlighted in gray)
suggests the BT pack ice as a sea ice covariate. If AIC are sorted from lowest to highest value,
the next model includes the sea ice covariate MIZ calculated with the NASA algorithm.
However, it shows a ΔAIC ~8 from the best model, and thus the NT MIZ is not well supported
by the data in comparison to the best model. The relationship between BT pack ice and breeding
success is negative [**Figure 11**]. In other words, a more extensive consolidated pack ice during
winter tends to reduce breeding success the following summer by limiting foraging
opportunities. The effect of the MIZ however was uncertain, contrary to what one may expect
given the increased opportunities for foraging within the MIZ. However, if we had only used ice
classifications based on the NASA Team algorithm, the model with the lowest AIC would have
suggested an importance of the MIZ. We would have then concluded a negative effect of the
MIZ on the breeding success of snow petrels, contrary to what one may expect given that the
MIZ is the main feeding habitat of the species. By using both algorithms, we instead conclude that the breeding success of snow petrels is negatively affected by the pack ice area as calculated
with the Bootstrap algorithm.

**5. Discussion**

While the main purpose for doing the classification of different ice categories is for
interdisciplinary studies of sea bird breeding success, the results may also be useful for
attribution of the observed sea ice changes. The positive trends in Antarctic sea ice extent are
currently poorly understood and are at odds with climate model forecasts that suggest the sea ice
should be declining in response to increasing greenhouse gases and stratospheric ozone depletion
[e.g. *Turner et al.*, 2013; *Bitz and Polvani*, 2012; *Sigmond and Fyfe*, 2010]. However, several
modeling studies, such as those used in the phase 5 Coupled Model Intercomparison Project
(CMIP5), have suggested that the sea ice increase over the last 36 years remains within the range
of intrinsic of internal variability [e.g. *Bitz and Polvani*, 2012; *Turner et al.*, 2013; *Mahlstein et
al.*, 2013; *Polvani and Smith,* 2013; *Swart and Fyfe*, 2013]. Earlier satellite from the 1960s and
1970s and from ship observations suggest periods of high and low sea ice extent, and thus high
natural variability [*Meier et al.*, 2013; *Gallaher et al.*, 2014]. Further evidence comes from ice
core climate records, which suggest that the climate variability observed in the Antarctic during
the last 50 years remains within the range of natural variability seen over the last several hundred
to thousands of years [*Thomas et al.*, 2013; *Steig et al.*, 2013]. Thus, we may require much
longer records to properly assess Antarctic sea ice trends in contrast to the Arctic, where negative
trends are outside the range of natural variability and are consistent with those simulated from
climate models.

[revised manuscript text omitted]

**Conclusions**

Antarctic sea ice plays an important role in the polar marine ecosystem. While total Antarctic
sea ice cover is expanding in response to atmospheric and oceanic variability that remains to be
fully understood, one may expect that these increases would also be manifested in either
equatorward progression of the MIZ or the consolidated pack ice or both, that in turn would
impact the entire trophic web, from primary productivity, to top predator species, such as
seabirds. In this study we identified several different ice categories using two different sets of
passive microwave sea ice concentration data sets. The algorithms are in agreement as to the
location of the northern edge of the total sea ice cover, but differ in regards to how much of the
ice cover consists of the marginal ice zone, the consolidated ice pack, the size of potential
polynyas as well as the amount of broken ice and open water within the consolidated ice pack.
Here we use sea ice concentration thresholds of $0.15 \leq SIC < 0.80$ to define the width of the MIZ
and $0.80 \leq SIC \leq 1.0$ to define the consolidated pack ice. Yet applying the same thresholds for
both sea ice algorithms results in a MIZ from the NASA Team algorithm that is on average twice
as large as in the Bootstrap algorithm and considerably more broken ice within the consolidated
pack ice. Total potential coastal polynya areas ($SIC \leq 0.80$) also differ between the algorithms,
though differences are generally smaller than for the MIZ and the consolidated pack ice. While
we do not precisely resolve polynyas, these potential coastal polynyas (i.e. open water areas near
the coast) are important foraging sites for sea birds.
While the spatial extents of the different ice classes may differ, the seasonal cycle is
generally consistent between both algorithms. Climatologically, the advance of the consolidated
ice pack happens over a much longer period (~7-8 months) than the retreat (~4-5 months), while
the MIZ exhibits a longer advance period (~8-10 months). This seasonal cycle in
expansion/contraction of the ice cover is in general agreement with results by *Stammerjohn et al.*
[2008] who showed sea ice retreat begins in September at the outer most edge of the sea ice and
continues poleward over the next several months. However, what these results show is that while
the pack ice starts to retreat around September, this in turn results in a further expansion of the
MIZ, the amount of which is highly dependent on which algorithm is used. The timing of when
the maximum polynya extent is reached however can differ by several months between the
algorithms in regions such as the Bellingshausen/Amundsen Sea and the Pacific Ocean.
Since the MIZ is an important region for phytoplankton biomass and productivity [e.g. *Park*
*et al.,* 1999], mapping seasonal and interannual changes in the MIZ is important for
understanding changes in top predator populations and distributions. However, as we show in
this study, results are highly dependent on which sea ice algorithm is used for delineating the
MIZ, which may result in different conclusions when using this data in ecosystem models. To hightlight this sensitivity, we examined the impact the winter MIZ and consolidated pack ice
area as derived from both algorithms would have on the breeding success of snow petrels the
following summer. The different proportions of MIZ and consolidated pack ice between
algorithms affected the inferences made from models tested even if trends were of the same sign.
Given the sensitivity of the relationships between the consolidated pack ice/MIZ and breeding
success of this species, caution is warranted when doing this type of analysis as different
relationships may emerge as a function of which sea ice data set is used in the analysis. Further
work is needed to validate the accuracy of the distribution of the MIZ and consolidated pack ice
from passive microwave so that the data will be more useful for future biological and ecosystem
studies.

**Acknowledgements**
This work is funded under NASA Grant NNX14AH74G and NSF Grant PLR 1341548. We are
grateful to Sharon Stammerjohn for her helpful comments on the manuscript. Gridded fields of
the different ice classifications from both algorithms are available via ftp by contacting J.
Stroeve. We thank all the wintering fieldworkers involved in the collection of snow petrel data at
Dumont d'Urville since more than 50 years, as well as Institut Paul Emile Victor (program IPEV
n°109, resp. H. Weimerskirch), Terres Australes et Antarctiques Françaises and Zone Atelier
Antarctique (CNRS-INEE) for support.

**Tables**

**Table 1.** Sea ice categories defined in this study.

[revised manuscript text omitted]

**Table 5.** Results of model selection for the relationship between pack ice and MIZ on breeding
success of snow petrel. The model with the lowest AIC is highlighted in gray. AIC scores are
often interpreted as difference between the best model (smallest AIC) and each model referred as
$\Delta$AIC. According to information theory, models with $\Delta$AIC < 2 are both likely [*Burnham and*
*Anderson*, 2002] but if a model shows a $\Delta$AIC > 4 it is unlikely in comparison with the best
model (smallest AIC).

[revised manuscript text omitted]

---

## Author Response (AR2)

**Suggestions for revision or reasons for rejection (will be published if the paper is accepted for final publication)**

Review of

Mapping and Assessing Variability in the Antarctic Marginal Ice Zone, the Pack Ice and Coastal Polynyas in two Sea Ice Algorithms with implications on Breeding Success of Snow Petrels - Revision 1

by Stroeve, J. C., et al.

This is revision #1 of the original manusript. The authors have improved the manuscript substantially and I appreciate that many of the reviewers' comments have been taken seriously and discussed properly. Thank you!

*Thank you for your positive and thorough comments and we agree that the paper has been strengthened thanks to the reviewers comments.*

The paper is ready to go - pending a few minor edits which I list below but which do not require further attention from my side.

L103-L105: I agree that the katabatic winds are responsible for generation of coastal polynyas and for keeping them open by more or less constantly advecting the new ice formed to the leeward side of the polynya. However, I am wondering how far away from the coast the sea ice still "feels" the impact of the katabatic winds which I would expect to loose their influence within a few 10 kilometers from the coast with the synoptic winds taking over. I am wondering what the authors' reflection on this is and whether this sentence isn't perhaps misleading a bit.

*This is an interesting question that is difficult to give a precise answer for. But we agree that coastal polynyas, while formed by katabatic winds and/or ocean upwelling may also be influenced by meridional winds that in turn drive the increase in extent. Comiso and Gordon 1988 discuss how years with peak polynya extents in the Weddell Sea are also years with peak sea ice extent. We slightly changed the sentence to state instead of "thus" to be "in part", included mention of the Weddell Sea and the Comiso and Gordon reference.*
*We also added the following sentence: However, few coastal polynyas are solely a result of katabatic outflow: topography, bathymetry and winds also play a large role [Massom et al., 1998].*

L122: The authors stated in their rebuttal letter that they could not find the TC reference for Ivanova et al. 2015. Here it is: Ivanova, N., Pedersen, L. T., Tonboe, R. T., Kern, S., Heygster, G., Lavergne, T., Sorensen, A., Saldo, R., Dybkjaer, G., Brucker, L., and Shokr, M.: Satellite passive microwave measurements of sea ice concentration: an optimal algorithm and challenges, The Cryosphere, 9, 1797-1817, doi:10.5194/tc-9-1797-2015, 2015.

*Thank you we have found it and referenced it.*

L149-161: I see that the authors have kept their definition of the MIZ. While I am fine with that, because a change of this definition would have meant to redo the study, I would have hoped to see two notes: i) waves often penetrate well beyond the 80% sea-ice concentration isoline (they break of consolidated sea ice). ii) the sea-ice type along most of the Antarctic sea-ice cover is pancake ice which differs from the Arctic.

*Done*

I did not find a notion about how to define the sea-ice concentration threshold to delineate polynyas in Strong and Rigor (2013). Therefore I feel that the authors also could have underlined the choice of SIC < 0.80 for defining the ice class "potential coastal polynya" with a reference as suggested in my previous review (e.g. Massom et al. 1998, Annals of Glaciology, 27, 420-426)

*As we mentioned we also have compared using 0.75 and 0.85 thresholds in another new paper (Li et al., 2016). We now also mention the Massom et al. paper but do so in the following paragraph as it fits better there.*

L203-211: The authors are using data on the NSIDC polar-stereographic grid which is not a true-area grid. How did the authors compute the grid-cell area? Did they take the readily available grid-cell area data files from NSIDC? I suggest to mention this in the manuscript. A reader taking your manuscript as a model to carry out a similar study might not want to run into biased ice-category extent estimates because of the latitude-dependent variation of the grid-cell area of the grid used.

*Done*

L362/363: Please check sentence: "... switch to positive in the while remaining ..."
*Done*

L375: "... as a function longitude ..." I guess an "of" is missing here.
*Done*

L392: I suggest to also write "Bellingshausen / Amundsen Sea" here, instead of "B/A Sea". The same applies to L625.
*Done*

I have one minor general editoral comment: The authors could check usage of a capital "S" in seas when they refer to two regions such as "Ross and Weddell Seas". Currently, this is written in an inconsistent way, sometimes with small "s", sometimes with capital "S".
*We replaced with small "s" when more than one Sea is being referred to. Only one instance was found.*

**Mapping and Assessing Variability in the Antarctic Marginal Ice**
**Zone, the Pack Ice and Coastal Polynyas in two Sea Ice Algorithms**
**with implications on Breeding Success of Snow Petrels**

Julienne C. Stroeve[1,2], Stephanie Jenouvrier[3,4], G. Garrett Campbell[1], Christophe Barbraud[4] and
Karine Delord[4]
[1]National Snow and Ice Data Center, Cooperative Institute for Research in Environmental
Sciences, University of Colorado, Boulder, CO, USA
[2]Center for Polar Observation and Modelling, University College London, London, UK
[3]Woods Hole Oceanographic Institution, Woods Hole, MA, USA
[4]Centre d'Etudes Biologiques de Chizé, UMR 7372 CNRS, 79360 Villiers en Bois, France

## Abstract
Sea ice variability within the marginal ice zone (MIZ) and polynyas plays an important role for
phytoplankton productivity and krill abundance. Therefore, mapping their spatial extent,
seasonal and interannual variability is essential for understanding how current and future changes
in these biologically active regions may impact the Antarctic marine ecosystem. Knowledge of
the distribution of MIZ, consolidated pack ice and coastal polynyas to the total Antarctic sea ice
cover may also help to shed light on the factors contributing towards recent expansion of the
Antarctic ice cover in some regions and contraction in others. The long-term passive microwave
satellite data record provides the longest and most consistent record for assessing the proportion
of the sea ice cover that is covered by each of these ice categories. However, estimates of the
amount of MIZ, consolidated pack ice and polynyas depends strongly on what sea ice algorithm
is used. This study uses two popular passive microwave sea ice algorithms, the NASA Team and
Bootstrap, and applies the same thresholds to the sea ice concentrations to evaluate the
distribution and variability in the MIZ, the consolidated pack ice and coastal polynyas. Results
reveal that the seasonal cycle in the MIZ and pack ice is generally similar between both
algorithms, yet the NASA Team algorithm has on average twice the MIZ and half the
consolidated pack ice area as the Bootstrap algorithm. Trends also differ, with the Bootstrap
algorithm suggesting statistically significant trends towards increased pack ice area and no
statistically significant trends in the MIZ. The NASA Team algorithm on the other hand
indicates statistically significant positive trends in the MIZ during spring. Potential coastal
polynya area and broken ice within the consolidated ice pack is also larger in the NASA Team
algorithm. The timing of maximum polynya area may differ by as much as 5 months between
algorithms. These differences lead to different relationships between sea ice characteristics and
biological processes, as illustrated here with the breeding success of an Antarctic seabird.

## 1. Introduction
Changes in the amount of the ocean surface covered by sea ice play an important role in the
global climate system. For one, sea ice and its snow cover have a high surface reflectivity, or
albedo, reflecting the majority of the sun's energy back to space. This helps to keep the polar regions cool and moderates the global climate. When sea ice melts or retreats, the darker (lower albedo) ocean is exposed, allowing the ocean to absorb solar energy and warm, which in turn melts more ice, creating a positive feedback loop. During winter, sea ice helps to insulate the ocean from the cold atmosphere, influencing the exchange of heat and moisture to the atmosphere with impacts on cloud cover, pressure distribution and precipitation. These in turn can lead to large-scale atmospheric changes, affecting global weather patterns [e.g. *Jaiser et al.*, 2012]. Sea ice also has important implications for the entire polar marine ecosystem, including sea ice algae, phytoplankton, crustaceans, fish, seabirds, and marine mammals, all of which depend on the seasonal cycle of ice formation in winter and ice melt in summer. For example, sea ice melt stratifies the water column, producing optimal light conditions for stimulating bloom conditions. Antarctic sea birds rely upon the phytoplankton bloom for their breeding success and survival [e.g. *Park et al.*, 1999].

In stark contrast to the Arctic, which is undergoing a period of accelerated ice loss [e.g. *Stroeve et al.*, 2012; *Serreze and Stroeve*, 2015], the Antarctic is witnessing a modest increase in total sea ice extent [*Parkinson and Cavalieri*, 2012; *Simmonds et al.*, 2015]. Sea ice around Antarctica reached another record high extent in September 2014, recording a maximum extent of more than 20 million km$^2$ for the first time since the modern passive microwave satellite data record began in October 1978. This follows previous record maxima in 2012 and 2013 [*Reid et al.*, 2015], resulting in an overall increase in Antarctic September sea ice extent of 1.1% per decade since 1979. While the observed increase is statistically significant, Antarctic's sea ice extent (SIE) is also highly variable from year to year and region to region [e.g. *Maksym et al.*, 2012; *Parkinson and Cavalieri*, 2012; *Stammerjohn et al.*, 2012]. For example, around the West Antarctic Peninsula (WAP), there have been large decreases in sea ice extent and sea ice duration [e.g. *Ducklow et al.*, 2012; *Smith and Stammerjohn*, 2001], coinciding with rapid warming since 1950 [*Ducklow et al.*, 2012].

The temporal variability of the circumpolar Antarctic sea ice extent is underscored by sea ice conditions in 2015 when the winter ice cover returned back to the 1981-2010 long-term mean. Also, recent sea ice assessments from early satellite images from the Nimbus program of the late 1960s indicate similarly high but variable SIE as observed over 2012-2014 [*Meier et al.,* 2013; *Gallaher et al.,* 2014]. Mapping of the September 1964 ice edge indicates that ice extent likely exceeded both the 2012 and 2013 record monthly-average maxima, at 19.7±0.3 million km$^2$. This was followed in August 1966 by an extent estimated at 15.9±0.3 million km$^2$, considerably smaller than the record low maximum extent of the modern satellite record (set in 1986). The circumpolar average also hides contrasting regional variability, with some regions showing either strong positive or negative trends with magnitudes equivalent to those observed in the Arctic [*Stammerjohn et al.*, 2012]. In short, interannual and regional variability in Antarctic sea ice is considerable, and while the current positive trend in circumpolar averaged Antarctic sea ice extent is important, it is not unprecedented compared to observations from the 1960s and it is not regionally distributed.

Several explanations have been put forward to explain the positive Antarctic sea ice trends. Studies point to anomalous short-term wind patterns that both grow and spread out the ice, related to the strength of the Amundsen Sea low pressure [e.g. *Turner et al.,* 2013; *Reid et al.*, 2015; *Holland and Kwok*, 2012]. Other studies suggest melt water from the underside of floating ice surrounding the continent has risen to the surface and contributed to a slight freshening of the surface ocean [e.g. *Bintanja et al.*, 2013]. While these studies have helped to better understand how the ice, ocean and atmosphere interact, 2012 to 2014 showed different regions and seasons contributing to the net positive sea ice extent, which has made it difficult to establish clear links
and suggests that no one mechanism can explain the overall increase.
While the reasons for the increases in total extent remain poorly understood, it is likely that
these changes are not just impacting total sea ice extent but also the distribution of pack ice, the
marginal ice zone (MIZ) and polynyas. The MIZ is a highly dynamic region of the ice cover,
defined by the transition between the open ocean and the consolidated pack ice. In the Antarctic,
wave action penetrates hundreds of kilometers into the ice pack, resulting in small rounded ice
floes from wave-induced fracture [*Kohout et al.,* 2014]. This in turn makes the MIZ region
particularly sensitive to both atmospheric and oceanic forcing, such that during quiescent
conditions, it may consist of a diffuse thin ice cover, with isolated thicker ice floes distributed
over a large (hundreds of kilometers) area. During high on-ice wind and wave events, the MIZ
region contracts to a compact ice edge with rafted ice pressed together in front of the solid ice
pack. The smaller the ice floes, the more mobile they are and large variability in ice conditions
can be found in response to changing wind and ocean conditions. Polynyas on the other hand are
open water areas near the continental margins [e.g. *Morales-Maqueda et al.*, 2004] that often
remain open as a result of strong katabatic winds flowing down the Antarctic plateau. The winds
continuously push the newly formed sea ice away from the continent, which in part influences
the outer ice edge as well, contributing to the overall increase in total ice extent in specific
regions around the Antarctic continent, such as within the Weddell Sea [*Comiso and Gordon,*
1988] where katabatic winds are persistent. However, few coastal polynyas are solely a result of
katabatic outflow: topography, bathymetry and winds also play a large role [*Massom et al.*,
1998].
Both polynyas and the MIZ are biologically important regions of the sea ice cover that have
implications for the entire trophic web, from primary productivity [*Yun et al.*, 2015], to top
predator species, such as seabirds. Near the ice edge and in the MIZ, the stable upper layer of the
water column is optimal for phytoplankton production [e.g. *Park et al.*, 1999]. This
phytoplankton bloom is subsequently exploited by zooplankton, with effects that cascade up to
fish, seabirds and marine mammals. Similarly, within polynyas there is a narrow opportunity for
phytoplankton growth, the timing of which plays an important role in both biogeochemical
cycles [*Smith and Barber*, 2007] and biological production [*Arrigo and van Dijken*, 2003; *Ainley*
*et al.*, 2010]. However, while studies have suggested that the timing of sea ice retreat is
synchronized with the timing of the phytoplankton bloom, other factors such as wind forcing
[*Chiswell*, 2011], thermal convection [*Ferrari et al.,* 2014] and iron availability [*Boyd et al*,
2007, and references therein] play important roles as well.
In this study we use the long-term passive microwave sea ice concentration data record to
evaluate variability and trends in the MIZ, the pack ice and polynyas from 1979 to 2014. A
complication arises however as to which sea ice algorithm to use. There are at least a dozen
algorithms available, spanning different time-periods, which give sea ice concentrations that are
not necessarily consistent with each other [*see Ivanova et al*., 2015; 2014 for more information].
To complicate matters, different studies have used different sea ice algorithms to examine sea ice
variability and attribution. For example, *Hobbs and Raphael* [2010] used the Had1SST1 sea ice
concentration data set [*Rayner et al.,* 2003], which is based on the NASA Team algorithm
[*Cavalieri et al.*, 1999], whereas *Raphael and Hobbs* [2014] relied on the Bootstrap algorithm
[*Comiso and Nishio*, 2008]. To examine the influence in the choice of sea ice algorithm on the
results, we use both the Bootstrap (BT) and NASA Team (NT) sea ice algorithms. Results are
evaluated hemispheric-wide and also for different regions. We then discuss the different implications resulting from the two different satellite estimates for biological impact studies. We
focus on the breeding success of snow petrels because seabirds have been identified as useful
indicators of the health and status of marine ecosystems [*Piatt and Sydeman*, 2007].

**2. Data and Methods**

To map different ice categories, the long-term passive microwave data record is used, which
spans several satellite missions, including the Scanning Multichannel Microwave Radiometer
(SMMR) on the Nimbus-7 satellite (October 1978 to August 1987), the Special Sensor
Microwave/Imager (SSM/I) sensors -F8 (July 1987 to December 1991), -F11 (December 1991 to
September 1995), -F13 (May 1995 to December 2007) and the Special Sensor Microwave
Imager/Sounder (SSMIS) sensor –F17 (January 2007- to present), both on the Defense
Meteorological Satellite Program's (DMSP) satellites. Derived sea ice concentrations (SICs)
from both the Bootstrap [*Comiso and Nishio*, 2008] and the NASA Team [*Gloersen et al.,* 1992;
*Cavalieri et al.,* 1999] are available from the National Snow and Ice Data Center (NSIDC) and
provide daily fields from October 1978 to present, gridded to a 25 km polar stereographic grid.
While a large variety of SIC algorithms are available, the lack of good validation has made it
difficult to determine which algorithm provides the most accurate results during all times of the
year and for all regions. Using two algorithms provides a consistency check on variability and
trends. Note that NSIDC has recently combined these two algorithms to build a climate data
record (CDR) [*Meier et al.,* 2013].
Using these SIC fields, we define six binary categories of sea ice based on different SIC
thresholds [**Table 1**]. Because the marginal ice zone is highly dynamic in time and space, it is
difficult to precisely define this region of the ice cover. *Wadhams* [1986] defined the MIZ as that
part of the ice cover close enough to the open ocean boundary to be impacted by its presence,
e.g. by waves. Thus the MIZ is typically defined as the part of the sea ice that is close enough to
the open ocean to be heavily influenced by waves, and it extends from the open ocean to the
dense pack ice. In this study, we define the MIZ as extending from the outer sea ice/open ocean
boundary (defined by SIC $\geq$ 0.15 ice fraction) to the boundary of the consolidated pack ice
(defined by SIC = 0.80). This definition was previously used by *Strong and Rigor* [2013] to
assess MIZ changes in the Arctic and matches the upper SIC limit used by the National Ice
Center in mapping the Arctic MIZ. However, we note that waves can penetrate well beyond the
80% SIC isoline and that the 80% SIC threshold may be different in the Arctic than the Antarctic
as the MIZ in the Antarctic largely consists of pancake ice. The consolidated ice pack is defined
as the area south of the MIZ with ice fractions between 0.80 $\leq$ SIC $\leq$ 1.0. Potential coastal
polynyas are defined as regions near the coast that have SIC < 0.80.
To automate the mapping of different ice categories, radial transects from 50 to 90S are
individually selected to construct one-dimensional profiles [**Figure 1**]. The algorithm first steps
from the outer edge until the 0.15 SIC is detected, providing the latitude of the outer MIZ edge.
Next, the algorithm steps from the outer MIZ edge until either the 0.80 SIC is encountered, or the
continent is reached. Data points along the transect between these SIC thresholds are flagged as
the MIZ. In this way, the MIZ includes an outer band of low sea ice concentrations that
surrounds a band of inner consolidated pack ice, *but* sometimes the MIZ also extends all the way
to the Antarctic coastline (as sometimes observed in summer). South of the MIZ, the
consolidated ice pack (0.80 $\leq$ SIC $\leq$ 1.0) is encountered; however, low sea ice concentrations can
appear near the coast inside the pack ice region as well. These are areas of potential coastal polynyas. While it is difficult to measure the fine scale location of a polynya at 25km spatial
resolution, the lower sea ice concentrations provide an indication of some open water near the
coast, which for sea birds provides a source of open water for foraging. We have previously
tested mapping polynyas using a SIC threshold of 0.75 and 0.85 for the NASA Team and
Bootstrap algorithms, respectively, and found that these thresholds provided consistent polynya
areas between the two algorithms and matched other estimates of the spatial distribution of
polynyas [see *Li et al.,* 2016]. In another study, *Massom et al.* [1998] used a threshold of 0.75
applied to the NASA Team algorithm. However, for this study we chose just one threshold, a compromise between the two algorithms, so that we can better determine the sensitivity of using
the same threshold on polynya area and timing of formation.
Using our method of radial transects, the algorithm then steps from the coast northward and
flags pixels with < 0.80 SIC until a 0.80 SIC pixel appears and defines that region as a potential
coastal polynya. Within the consolidated pack ice (and away from the coast), it is also possible to
encounter instances where $0.15 < SIC < 0.80$ or $SIC < 0.15$. These are flagged as open pack ice
and open water areas within the consolidated pack ice, respectively. Finally, an ocean mask
derived from climatology and distributed by NSIDC was applied to remove spurious ice
concentrations at the ice edge as a result of weather effects.
**Figure 2** shows sample images of the classification scheme as applied to the NASA Team
and Bootstrap algorithms on days 70 (March 11) and 273 (September 30), respectively, in 2013.
During the fall and winter months when the ice cover is expanding there is a well-established
consolidated pack ice region, surrounded by the outer MIZ. Coastal polynyas are also found
surrounding the continent in both algorithms. The BT algorithm tends to show a larger
consolidated ice pack than NT, particularly during the timing of maximum extent. During the
melt season there is mixing of low and high ice concentrations, leading to mixtures of different
categories, which is still seen to some extent in the March images. However, during March areas
of polynyas (green), open water (pink) and open pack ice (orange) appear to extend from the
coastline in some areas (e.g. southern Weddell and Ross seas). While any pixel with SIC < 0.8
adjacent to the coastal boundary is flagged as potential polynya when stepping northwards, if a
pixel is already flagged as MIZ or consolidated pack ice when stepping southwards, it remains
flagged as MIZ or pack ice. After that analysis, a check for pixels with SICs less than 0.8 is done
to flag for broken ice or open water. Thus, during these months (e.g. December to February or
March), the physical interpretation of the different ice classes may be less useful.
Using the binary classification scheme, daily gridded fields at each 25 km pixel are obtained.
Using this gridded data set we then obtain regional averages using the true area per pixel for the

[revised manuscript text omitted]

pack ice trends are spatially consistent between both algorithms, though not all trends are
statistically significant, particularly for the NT algorithm. The largest consistency occurs in the
the western Ross Sea, where positive trends are seen in both algorithms, statistically significant
from March to November (p<0.01) in the BT algorithm, and from January to July and October to
November in the NT algorithm. Note also that both algorithms show statistically significant
positive trends in the MIZ from January to March in the western Ross Sea and generally negative
trends in the eastern Ross Sea. This pattern switches from June to December, with mostly
negative MIZ trends in the western Ross Sea and positive trends in the eastern Ross Sea. In
particular, the statistically significant positive trends in the MIZ in the NT algorithm occur at the
time of year with the largest overall trends in the SIE in this region. This would suggest perhaps
different interpretation of processes impacting the overall ice expansion in the Ross Sea
depending on which algorithm is used.

In the Bellingshausen/Amundsen Sea, statistically significant positive trends in pack ice are
limited to May through August in the NT algorithm and June and July in the BT algorithm. The
positive NT pack ice trends are offset by negative trends in the NT MIZ. Both algorithms exhibit
negative pack ice trends during other months that are consistent between the algorithms, though
larger in magnitude for the BT algorithm. This is generally compensated by statistically
significant negative trends in the NT MIZ to give an overall negative decline of total extent.
Trends in the pack ice are also consistent between algorithms in the Weddell Sea, with
statistically significant trends generally occurring at the same longitude and during the same
months. The positive pack ice trends in MAM (NT) or MAMJ (BT) are confined to a very
narrow longitude band which moves to the east with progressing season. Then in June, and
continuing for several months, negative pack ice trends occur. For both algorithms, trends in the
MIZ are generally not statistically significant, except for some positive trends in the eastern
Weddell Sea from January to March and negative trends mostly from June to November near
330 degrees longitude.
Finally, in the Pacific and Indian Oceans we again see spatial consistency in pack ice and
MIZ trends for both algorithms, with generally larger (smaller) pack ice (MIZ) trends for the BT
algorithm, though trends are closer in magnitude in the Pacific sector from March to July. Pack
ice trends are generally positive, more in BT than NT and trends in MIZ extent basically vary
around zero with exceptions during August through December in both algorithms in the Pacific
Ocean.
In summary, while the magnitude of trends differs between both algorithms, there is general
spatial consistency in the patterns of positive and negative trends in the consolidated pack ice
and the MIZ. Results suggest that positive trends in total SIE are generally a result of statistically
significant positive trends in the consolidated pack ice in the BT algorithm in all sectors of the
Antarctic, except for the Bellingshausen/Amundsen Sea sector and the Weddell Sea during ice
retreat. The NT algorithm on the other hand suggests more instances of statistically significant
positive trends in the MIZ, though this is highly regionally dependent.

**3.2.3 Seasonal Trends in MIZ and Pack Ice Width**
Finally, we compute the overall width of the MIZ and pack ice following *Strong and Rigor*
[2013] and produce seasonal means. Briefly, following the classification of each ice category,
latitude boundaries are computed for each longitude and each day. These are averaged for each
month to provide monthly mean latitude boundaries at each longitude. The boundaries are
subsequently converted to width in km, and averaged for all longitudes. Finally, seasonal means
are derived.
Time-series of seasonal means of the circumpolar MIZ width and pack ice width are shown
in **Figure 10** for all seasons except summer when the results are noisy. As we may expect
following the previous results, the NT MIZ width is larger and the pack ice width is smaller than
the seen in the BT algorithm. During autumn (MAM) however, the differences in widths for both
the MIZ and the pack ice between the algorithms are largely reduced compared to the other
seasons. For example the difference in 1979-2014 pack ice width between the algorithms during
MAM is 60 km, 121 km in JJA and 139 km in SON. Similarly, the long-term mean MIZ width
differences are 54 km (MAM), 74 km (JJA) and 83 km (SON). In addition, during autumn,
trends in the MIZ and pack ice are largely consistent between the two algorithms, with no trend
in the MIZ and increases in the pack ice on the order of 21.2 km dec$^{-1}$ and 20.0 km dec$^{-1}$
(p<0.01) for the BT and NT algorithms, respectively. This is the season with the largest trends in
the pack ice width, representing a 21% widening over the satellite record.

During winter (JJA) and spring (SON) however, the NT and BT algorithms exhibit opposing
trends in the MIZ with the NT algorithm indicating an increase, and the BT a decrease. The
largest positive trend in the MIZ width occurs during spring at a rate of +10.3 km dec$^{-1}$ (p<0.01)
in the NT algorithm, indicating a 6% widening since 1979. This widening is a result of the MIZ
moving slightly equatorward rather than expanding southwards (as also seen in Figure 7).
However, this is an increase of only about 1 to 1.5 grid cells over the entire data record, and
despite a statistically significant trend, there remains substantial interannual variability in the
SON MIZ width, with the maximum width recorded in 2003 (310 km) and the minimum in 1985
(217 km), with a mean SON MIZ width of 248 km. The trend during winter is considerably
smaller at +2.7 km dec$^{-1}$, as a result of expansion both equatorward and southwards, yet it is not
statistically significant.
For the pack ice, both sea ice algorithms show statistically significant positive trends towards
increased width of the pack ice, which are also nearly identical during winter at +18.7 and +18.1
km dec$^{-1}$ (p<0.01) for the BT and NT algorithms, respectively. This represents a widening of the
pack ice of approximately 11% from 1979 to 2014 during winter. As one may expect, differences
in the pack ice width between the algorithms are largely found in spring as a result of the MIZ
expanding in the NT algorithm. Therefore, during SON the trends in the width of the NT pack
ice are smaller, with trends of +10.0 (p<0.05) km dec$^{-1}$ compared to +16.7 (p<0.01) for the BT
algorithm.
Finally it is important to point out that the interannual variability in the pack ice is similar
between both data sets despite differences in magnitude. Correlations between the two
algorithms are: 0.96 (MAM), 0.92 (JJA) and 0.77 (SON). The reason for the weaker correlation
in SON is not entirely clear. For the MIZ, interannual variability is generally about twice as large
in the NASA Team algorithm and the two data sets are not highly correlated except for autumn,
with correlations of 0.67 (MAM), 0.39 (JJA) and 0.43 (SON).

**4. Implications for a Seabird**

Here we use data on the MIZ and the consolidated ice pack from both algorithms to
understand the role of sea ice habitat on breeding success of a seabird, the snow petrel
*Pagodroma nivea*. As mentioned in the introduction, the MIZ is a biologically important region
because it is an area of high productivity and provides access to food resources needed by
seabirds [*Ainley et al.*, 1992]. During winter, productivity is reduced at the surface in open water,
while it is concentrated within the ice habitat, especially within the ice floes [*Ainley et al.*, 1986].
This patchy distribution of food availability within the MIZ and pack ice provides feeding
opportunities for seabirds such as the snow petrel. Observations suggest that the snow petrel
forages more successfully in areas close to the ice edge and within the MIZ than in consolidated
ice conditions [*Ainley et al.*, 1984, 1992].
Breeding success of snow petrels depends on sufficient body condition of the females, which
in part reflects favorable environmental and foraging conditions prior to the breeding season.
Indeed, female snow petrels in poor early body condition are not able to build up the necessary
body reserves for successful breeding [*Barbraud and Chastel*, 1999]. Breeding success was
found to be higher during years with extensive sea ice cover during the preceding winter
[*Barbraud and Weimerskirch*, 2001]. This is in part because winters with extensive sea ice are
associated with higher krill abundance the following summer [*Flores et al.*, 2012; *Loeb et al.*,
1997; *Atkinson et al.*, 2004], thereby increasing the resource availability during the breeding season. However, extensive winter sea ice may protect the under ice community from predation
and thus reduce food availability, in turn affecting breeding success [*Olivier et al.,* 2005]. By
distinguishing between the areas of MIZ and pack ice, we can expect a better understanding of
the role of sea ice on food availability and hence breeding success of snow petrels.
In the following, we expect that an extensive pack ice during winter may reduce breeding
success the following breeding season by protecting the under ice community from predation,
while an extensive MIZ may increase breeding success by providing easier access to foraging.
With the classifications as defined by both algorithms we calculated the MIZ and pack ice area in
a wide rectangular sector defined by the migration route of the snow petrel [*Delord et al.,* 2016]
from April to September [see **Table 4** for latitude and longitude limits]. This is the first time that
appropriate areas of the observed foraging range are used to study the carry over effect of winter
conditions on the breeding performance of snow petrel, as this information did not
existed previously. Using these locations, we averaged the MIZ and pack ice extents over the
entire winter from April to September. We next employed a logistic regression approach to study
the effects of MIZ and pack ice area within this sector and evaluate the impacts on breeding
success the following summer. The response variable was the number of chicks $C_t$ in a breeding
season $t$, from 1979 to 2014 collected at Terre Adélie, Dumont D'Urville [*Barbraud and*
*Weimerskirch*, 2001, *Jenouvrier et al.*, 2005].
Effects of MIZ and pack ice area were analyzed using Generalized Linear Models (GLM)
with logit-link functions and binomial errors fitted in R using the package glm.
Specifically, the response variable is the number of chicks $C_t$ in a breeding season $t$, from 1979 to
2014 collected at Terre Adelie, Dumont D'Urville [*Barbraud and Weimerskirch*, 2001,
*Jenouvrier et al.*, 2005]. It follows a binomial distribution, such that $C_t \sim Bin(\mu_t, N_t)$, where $N_t$ is
the number of breeding pairs and $\mu_t$ is the breeding success in year $t$. The breeding success is a
function of the MIZ and pack ice covariates at time $t$ (COV) such as:

$$\mu_t = \beta_0 + \beta_1 COV_{(t)}$$

To select the covariate that most impacts the breeding success of snow petrels, we applied the
information-theoretic (I-T) approaches [*Burnham et al.*, 2011]. This is based on quantitative
measures of the strength of evidence for each hypothesis (Hi) rather than on "testing" null
hypotheses based on test statistics and their associated P values. To quantify the strength of
evidence for each hypothesis (Hi) – here the effect of each covariate on the breeding success-
we used the common criteria AIC (the Akaike's Information Criteria), where AIC = - 2 log(L) +
2K [*Akaike,* 1973]. The term, -2 log(L), is the "deviance" of the model, with log(L) the
maximized log-likelihood and K the total number of estimable parameters in the model. The
chosen model is the one that minimizes the AIC, in orther words, minimizes the Kullback-
Leibler distance between the model and truth. The ability of two models to describe the data was
assumed to be "not different" if the difference in their AIC was < 2 [*Burnham and Anderson*,
2002]. Note the AIC is a way of selecting a model from a set of models based on information
theory [*Burnham and Anderson*, 2002], and is largely used in biological sciences. While non-
linear models may be more appropriate as ecological system relationships are likely more
complex than linear relationships, without *a priori* knowledge of the mechanisms that could lead
to such non-linear relationships, it is extremely difficult to set meaningful hypothesis to be
included in the model selection.
**Table 5** summarizes model selection. The model with the lowest AIC (highlighted in gray)
suggests the BT pack ice as a sea ice covariate. If AIC are sorted from lowest to highest value, the next model includes the sea ice covariate MIZ calculated with the NASA algorithm.
However, it shows a ΔAIC ~8 from the best model, and thus the NT MIZ is not well supported
by the data in comparison to the best model. The relationship between BT pack ice and breeding
success is negative [**Figure 11**]. In other words, a more extensive consolidated pack ice during
winter tends to reduce breeding success the following summer by limiting foraging
opportunities. The effect of the MIZ however was uncertain, contrary to what one may expect
given the increased opportunities for foraging within the MIZ. However, if we had only used ice
classifications based on the NASA Team algorithm, the model with the lowest AIC would have
suggested an importance of the MIZ. We would have then concluded a negative effect of the
MIZ on the breeding success of snow petrels, contrary to what one may expect given that the
MIZ is the main feeding habitat of the species. By using both algorithms, we instead conclude
that the breeding success of snow petrels is negatively affected by the pack ice area as calculated
with the Bootstrap algorithm.

**5. Discussion**

While the main purpose for doing the classification of different ice categories is for
interdisciplinary studies of sea bird breeding success, the results may also be useful for
attribution of the observed sea ice changes. The positive trends in Antarctic sea ice extent are
currently poorly understood and are at odds with climate model forecasts that suggest the sea ice
should be declining in response to increasing greenhouse gases and stratospheric ozone depletion
[e.g. *Turner et al.*, 2013; *Bitz and Polvani*, 2012; *Sigmond and Fyfe*, 2010]. However, several
modeling studies, such as those used in the phase 5 Coupled Model Intercomparison Project
(CMIP5), have suggested that the sea ice increase over the last 36 years remains within the range
of intrinsic of internal variability [e.g. *Bitz and Polvani*, 2012; *Turner et al.*, 2013; *Mahlstein et
al.*, 2013; *Polvani and Smith,* 2013; *Swart and Fyfe*, 2013]. Earlier satellite from the 1960s and
1970s and from ship observations suggest periods of high and low sea ice extent, and thus high
natural variability [*Meier et al.*, 2013; *Gallaher et al.*, 2014]. Further evidence comes from ice
core climate records, which suggest that the climate variability observed in the Antarctic during
the last 50 years remains within the range of natural variability seen over the last several hundred
to thousands of years [*Thomas et al.*, 2013; *Steig et al.*, 2013]. Thus, we may require much
longer records to properly assess Antarctic sea ice trends in contrast to the Arctic, where negative
trends are outside the range of natural variability and are consistent with those simulated from
climate models.
While many assessments of how Antarctic sea ice trends and variability compare with
climate models have focused on the net circumpolar sea ice extent, it is the regional variability
that becomes more important. For example, *Hobbs et al.* [2015] argue that when viewing trends
on a regional basis, the observed summer and autumn trends fall outside of the range of natural
variability as simulated by present-day climate models, with the signal dominated by opposing
trends in the Ross Sea and the Bellingshausen/Amundsen seas. These results have questioned the
ability of climate models to correctly simulate processes at the regional level and within the
southern ocean-atmosphere-sea ice coupled system.
The net take-away point from these studies is that the net circumpolar changes in sea ice
extent do not enhance our understanding of how the Antarctic sea ice is changing. Instead our
focus should be on what drives regional and seasonal sea ice changes, including feedbacks and
competing mechanisms. The results of this study may help to better understand regional and total changes in Antarctic sea ice by focusing not only on the total ice area, but also on how the
consolidated pack ice, the marginal ice zone and coastal polynyas are changing. Differences in
climatologies and trends of the different ice classes may suggest different processes are likely
contributing to their seasonal and interannual variability. In addition, the different contributions
of ice categories towards the overall expansion of the Antarctic sea ice cover between algorithms
may in turn influence attribution of the observed increase in SIE. For example, within the highly
dynamic MIZ region, intense atmosphere-ice-ocean interactions take place [e.g. *Lubin and*
*Massom*, 2006] and thus an expanding or shrinking MIZ may help to shed light on the relative
importance of atmospheric or oceanic processes impacting the observed trends in total SIE.
Another issue is whether or not new ice is forming along the outer edge of the pack ice or if it is
all being dynamically transported from the interior.
However, a complication exists, what sea ice algorithm should be used for such assessments?
In this study we focused on using passive microwave satellite data for defining the different ice
categories used here as it is the longest time-series available and is not limited by polar darkness
or clouds. However, results are highly dependent on which sea ice algorithm is used to look at
the variability in these ice classes, which will also be important in assessing processes
contributing to these changes as well as implications of these changes to the polar marine
ecosystem. In this study, the positive trends in circumpolar sea ice extent over the satellite data
record are primarily driven by statistically significant trends ($p<0.05$) in expansion of the
consolidated pack ice in both sea ice algorithms. However, an exception occurs in the NASA
Team sea ice algorithm after the ice pack reaches its seasonal maximum extent when the positive
trends in the pack ice are no longer as large, nor statistically significant. Instead, positive trends
in the MIZ dominate during September and October ($p<0.10$). This is in stark contrast to the
Bootstrap algorithm, which shows a declining MIZ area from March through November.
The algorithms also give different proportions of how much the total ice cover consists of
consolidated ice, MIZ or polynya area. In some regions, such as the Pacific Ocean sector, the NT
algorithm suggests the MIZ is the dominant ice category whereas in the BT algorithm, the pack
ice is dominant, which is true for all sectors analyzed in the Bootstrap algorithm. Considering the
circumpolar ice cover, the MIZ in the NASA Team algorithm is on average twice as large as in
the Bootstrap algorithm. In the Arctic, *Strong and Rigor* [2013] found the NASA Team
algorithm gave about three times wider MIZ than the Bootstrap algorithm. In this case, the
Bootstrap results agreed more with MIZ widths obtained from the National Ice Center (NIC).
While we find consistency in trends in pack ice and the MIZ, there are some important
differences that may influence interpretation of processes governing sea ice changes. For
example, in the Ross Sea, the largest regional positive trends in total SIE are found at a rate of
119,000 km$^2$ per decade [e.g. *Turner et al*., 2015], accounting for about 60% of the circumpolar
ice extent increase. This is entirely a result of large positive trends in the pack ice in the BT
algorithm from March to November ($p<0.01$) whereas the NT algorithm shows statistically
significant increases in the MIZ. Several studies have suggested a link between sea ice anomalies
in the Ross Sea and the wind-field associated with the Amundsen Sea Low (ASL) [e.g. *Fogt et*
*al*., 2012; *Hosking et al*., 2013; *Turner et al*., 2012]. The strengthened southerly winds over the
Ross Sea cause a more compacted and growing consolidated ice cover in the BT algorithm at the
expense of a shrinking MIZ, whereas in the NT algorithm the area of the MIZ is increasing more
than the pack ice during autumn, which may suggest a smaller sensitivity to thin ice growing in
openings and leads for BT than for NT. While this is true as averaged over the entire Ross Sea
sector, Figure 9 highlights that the area-averaged trends hide important spatial variability.

[revised manuscript text omitted]

**Conclusions**

Antarctic sea ice plays an important role in the polar marine ecosystem. While total Antarctic
sea ice cover is expanding in response to atmospheric and oceanic variability that remains to be
fully understood, one may expect that these increases would also be manifested in either
equatorward progression of the MIZ or the consolidated pack ice or both, that in turn would
impact the entire trophic web, from primary productivity, to top predator species, such as
seabirds. In this study we identified several different ice categories using two different sets of
passive microwave sea ice concentration data sets. The algorithms are in agreement as to the
location of the northern edge of the total sea ice cover, but differ in regards to how much of the
ice cover consists of the marginal ice zone, the consolidated ice pack, the size of potential
polynyas as well as the amount of broken ice and open water within the consolidated ice pack.
Here we use sea ice concentration thresholds of $0.15 \leq SIC < 0.80$ to define the width of the MIZ
and $0.80 \leq SIC \leq 1.0$ to define the consolidated pack ice. Yet applying the same thresholds for
both sea ice algorithms results in a MIZ from the NASA Team algorithm that is on average twice
as large as in the Bootstrap algorithm and considerably more broken ice within the consolidated
pack ice. Total potential coastal polynya areas ($SIC \leq 0.80$) also differ between the algorithms,
though differences are generally smaller than for the MIZ and the consolidated pack ice. While
we do not precisely resolve polynyas, these potential coastal polynyas (i.e. open water areas near
the coast) are important foraging sites for sea birds.
While the spatial extents of the different ice classes may differ, the seasonal cycle is
generally consistent between both algorithms. Climatologically, the advance of the consolidated
ice pack happens over a much longer period (~7-8 months) than the retreat (~4-5 months), while
the MIZ exhibits a longer advance period (~8-10 months). This seasonal cycle in
expansion/contraction of the ice cover is in general agreement with results by *Stammerjohn et al.*

[2008] who showed sea ice retreat begins in September at the outer most edge of the sea ice and
continues poleward over the next several months. However, what these results show is that while
the pack ice starts to retreat around September, this in turn results in a further expansion of the
MIZ, the amount of which is highly dependent on which algorithm is used. The timing of when
the maximum polynya extent is reached however can differ by several months between the
algorithms in regions such as the Bellingshausen/Amundsen Sea and the Pacific Ocean.
Since the MIZ is an important region for phytoplankton biomass and productivity [e.g. *Park*
*et al.,* 1999], mapping seasonal and interannual changes in the MIZ is important for
understanding changes in top predator populations and distributions. However, as we show in
this study, results are highly dependent on which sea ice algorithm is used for delineating the
MIZ, which may result in different conclusions when using this data in ecosystem models. To
hightlight this sensitivity, we examined the impact the winter MIZ and consolidated pack ice
area as derived from both algorithms would have on the breeding success of snow petrels the
following summer. The different proportions of MIZ and consolidated pack ice between
algorithms affected the inferences made from models tested even if trends were of the same sign.
Given the sensitivity of the relationships between the consolidated pack ice/MIZ and breeding
success of this species, caution is warranted when doing this type of analysis as different
relationships may emerge as a function of which sea ice data set is used in the analysis. Further
work is needed to validate the accuracy of the distribution of the MIZ and consolidated pack ice
from passive microwave so that the data will be more useful for future biological and ecosystem
studies.

**Acknowledgements**
This work is funded under NASA Grant NNX14AH74G and NSF Grant PLR 1341548. We are
grateful to Sharon Stammerjohn for her helpful comments on the manuscript. Gridded fields of
the different ice classifications from both algorithms are available via ftp by contacting J.
Stroeve. We thank all the wintering fieldworkers involved in the collection of snow petrel data at
Dumont d'Urville since more than 50 years, as well as Institut Paul Emile Victor (program IPEV
n°109, resp. H. Weimerskirch), Terres Australes et Antarctiques Françaises and Zone Atelier
Antarctique (CNRS-INEE) for support.

and primary production in the marginal ice zone of the northwestern Weddell Sea during
austral summer, *Polar Biol., 21*, *251*–261.

[revised manuscript text omitted]

**List of Figures**

[Figure]

**Figure 1.** Example of a radial profile from 50 to 90S at -11.60 degrees West on 3 September
1990, showing the different sea ice classifications found along this transect.

[Figure]

Figure 2: Samples of ice classification on day 70 (March) and day 273 (September) 2013.
Results are shown for both the NASA Team (top) and Bootstrap (bottom) sea ice algorithms. The
MIZ (red) represents regions of sea ice concentration between 15 and 80% from the outer ice
edge, the pack ice is shown in light purple, representing regions of greater than 80% sea ice
concentration. Orange regions within the pack ice represent coherent regions of less than 80%
sea ice concentration, pink areas open water and green regions of less than 80% sea ice
concentration near the Antarctic coastline. Dark blue represents the ocean mask applied to
remove spurious ice concentrations beyond the ice edge.

[Figure]

**Figure 3.** Southern hemisphere regions as defined by *Parkinson and Cavalieri* [2012].

[Figure]

**Figure 4**. Location of the mean 1981-2010 outer marginal ice edge for both the NASA Team and
Bootstrap algorithms.

[Figure]

**Figure 5.** Long-term (1979-2014) and standard deviation (shading) of the seasonal cycle in total
Antarctic extent of the consolidated pack ice, the outer marginal ice zone, polynyas, open pack
ice (or broken ice within the pack ice), and inner open water. There are essentially no scattered
ice floes outside of the MIZ. NASA Team results are shown on the left and the Bootstrap on the
right.

[Figure]

**Figure 6.** Long-term (1979-2014) seasonal cycle in regional sea ice extent of the consolidated pack ice, the outer marginal ice zone, polynyas, open pack ice (or broken ice within the pack ice), and inner open water. Results for the NASA Team algorithm are shown on the left and Bootstrap on the right, and for the Ross, Bellingshausen/Amundsen, Weddell, Indian and Pacific Oceans.

[Figure]

**Figure 7.** Expansion (red) or contraction (blue) of the outer ice edge (top), the width of the
marginal ice zone (middle) and the width of the pack ice from 1979 to 2014 during the month of
September relative to 60S.

[Figure]

**Figure 8.** Daily trends (1979 to 2014) in the consolidated pack ice, the outer MIZ and potential
coastal polynyas for the entire Antarctic sea ice cover for the NASA Team (left) and Bootstrap
(right) algorithms. Trends are provided in $10^6$ km$^2$ a$^{-1}$.

[Figure]

**Figure 9.** Daily (1979-2014) trends in regional sea ice extent of the consolidated pack ice (top)
and the outer marginal ice zone (bottom). Results for the NASA Team algorithm (left) and
Bootstrap (right) are shown as a function of longitude. Trends are provided in $10^6$ km$^2$ a$^{-1}$. Note
the difference in color bar scales. Regions not statistically significant are highlighted.

[Figure]

**Figure 10.** Time-series of seasonal mean MAM (top), JJA (middle) and SON (bottom) marginal
ice zone (left) and consolidated pack ice (right) for both sea ice algorithms; NASA Team is
shown in red, Bootstrap in black. Shading represents one standard deviation. Note the difference
in y-axis between the pack ice and the MIZ plots.

[Figure]

[Figure]

**Figure 11.** Breeding success of snow petrel (top) since the 1960s and the effect of the Bootstrap
consolidated pack ice area (x-axis) on the breeding success of snow petrels (y-axis) (bottom).